# ScaLearn: Simple and Highly Parameter-Efficient Task Transfer by Learning to Scale

## Abstract

Multi-task learning (MTL) has shown considerable practical benefits, particularly when using pre-trained language models (PLMs). While this is commonly achieved by simultaneously learning $n$ tasks under a joint optimization procedure, recent methods such as AdapterFusion structure the problem into two distinct stages: (i) task learning, where knowledge specific to a task is encapsulated within sets of parameters (e.g., adapters), and (ii) transfer, where this already learned knowledge is leveraged for a target task. This separation of concerns provides numerous benefits, such as promoting reusability, and addressing cases involving data privacy and societal concerns; on the flip side, current two-stage MTL methods come with the cost of introducing a substantial number of additional parameters. In this work, we address this issue by leveraging the usefulness of linearly scaling the output representations of source adapters for transfer learning. We introduce ScaLearn, a simple and highly parameter-efficient two-stage MTL method that capitalizes on the knowledge of the source tasks by learning a minimal set of scaling parameters that enable effective knowledge transfer to a target task. Our experiments on three benchmarks (GLUE, SuperGLUE, and HumSet) show that our ScaLearn, in addition to facilitating the benefits of two-stage MTL, consistently outperforms strong baselines with only a small number of transfer parameters – roughly 0.35% of those of AdapterFusion. Remarkably, we observe that ScaLearn maintains its strong abilities even when further reducing parameters through uniform scaling and layer-sharing, achieving similarly competitive results with *only 8 transfer parameters* for each target task. Our proposed approach thus demonstrates the power of simple scaling as a promise for more efficient task transfer.[1]

## 1 Introduction

With the wide availability of pre-trained language models (PLMs) as the backbone of language processing, multi-task learning (MTL) has shown significant benefits, especially for tasks with possible conceptual commonalities (Ruder, 2017; Zhang & Yang, 2022; Raffel et al., 2020). The traditional paradigm in MTL is to formulate a joint optimization objective based on a set of tasks and train a single model to simultaneously learn and transfer the knowledge relevant to the tasks. This *joint MTL* approach can be realized by fine-tuning a PLM (Liu et al., 2019a; Stickland & Murray, 2019), or, more recently, by using parameter-efficient, often modularized, MTL approaches (Mahabadi et al., 2021b; Zeng et al., 2023; Pilault et al., 2021; Asai et al., 2022; Ponti et al., 2023; Caccia et al., 2022).

As an alternative to the joint MTL paradigm, some works such as ADAPTERFUSION (Pfeiffer et al., 2021) clearly distinguish task training from transfer learning, assigning dedicated parameters to each of these aspects. In this paradigm, referred to as *two-stage MTL*, first each *source task* is trained separately and stored into a separate module like an adapter (Houlsby et al., 2019), and then a task transfer layer is trained for a given *target task* using information from an *arbitrary* set of source tasks. This separation of concerns between task and transfer learning offers valuable benefits: (1) Learning a separate transfer layer for each target task in a two-stage MTL approach reduces the potentially destructive effects of transfer learning on specific tasks, as the transfer layer parameters corresponding to each target task can independently decide what information should be used from the available source tasks. As shown in our experiments, this supports the effectiveness of transfer learning, making it less sensitive to task selection. (2) Since the source tasks can simply be taken from already trained modules (no need for re-training), two-stage approaches particularly promote

---

[1]Our code is available at *URL upon deanonymization*.

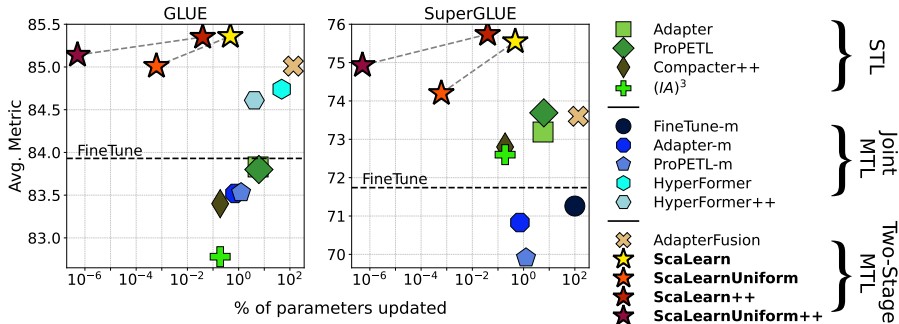

**Figure 1:** Performance and parameter-efficiency of single task learning (STL), and joint/two-stage MTL methods, evaluated on GLUE (Wang et al., 2019b) and SuperGLUE (Wang et al., 2019a) using RoBERTa$_{\text{BASE}}$ (Liu et al., 2019b). The reported values for the two-stage MTL methods only consider the ones in the respective transfer layers. The full details of the learnable parameters and performance results are provided in Section 6.

reusability – a principle of Green AI (Scells et al., 2022; Schwartz et al., 2020). Further, they provide a practical solution to cases involving issues such as data privacy and/or fairness constraints, as a pre-trained module can readily provide the (e.g., already debiased) functionality of the source task even without the need to have access to its training data (Lauscher et al., 2021; Kumar et al., 2023).

Despite these benefits, current two-stage MTL solutions introduce significantly more learnable parameters in comparison with recent joint MTL ones, exacerbated by the fact that the number of parameters in two-stage methods increases linearly with the number of target tasks. As an example, in our experiment setup with eight target tasks using RoBERTa$_{\text{BASE}}$ (Liu et al., 2019b), ADAPTER-FUSION introduces $\sim 134\%$ new parameters for transfer learning, while HYPERFORMER++ (Mahabadi et al., 2021b) conducts joint MTL by adding $\sim 4\%$ (around 5 Million) trainable parameters (details in Table 1 and Section 6). To date, this high number of parameters requiring optimization is in stark contrast to the promise of green AI given by the modularized nature of two-stage MTL.

In this work, we propose a highly parameter-efficient and effective two-stage MTL method by scaling the output representations of source adapters using encoder PLMs. Learning scaling vectors applied to input representations has recently been introduced to fulfill various objectives such as task learning, domain adaptation, and bias mitigation (Liu et al., 2022; Ilharco et al., 2023; Masoudian et al., 2023). In the work at hand, we first analyze the effect of scaling output vectors of source adapters on transfer learning, examined by linearly probing the performance on a given target task. We observe that (1) the degree of scaling of source adapter representations is not necessarily linearly correlated with the transfer learning performance on a target task; (2) when summing two scaled adapter representations, the optimal scaling coefficients often do not sum up to 1. Building on these findings, we introduce SCALEARN, a novel two-stage MTL method that learns to transfer the knowledge of the source adapters using a small set of scaling parameters. For a given target task, SCALEARN introduces a set of parameters that scale the output representation of each source adapter and combine the resulting scaled representations by simply taking the element-wise sum. SCALEARN learns to apply a (linear) scaling transformation without imposing any constraint on the relation of the scaling coefficients across source tasks, where the parameters are optimized using common gradient descent methods. This approach results in high parameter-efficiency, such that – following the mentioned experiment setting – SCALEARN only adds $\sim 0.47\%$ (around 0.5 million) new parameters. We further introduce an even more parameter-efficient variation through uniform scaling (SCALEARNUNIFORM), where each scaling vector is reduced to a single scaling parameter. Finally, by sharing the parameters across the layers, we achieve our most parameter-efficient variation (SCALEARNUNIFORM++), only containing 64 parameters for transfer learning.

We conduct a large set of transfer learning experiments on the GLUE (Wang et al., 2019b), SuperGLUE (Wang et al., 2019a), and HumSet (Fekih et al., 2022) benchmarks using the RoBERTa model (BASE and LARGE) (Liu et al., 2019b), and compare the parameter-efficiency and performance of SCALEARN with strong joint and two-stage MTL baselines. Figure 1 summarizes our results on GLUE and SuperGLUE. Our results show that SCALEARN, while providing high efficiency and the benefits of the two-stage MTL paradigm, consistently outperforms the baselines. Interestingly, the overall performance of SCALEARN remains highly competitive and only marginally different in its more parameter-efficient variations. Our results also show the advantage of two-stage models in avoiding destructive effects during transfer learning, particularly on the SuperGLUE and HumSet benchmarks (cf. Section 6). Finally, SCALEARN exhibits strong performance in few-shot settings,

outperforming both regular adapters and ADAPTERFUSION when trained only on a handful of data points. Overall, with SCALEARN we leverage the power of scaling as a viable, non-destructive, simple-to-implement, and highly parameter-efficient solution to the current shortcomings of two-stage and joint MTL methods, paving the future for more effective and efficient task transfer.

## 2 BACKGROUND

In task transfer learning, we consider a PLM as well as two sets $S$ and $T$, representing the source and target tasks, respectively. The aim of MTL is to leverage the information of tasks in $S$ to improve the generalization on tasks in $T$.

**Single Task Learning (STL).** In this basic setting, a separate set of parameters is optimized on each task ($S = T$) without any knowledge transfer between tasks. STL can be done by fine-tuning the PLM parameters or by introducing more parameter-efficient modules into the model, such as adapter modules (Pfeiffer adapters (Houlsby et al., 2019; Pfeiffer et al., 2021), PROPETL (Zeng et al., 2023), or COMPACTER++ (Mahabadi et al., 2021a)), $(IA)^3$ (Liu et al., 2022), prefix-tuning (Li & Liang, 2021), or LoRA (Hu et al., 2022), each with $\Theta_s$ parameters for each task $s$.

**Joint MTL.** This approach is commonly done by having a unified model for all tasks ($S = T$), and a joint optimization objective that simultaneously optimizes the model using samples from all tasks (Ruder, 2017). The general joint MTL objective can be formulated as $\mathcal{L}_{\text{joint}} = \sum_{s=1}^{|S|} \alpha_s \mathcal{L}_s$, where $\alpha_s$ is the sampling weight of task $s$. This optimization objective can be used to fine-tune the parameters of a PLM (Liu et al., 2019a; Stickland & Murray, 2019; Raffel et al., 2020), or those of a modularized architecture (Mahabadi et al., 2021b; Pilault et al., 2021; Ponti et al., 2023). Despite the benefit of having one unified model, the joint loss often causes tasks to compete with each other for learning capacity, leading to the *task interference problem* (Xin et al., 2022; McCloskey & Cohen, 1989; Kirkpatrick et al., 2017). This makes the joint MTL paradigm particularly sensitive to the selection of tasks (Xin et al., 2022), while various methods in the literature have aimed to address this issue (e.g., Kendall et al. (2018); Pilault et al. (2021); a brief review is provided in Section 7).

**Two-stage MTL.** In contrast to joint MTL, two-stage MTL methods optimize each target task independently, bypassing the issue of task interference (Pfeiffer et al., 2021). Similarly to STL, a parameter-efficient module is first learned for each *source* task $s$ with parameters $\Theta_s$. In principle, two-stage MTL methods can simply use already pre-trained modules (such as adapters), saving the costs of re-training modules on each task. This facilitates the re-use of existing parameter-efficient modules for each source task[2], which may vary in performance and/or take into account additional constraints such as fairness and bias mitigation (Pfeiffer et al., 2023; Kumar et al., 2023; Lauscher et al., 2021). Moreover, it also removes the need for accessing the training data of the source tasks (e.g., due to data privacy), so far as the source task's functionality is solely provided via parameter-efficient modules. Next, given $|S|$ (pre-trained and frozen) source task modules, two-stage MTL methods define and optimize a transfer layer for each target task to leverage the knowledge of source tasks to solve the target task. This stage introduces $\Omega_t$ new parameters for each target task $t$.

ADAPTERFUSION (Pfeiffer et al., 2021) introduces an implementation of the two-stage approach with strong MTL performance (Pfeiffer et al., 2023). It uses an attention mechanism as its transfer layer, inserted into each layer of the PLM, after the source adapters. More specifically, given the output vector of each source adapter $s$ in each layer $l$, referred to as $\boldsymbol{o}_s^l$, the attention layer (with target task $t$ as query and source tasks $S$ as keys and values) learns to assign a weight $\omega_s^l$ to each source task. The final output of the target task $t$ in this layer is calculated as:

$$\boldsymbol{o}_t^l = \sum_{s=1}^{|S|} \omega_s^l \boldsymbol{o}_s^l, \quad \text{where} \sum_{s=1}^{|S|} \omega_s^l = 1 \tag{1}$$

Regardless of how the weights are calculated, the method can be seen as a weighted summation of source output vectors, where the weights form a categorical probability distribution. In the following section, we provide an analysis on the effect of these weights in transfer learning.

## 3 ANALYSIS ON SCALING OUTPUT REPRESENTATIONS

We seek to leverage simple scaling as a novel composition method in transfer learning. To understand the effect of scaling, we now conduct preliminary experiments in which we scale the output

---

[2]E.g., through sharing platforms such as AdapterHub (`https://adapterhub.ml/`) (Pfeiffer et al., 2020).

representations of adapters – in isolation and combining two of them each. We use the popular GLUE (Wang et al., 2019b) and SuperGLUE (Wang et al., 2019a) benchmarks, utilizing a selection of their tasks (owing to the high number of possible combinations), including entailment, paraphrase detection, sentiment analysis, question answering, and commonsense reasoning tasks. We train a Pfeiffer adapter (Pfeiffer et al., 2021) on each task using the encoder PLM RoBERTa$_{\text{BASE}}$ (Liu et al., 2019b). In our probing-like setup (Tenney et al., 2019), we freeze both the PLM and adapter weights and train a new task head on target task $t$ each time we change the scaling factor. Complete descriptions of the datasets, hyperparameters, and training procedure are provided in Section 5 and Appendix A.1. Additional experiments and results on further tasks are provided in Appendix A.2.

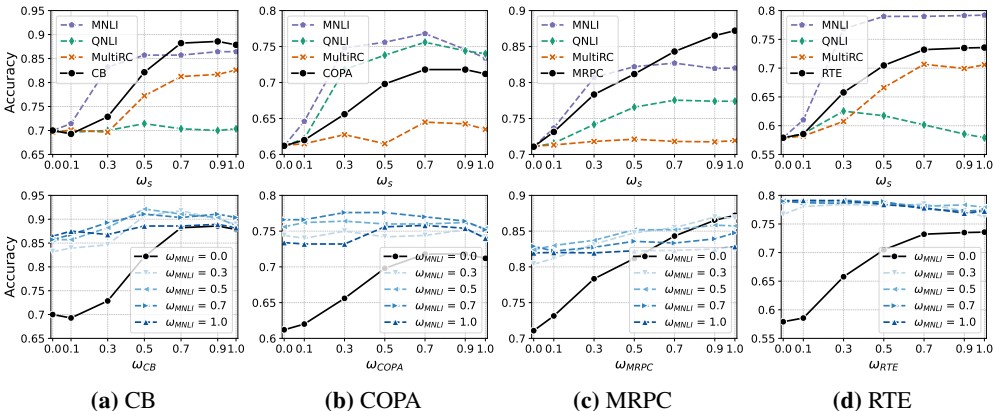

**(a)** CB          **(b)** COPA          **(c)** MRPC          **(d)** RTE

**Figure 2:** Probing results of 4 target tasks in various transfer learning conditions. (Top) Effect of scaling the output representations of adapters by weight $\omega_s$ using different source adapters. (Bottom) Effect of combining independently scaled output representations of two adapters trained on the target task and MNLI, respectively. Each point shows the mean over 5 seeds. Full results are reported in Appendix A.2.

We start by analyzing the performance change of a target task when scaling the output representations of the adapter of *one* given source task. We define $\omega_s$ as the scaling value in the range of $[0, 1]$, multiplied by the output representations $\boldsymbol{o}_s^l$ of the source task $s$ in all layers, such that $\boldsymbol{o}_t^l = \omega_s \boldsymbol{o}_s^l$. Figure 2 (Top) shows the probing results on four target tasks (each column), given various scaling weights applied to four source tasks (one of which is the respective target task). The results show that, while increasing the scaling weights generally improves the performance, the optimal value is not necessarily at $\omega_s = 1$. In particular, there exist instances with $0 < \omega_s < 1$ reaching better performance than $\omega_s = 1$. This suggests that *partial knowledge transfer* of tasks may be more beneficial. Notably, and as also reported in previous studies (Poth et al., 2021; Pruksachatkun et al., 2020), some source tasks such as MNLI show strong transfer learning abilities.

Next, we go one step further by assessing the scaled *combination* of the output vectors of two adapters. We focus on MNLI as one of the source tasks given its observed benefit in transfer learning, and set the second source adapter (denoted by $s$) to the one corresponding to the target task. We use two scaling parameters $\omega_{\text{MNLI}}$ and $\omega_s$ to scale $\boldsymbol{o}_{\text{MNLI}}^l$ and $\boldsymbol{o}_s^l$, respectively. The resulting output vector is defined as: $\boldsymbol{o}_t^l = \omega_s \boldsymbol{o}_s^l + \omega_{\text{MNLI}} \boldsymbol{o}_{\text{MNLI}}^l$. Figure 2 (Bottom) shows the results for various values of $\omega_{\text{MNLI}}$ and $\omega_s$. Combining the information encapsulated within multiple adapters through scaling can result in improved performance. Interestingly, in some cases, the best combination of $\omega_{\text{MNLI}}$ and $\omega_s$ does not add up to 1, i.e., $\omega_t + \omega_s \neq 1$. This finding stands in contrast to the established practice of forcing the scaling coefficients to sum up to 1 (e.g., as in ADAPTERFUSION, as shown in Eq. 1).

These initial experiments – while only covering a simple combination of up to two source tasks – provide insights into the benefits of scaling representations for transfer learning: (1) scaling output vectors is an effective method for controlling the (partial or full) activation of the knowledge contained in an adapter module; (2) an optimal configuration of the scaling parameter will, in many cases, lead to superior results on the target task; (3) the optimal weights do not necessarily sum up to 1. These observations provide strong motivation for designing a method to *combine* representations from several adapters by scaling their output vectors, presented in the next section.

## 4   SCALEARN – LEARNING TO SCALE FOR KNOWLEDGE TRANSFER

Building on our findings from Section 3, we present SCALEARN, a novel two-stage transfer learning method to combine the knowledge of source adapters by scaling their output representations.

Our core contribution regards the transfer layer, built on the output of the tasks' modular networks. Similar to Pfeiffer et al. (2021), we utilize adapter modules for the task learning layer. In particular, the output representation of the adapter of source task $s$ at layer $l$ is defined as: $\boldsymbol{o}_s^l = \boldsymbol{U}_s^l(\text{ReLU}(\boldsymbol{D}_s^l(\boldsymbol{x}_s^l))) + \boldsymbol{x}_s^l$, where $\boldsymbol{x}_s^l$ is the input vector, and $\boldsymbol{U}_s^l$ and $\boldsymbol{D}_s^l$ denote the up- and down-projection parameter matrices, respectively.

Our introduced SCALEARN linearly scales and combines the output representations of the source adapters, $\boldsymbol{o}_1^l, \ldots, \boldsymbol{o}_{|S|}^l$, to achieve the objective of target task $t$. We define two variations of the scaling operation: *non-uniform* which applies a scaling vector to each output vector using the element-wise product (SCALEARN), and the more parameter-efficient *uniform* that scales each vector only with a scalar parameter (SCALEARNUNIFORM). These variations are formulated below:

$$\text{SCALEARN}: \ \boldsymbol{o}_t^l = \sum_{s=1}^{|S|} \boldsymbol{\omega}_s^l \odot \boldsymbol{o}_s^l \qquad \text{SCALEARNUNIFORM}: \ \boldsymbol{o}_t^l = \sum_{s=1}^{|S|} \omega_s^l \boldsymbol{o}_s^l, \qquad (2)$$

where $\odot$ denotes the Hadamard product, and $\boldsymbol{\omega}_s^l$ and $\omega_s^l$ are learnable vector and scalar parameters, respectively. Inspired by previous studies (Mahabadi et al., 2021a; Zeng et al., 2023; Bai et al., 2022; Goldberg, 2019; Jawahar et al., 2019), we further increase parameter-efficiency by learning shared scaling parameters among all layers, formulated as follows:

$$\text{SCALEARN++}: \ \boldsymbol{o}_t^l = \sum_{s=1}^{|S|} \boldsymbol{\omega}_s \odot \boldsymbol{o}_s^l \qquad \text{SCALEARNUNIFORM++}: \ \boldsymbol{o}_t^l = \sum_{s=1}^{|S|} \omega_s \boldsymbol{o}_s^l, \qquad (3)$$

where, similarly, $\boldsymbol{\omega}_s$ and $\omega_s$ are learnable vector and scalar parameters, but shared among all layers. In all the mentioned methods, to optimize the transfer parameters $\Omega$, we use gradient descent as an easy-to-implement and straightforward solution. On the basis of our experiments, we find that our approach provides highly competitive results on a wide range of tasks (cf. Section 6). Furthermore, we emphasize that SCALEARN models do not force any distributional properties on the $\omega$ values, as commonly imposed in previous work Pfeiffer et al. (2021); Chronopoulou et al. (2023); Xin et al. (2022) through functions such as softmax and average.

**Parameter-efficiency of SCALEARN.** To have a clear view of the parameter-efficiency of the introduced models, we continue by analyzing the number of learnable parameters in the transfer layer. The SCALEARN variant introduces $d \times L \times |S|$ transfer parameters for a single target task, where $d$ is the embedding size and $L$ denotes the number of layers. The overall number of parameters for all target tasks then becomes $d \times L \times |S| \times |T|$. Moving to SCALEARNUNIFORM, this number reduces to $L \times |S| \times |T|$. The SCALEARN++ spares the $L$ term and has $d \times |S| \times |T|$ transfer parameters. Finally, the most parameter-efficient variant SCALEARNUNIFORM++ only adds $|S| \times |T|$ parameters. Note that the new task head parameters are learned jointly with the transfer parameters for each task.

As a point of comparison, the number of transfer parameters of ADAPTERFUSION is $3 \times d^2 \times L \times |T|$ (discarding bias and task head parameters), corresponding to the query, key, and value matrices of the attention mechanism. Comparing the formulas, we observe that our methods are far more parameter-efficient, since in practice $|S| \ll d$, and hence the $d \times L$ term in SCALEARN becomes much smaller than $d^2$ in ADAPTERFUSION. Compared to the joint MTL paradigm, despite the linear increase of parameters with $|T|$, our SCALEARN* models still provide high parameter-efficiency. This stems from the fact that $|T| \ll d$, and hence reducing the effect of $d$ – which is fully eliminated in the uniform variants – leaves a more significant impact on parameter-efficiency.

## 5 EXPERIMENT SETUP

**Tasks and datasets.** We conduct our experiments on the GLUE and SuperGLUE benchmarks, respectively, each consisting of 8 tasks, as well as on the HumSet (Fekih et al., 2022) benchmark. HumSet is a multilingual classification dataset for humanitarian crisis response consisting of 5 tasks. Additionally, we use a combination of *all* GLUE and SuperGLUE tasks resulting in 15 datasets[3].

**PLM backbones.** We use RoBERTa$_{\text{BASE}}$ and RoBERTa$_{\text{LARGE}}$ (Liu et al., 2019b) on GLUE and SuperGLUE. For the experiments on HumSet, following (Fekih et al., 2022) we utilize the multilingual XLM-R$_{\text{BASE}}$ and XLM-R$_{\text{LARGE}}$ (Conneau et al., 2020) as this dataset consists of multiple languages.

**Models and baselines.** We conduct experiments on four variants of our model, namely **SCALEARN**, **SCALEARNUNIFORM**, **SCALEARN++**, and **SCALEARNUNIFORM++**. As a direct baseline, we

---

[3]The RTE task is contained in both GLUE and SuperGLUE.

**Table 1:** Percentage and number of trainable parameters per model (excluding task head parameters), when training on 8 tasks (as in GLUE/SuperGLUE) using RoBERTa$_{BASE}$. For two-stage MTL, source and target tasks are the same ($|S|=|T|=8$), and the task parameters consist of $|S|$ adapters, thus $\Theta = 8 \times 0.72\% = 5.74\%$.

| Category | Model | Parameters (one task) | Parameters (all tasks) | |
|---|---|---|---|---|
| STL | FINETUNE | 100.00% (125M) | 800.00% (125M) | - |
| | ADAPTER | 0.72% (895K) | 5.74% (7M) | - |
| | PROPETL | 0.77% (959K) | 6.16% (8M) | - |
| | COMPACTER++ | 0.02% (29K) | 0.19% (235K) | - |
| | $(IA)^3$ | 0.05% (57K) | 0.37% (455K) | - |
| Joint MTL | FINETUNE-M | - | 100.00% (125M) | - |
| | ADAPTER-M | - | 0.72% (895K) | - |
| | PROPETL-M | - | 1.24% (1.5M) | - |
| | HYPERFORMER | - | 47.67% (59M) | - |
| | HYPERFORMER++ | - | 4.09% (5M) | - |
| | | Transfer ($\Omega_t$) (target task $t$) | Transfer ($\Omega$) (all target tasks) | Task ($\Theta$) + Transfer ($\Omega$) (source adapters + transfer layers) |
| Two-Stage MTL | ADAPTERFUSION | 17.05% (21M) | 136.40% (170M) | $5.74\% + 136.40\% = 142.14\%$ (177M) |
| | SCALEARN | 0.06% (74K) | 0.47% (590K) | $5.74\% + 0.47\% = 6.21\%$ (8M) |
| | SCALEARNUNIFORM | 0.00% (96) | 0.00% (768) | $5.74\% + 0.00\% = 5.74\%$ (7M) |
| | SCALEARN++ | 0.00% (6K) | 0.04% (49K) | $5.74\% + 0.04\% = 5.79\%$ (7M) |
| | SCALEARNUNIFORM++ | 0.00% (8) | 0.00% (64) | $5.74\% + 0.00\% = 5.74\%$ (7M) |

compare our models with **ADAPTERFUSION**, a common two-stage MTL method that shares the same conceptual properties. We also compare our models with **ADAPTERSOUP** (Chronopoulou et al., 2023), performing weight-space averaging over adapter weights of the 5 most similar tasks according to their sentence similarity. We adapt their approach to our setup (cf. Appendix A.1). In all two-stage MTL methods, source and target tasks are the same, containing the tasks of the underlying benchmark. For each target task, they learn a transfer layer (except for **ADAPTERSOUP**) and a new task head.

We select a set of strong STL baselines: **FINETUNE**, fully fine-tuning the PLM, **ADAPTER** Houlsby et al. (2019) learning an adapter module for each task, **PROPETL** (Zeng et al., 2023) a more memory-efficient variation based on parameter sparsification and **COMPACTER++** (Mahabadi et al., 2021a) a highly parameter-efficient variation that leverages parameter-sharing between layers. As another STL baseline, we train $(IA)^3$ (Liu et al., 2022), which learns scaling vectors applied to the key and value matrices and intermediate activations in the feed-forward layer of the PLM.

Furthermore, we conduct experiments on several joint MTL baselines, namely **FINETUNE-M**, **ADAPTER-M**, and **PROPETL-M**, the fully fine-tuned, adapter-based, and ProPETL-based joint MTL variants, respectively; and, finally, **HYPERFORMER** and **HYPERFORMER++** (Karimi Mahabadi et al., 2021). FINETUNE-M updates all PLM parameters, ADAPTER-M adds a single adapter module shared for all tasks, and PROPETL-M combines sparse layer- and task-specific masks through a logical OR operation. Based on task-specific embeddings, HYPERFORMER and HYPER-FORMER++ generate module parameters by a shared hypernetwork. In all adapter-based models, we use a reduction factor of 16, and, following Pfeiffer et al. (2021), insert the modules after the feed-forward layer of the PLM. Furthermore, to allow a fair comparison, we adapt PROPETL-M, HYPERFORMER, and HYPERFORMER++ to this setting by inserting the adapter modules only after the feed-forward block of the PLM. To accommodate possible variations in performance, we train each model on multiple seeds, and report the mean and standard deviation over multiple runs.

The full details of the experiment setup regarding the benchmarks and their splits, infrastructure, training, and hyperparameters are provided in Appendix A.1. To further enable the reproducibility of our results, our code, including documentation, is available at *URL upon deanonymization*.

## 6 RESULTS

### 6.1 PARAMETER-EFFICIENCY ANALYSIS

Table 1 provides a comprehensive overview of the number of learnable parameters of the models in our experiment setting on GLUE and SuperGLUE: RoBERTa$_{BASE}$ as the backbone PLM, 8 source tasks, and the same 8 tasks as target tasks ($|S|=|T|=8$). Starting from the STL models, the first and middle columns report the number of trainable parameters for one and all tasks, respectively. The joint MTL models learn all tasks simultaneously, and hence only contain values in the middle column. For the two-stage MTL models, we report the number of trainable parameters of the

**Table 2:** Evaluation results on GLUE using RoBERTa$_{\text{BASE}}$. (Top) STL models, only learning a single task at a time. (Middle) Joint MTL methods, learning all tasks simultaneously. (Bottom) Two-stage MTL methods, composing the knowledge of several source adapters. The overall best results are underlined, and the best results among the two-stage MTL models are shown in **bold**.

| Model | MNLI | QQP | QNLI | SST-2 | STS-B | MRPC | RTE | CoLA | Avg. |
|---|---|---|---|---|---|---|---|---|---|
| FINETUNE | $86.61_{0.51}$ | $90.32_{0.15}$ | $91.78_{0.28}$ | $93.33_{0.48}$ | $90.53_{0.22}$ | $86.94_{1.52}$ | $73.47_{2.05}$ | $58.46_{4.03}$ | $83.93_{0.60}$ |
| ADAPTER | $86.50_{0.33}$ | $90.18_{0.11}$ | $92.25_{0.19}$ | $93.65_{0.71}$ | $90.23_{0.41}$ | $86.64_{1.07}$ | $72.89_{2.54}$ | $58.28_{2.50}$ | $83.83_{0.48}$ |
| PROPETL | $86.19_{0.25}$ | $88.88_{0.48}$ | $92.05_{0.80}$ | $93.81_{0.72}$ | $90.03_{0.35}$ | $85.93_{1.22}$ | $74.19_{2.03}$ | $59.29_{2.07}$ | $83.80_{0.42}$ |
| COMPACTER++ | $85.62_{0.42}$ | $88.84_{0.70}$ | $91.79_{0.39}$ | $93.58_{0.34}$ | $89.67_{0.54}$ | $87.21_{0.61}$ | $72.02_{2.21}$ | $58.49_{2.58}$ | $83.40_{0.45}$ |
| $(IA)^3$ | $83.78_{0.88}$ | $88.37_{0.20}$ | $90.57_{0.38}$ | $93.35_{0.30}$ | $89.93_{0.30}$ | $87.11_{1.14}$ | $72.56_{2.23}$ | $56.57_{5.39}$ | $82.78_{1.36}$ |
| FINETUNE-M | $84.95_{0.36}$ | $89.76_{0.12}$ | $90.91_{0.07}$ | $92.58_{0.76}$ | $86.14_{0.53}$ | $83.42_{0.50}$ | $80.99_{2.54}$ | $49.12_{1.74}$ | $82.23_{0.41}$ |
| ADAPTER-M | $86.03_{0.18}$ | $89.69_{0.01}$ | $91.58_{0.30}$ | $93.35_{0.41}$ | $88.71_{0.49}$ | $86.76_{0.92}$ | $80.26_{1.96}$ | $51.79_{1.23}$ | $83.52_{0.32}$ |
| PROPETL-M | $85.23_{0.45}$ | $87.82_{0.16}$ | $91.37_{0.52}$ | $93.88_{0.44}$ | $90.27_{0.22}$ | $86.36_{1.82}$ | $78.58_{0.90}$ | $54.71_{1.12}$ | $83.53_{0.31}$ |
| HYPERFORMER | $86.08_{0.46}$ | $89.13_{0.23}$ | $91.81_{0.07}$ | $93.16_{0.99}$ | $90.63_{0.32}$ | $87.01_{0.87}$ | $82.79_{1.68}$ | $57.30_{2.21}$ | $84.74_{0.39}$ |
| HYPERFORMER++ | $86.38_{0.18}$ | $88.81_{0.29}$ | $91.99_{0.17}$ | $93.27_{0.11}$ | $90.80_{0.12}$ | $87.83_{1.42}$ | $83.75_{0.78}$ | $54.05_{3.30}$ | $84.61_{0.46}$ |
| ADAPTERFUSION | $86.82_{0.04}$ | $90.23_{0.01}$ | $92.48_{0.15}$ | $93.23_{0.95}$ | $90.37_{0.20}$ | $\mathbf{88.41_{0.49}}$ | $79.49_{2.21}$ | $59.04_{1.69}$ | $85.01_{0.37}$ |
| ADAPTERSOUP | $63.47_{0.37}$ | $81.63_{0.23}$ | $78.00_{0.20}$ | $90.75_{0.24}$ | $80.17_{0.18}$ | $75.00_{1.18}$ | $62.09_{0.64}$ | $41.06_{1.68}$ | $71.52_{0.59}$ |
| SCALEARN | $86.97_{0.09}$ | $90.32_{0.10}$ | $92.51_{0.17}$ | $93.88_{0.18}$ | $\mathbf{90.96_{0.16}}$ | $87.75_{0.58}$ | $\mathbf{82.06_{1.37}}$ | $58.47_{1.76}$ | $\underline{\mathbf{85.36_{0.55}}}$ |
| SCALEARNUNIFORM | $86.93_{0.10}$ | $\mathbf{90.38_{0.11}}$ | $\mathbf{92.53_{0.28}}$ | $93.58_{0.20}$ | $90.08_{0.07}$ | $87.57_{0.86}$ | $80.07_{1.18}$ | $59.04_{1.05}$ | $85.02_{0.49}$ |
| SCALEARN++ | $\mathbf{87.06_{0.03}}$ | $90.04_{0.12}$ | $92.03_{1.10}$ | $\mathbf{94.15_{0.30}}$ | $90.62_{0.13}$ | $88.21_{0.63}$ | $80.87_{1.05}$ | $\underline{\mathbf{59.82_{0.78}}}$ | $85.35_{0.52}$ |
| SCALEARNUNIFORM++ | $86.98_{0.17}$ | $\mathbf{90.38_{0.01}}$ | $\mathbf{92.53_{0.28}}$ | $94.11_{0.07}$ | $90.18_{0.19}$ | $87.43_{0.63}$ | $80.04_{0.99}$ | $59.45_{0.67}$ | $85.14_{0.38}$ |

**Table 3:** Evaluation results on SuperGLUE using RoBERTa$_{\text{BASE}}$.

| Model | ReCoRD | MultiRC | BoolQ | WiC | WSC | COPA | CB | RTE | Avg. |
|---|---|---|---|---|---|---|---|---|---|
| FINETUNE | $71.61_{0.84}$ | $71.64_{1.15}$ | $76.80_{1.34}$ | $66.38_{2.08}$ | $63.46_{0.00}$ | $68.60_{6.74}$ | $81.96_{4.33}$ | $73.47_{2.05}$ | $71.74_{2.32}$ |
| ADAPTER | $79.02_{0.62}$ | $72.84_{0.48}$ | $76.71_{1.38}$ | $65.58_{1.56}$ | $63.46_{0.00}$ | $70.20_{4.13}$ | $84.82_{3.18}$ | $72.89_{2.54}$ | $73.19_{1.74}$ |
| PROPETL | $80.29_{0.24}$ | $73.07_{0.49}$ | $76.58_{0.78}$ | $66.60_{1.65}$ | $63.46_{0.00}$ | $70.60_{3.86}$ | $84.46_{3.86}$ | $74.19_{2.03}$ | $73.69_{1.53}$ |
| COMPACTER++ | $77.69_{2.67}$ | $70.44_{0.57}$ | $75.88_{0.96}$ | $66.46_{1.63}$ | $63.46_{0.00}$ | $68.30_{4.00}$ | $87.68_{3.62}$ | $72.02_{2.21}$ | $72.74_{1.96}$ |
| $(IA)^3$ | $75.27_{0.23}$ | $70.32_{0.49}$ | $76.31_{0.79}$ | $67.07_{1.68}$ | $63.35_{0.32}$ | $69.30_{3.37}$ | $87.32_{4.57}$ | $72.56_{2.23}$ | $72.69_{1.71}$ |
| FINETUNE-M | $72.21_{0.28}$ | $72.11_{0.68}$ | $76.39_{3.07}$ | $52.19_{1.11}$ | $63.46_{0.00}$ | $74.33_{3.40}$ | $84.52_{0.84}$ | $74.85_{7.42}$ | $71.26_{2.10}$ |
| ADAPTER-M | $72.43_{0.64}$ | $72.46_{0.43}$ | $75.32_{2.78}$ | $51.99_{1.74}$ | $59.94_{2.97}$ | $71.67_{3.40}$ | $86.31_{1.68}$ | $76.53_{1.06}$ | $70.83_{1.84}$ |
| PROPETL-M | $73.14_{0.19}$ | $72.07_{0.58}$ | $73.91_{3.27}$ | $50.73_{0.99}$ | $59.62_{5.44}$ | $74.00_{3.27}$ | $82.14_{1.46}$ | $73.65_{4.83}$ | $69.91_{2.38}$ |
| HYPERFORMER | $65.93_{4.47}$ | $33.54_{33.54}$ | $74.01_{1.10}$ | $55.49_{1.72}$ | $52.88_{10.58}$ | $55.50_{2.50}$ | $71.43_{7.14}$ | $61.73_{9.03}$ | $58.81_{8.76}$ |
| HYPERFORMER++ | $24.50_{8.13}$ | $19.47_{27.53}$ | $62.17_{0.00}$ | $50.00_{0.00}$ | $63.46_{0.00}$ | $54.33_{3.30}$ | $49.40_{0.84}$ | $49.09_{2.56}$ | $46.55_{5.30}$ |
| ADAPTERFUSION | $78.82_{0.49}$ | $71.79_{1.67}$ | $76.72_{0.55}$ | $66.57_{1.24}$ | $\mathbf{63.46_{0.00}}$ | $73.10_{4.51}$ | $82.32_{2.85}$ | $76.03_{2.38}$ | $73.60_{1.71}$ |
| ADAPTERSOUP | $64.26_{0.13}$ | $33.62_{4.28}$ | $68.84_{0.31}$ | $58.53_{0.60}$ | $\mathbf{63.46_{0.00}}$ | $52.40_{2.41}$ | $70.89_{0.86}$ | $57.83_{0.93}$ | $58.73_{1.19}$ |
| SCALEARN | $79.52_{0.06}$ | $\mathbf{73.22_{0.44}}$ | $\mathbf{77.27_{0.68}}$ | $66.35_{1.20}$ | $\mathbf{63.46_{0.00}}$ | $74.80_{2.15}$ | $90.89_{2.59}$ | $78.88_{2.14}$ | $75.55_{1.16}$ |
| SCALEARNUNIFORM | $\mathbf{80.13_{0.38}}$ | $71.91_{0.60}$ | $76.06_{0.41}$ | $67.37_{1.22}$ | $62.50_{1.27}$ | $71.20_{1.23}$ | $89.11_{1.97}$ | $75.31_{0.90}$ | $74.20_{1.00}$ |
| SCALEARN++ | $\mathbf{80.13_{0.09}}$ | $72.71_{0.57}$ | $76.44_{0.53}$ | $67.13_{1.24}$ | $62.26_{2.28}$ | $\mathbf{75.20_{1.93}}$ | $\mathbf{93.04_{2.14}}$ | $\mathbf{79.03_{0.95}}$ | $\underline{\mathbf{75.74_{1.22}}}$ |
| SCALEARNUNIFORM++ | $79.79_{0.14}$ | $71.75_{0.38}$ | $76.13_{0.52}$ | $\mathbf{67.87_{0.89}}$ | $\mathbf{63.46_{0.00}}$ | $74.00_{1.70}$ | $91.61_{2.53}$ | $74.84_{1.58}$ | $74.93_{0.97}$ |

transfer layer for one target task ($\Omega_t$) in the first column, the same for all target tasks in the middle ($\Omega$), and the sum of the number of transfer ($\Omega$) and source adapter parameters ($\Theta$) in the last column. We deliberately organize the transfer parameters of the two-stage models ($\Omega$) under the corresponding numbers of other models in the middle column since the two-stage paradigm benefits from already trained adapters and only needs to learn the transfer layer. The last column is provided for completeness in the case that the adapters should also be trained.

Comparing the results of the two-stage MTL methods in the transfer layer, ADAPTERFUSION is expectedly far less parameter-efficient than SCALEARN models, where SCALEARNUNIFORM++ only requires 64 parameters. The variants of SCALEARN add considerably fewer transfer parameters compared to the overall parameters of the particularly efficient joint MTL methods. Moreover, the SCALEARN models still remain comparable when also taking into account the source adapter parameters. Considering these results, in the following we report and discuss the evaluation results in transfer learning and few-shot learning on the respective benchmarks.

## 6.2 TRANSFER LEARNING PERFORMANCE

**Results on GLUE.** Table 2 shows the evaluation results on the GLUE benchmark using RoBERTa$_{\text{BASE}}$. The evaluation metrics are Pearson's correlation for STS-B, Matthews' correlation for CoLA, and accuracy for the rest. We average the results over several runs and report the corresponding standard deviation in the subscripts. Overall, the two-stage models obtain strong gains, outperforming STL and joint MTL models. Remarkably, all variants of SCALEARN, including the highly parameter-efficient SCALEARNUNIFORM++ achieve similarly good results with only a fraction of the parameters of ADAPTERFUSION. Comparing the different variations of our method, while SCALEARN shows the best results, the other models also perform highly competitively.

**Results on SuperGLUE.** Table 3 shows the results on SuperGLUE for all methods considered. The evaluation metrics are F1 for MultiRC and ReCoRD and accuracy for other tasks. We observe similar patterns on this benchmark: two-stage models generally outperform other baselines. In

**Table 4:** Evaluation results on HumSet using XLM-R$_{\text{BASE}}$.

| Model | Sectors | Pillars 1D | Subpillars 1D | Pillars 2D | Subpillars 2D | Avg. |
|---|---|---|---|---|---|---|
| FINETUNE | $71.99_{0.32}$ | $50.40_{0.24}$ | $43.76_{0.67}$ | $61.04_{0.26}$ | $41.68_{0.62}$ | $53.77_{0.42}$ |
| ADAPTER | $71.38_{0.28}$ | $51.02_{1.23}$ | $43.26_{0.82}$ | $61.43_{0.91}$ | $42.46_{0.51}$ | $53.91_{0.75}$ |
| PROPETL | $71.69_{0.86}$ | $49.69_{1.30}$ | $41.63_{0.84}$ | $60.58_{0.91}$ | $39.85_{1.10}$ | $52.69_{1.00}$ |
| COMPACTER++ | $69.97_{1.89}$ | $37.37_{7.99}$ | $37.76_{2.14}$ | $58.13_{1.64}$ | $33.10_{9.00}$ | $47.26_{4.53}$ |
| $(IA)^3$ | $70.22_{0.97}$ | $45.55_{3.43}$ | $40.05_{3.15}$ | $58.54_{1.38}$ | $39.27_{1.01}$ | $50.73_{1.99}$ |
| FINETUNE-m | $51.75_{3.62}$ | $22.65_{12.88}$ | $13.54_{6.06}$ | $33.27_{21.23}$ | $12.42_{3.39}$ | $26.73_{9.44}$ |
| ADAPTER-m | $56.20_{2.72}$ | $28.53_{14.56}$ | $16.53_{9.46}$ | $35.90_{17.36}$ | $18.89_{2.64}$ | $31.21_{9.35}$ |
| PROPETL-m | $59.80_{10.09}$ | $26.10_{14.36}$ | $29.57_{7.40}$ | $37.53_{12.08}$ | $30.35_{5.91}$ | $36.67_{9.97}$ |
| HYPERFORMER | $71.08_{1.04}$ | $40.65_{6.93}$ | $34.16_{3.37}$ | $46.22_{14.11}$ | $32.47_{4.46}$ | $44.92_{5.98}$ |
| HYPERFORMER++ | $60.42_{9.79}$ | $22.07_{7.45}$ | $20.35_{7.04}$ | $30.55_{19.83}$ | $18.90_{10.84}$ | $30.46_{10.99}$ |
| ADAPTERFUSION | $72.05_{0.12}$ | $49.63_{0.53}$ | $43.15_{0.38}$ | $60.68_{0.23}$ | $42.14_{0.46}$ | $53.53_{0.35}$ |
| ADAPTERSOUP | $56.81_{1.90}$ | $30.09_{0.40}$ | $21.84_{0.55}$ | $40.71_{0.98}$ | $17.89_{2.02}$ | $33.47_{1.17}$ |
| SCALEARN | $72.36_{0.05}$ | $51.63_{0.61}$ | $44.06_{0.37}$ | $61.52_{0.11}$ | $\underline{\mathbf{42.81}}_{0.63}$ | $\underline{\mathbf{54.48}}_{0.35}$ |
| SCALEARNUNIFORM | $72.20_{0.14}$ | $50.08_{0.79}$ | $42.97_{0.70}$ | $60.62_{0.16}$ | $41.95_{0.60}$ | $53.56_{0.48}$ |
| SCALEARN++ | $\underline{\mathbf{72.38}}_{0.27}$ | $\underline{\mathbf{51.66}}_{0.27}$ | $\underline{\mathbf{44.23}}_{0.50}$ | $\underline{\mathbf{61.66}}_{0.13}$ | $42.21_{0.21}$ | $54.43_{0.28}$ |
| SCALEARNUNIFORM++ | $72.02_{0.32}$ | $50.78_{0.41}$ | $42.60_{0.85}$ | $60.82_{0.14}$ | $42.14_{0.72}$ | $53.67_{0.49}$ |

this benchmark, SCALEARN and SCALEARN++ improve upon ADAPTERFUSION by 2 percentage points of the average results. Notably, we observe performance drops for various joint MTL models in comparison to other models (up to $-27\%$ when comparing HYPERFORMER++ and ADAPTER). This may be a signal of the sensitivity of these models to the selection of tasks. Furthermore, the subpar performance of AdapterSoup suggests that calculating weights using sentence similarity is not appropriate for our specific problem setup. In contrast, the other two-stage MTL models (and, in particular, our SCALEARN models) do not show any considerable performance decreases.

**Results on HumSet.** Table 4 shows the results on HumSet using XLM-R$_{\text{BASE}}$ with the F1-score as the evaluation metric. Similarly, SCALEARN performs the best among all the methods, whereas the more parameter-efficient variants of SCALEARN are only marginally weaker in performance. On this benchmark, in particular, all joint MTL methods show poor performance, highlighting the sensitivity of these methods to task selection (up to $-27\%$ for STL and MTL versions of FINETUNE).

We conduct an ablation study on the effect on different combinatorial operators in SCALEARN, reported in Appendix A.3. In Appendix A.4, we provide further experiments and analyses of the results along with the results of GLUE and SuperGLUE using RoBERTa$_{\text{LARGE}}$, HumSet using XLM-R$_{\text{LARGE}}$, and for the combination of all tasks from GLUE and SuperGLUE. Finally, we provide an analysis of the scaling coefficients of SCALEARNUNIFORM and SCALEARNUNIFORM++ in Appendix A.5, revealing the effect of various source adapters on a target task.

## 6.3 FEW-SHOT TRANSFER LEARNING

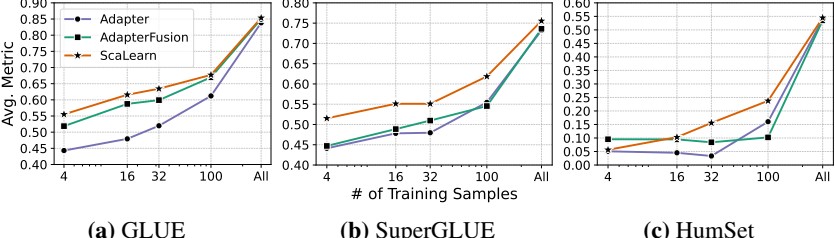

**(a)** GLUE      **(b)** SuperGLUE      **(c)** HumSet

**Figure 3:** Few-shot transfer learning results with $k = \{4,16,32,100\}$ training samples for each target task using the BASE models of RoBERTa and XLM-R. Full results over several runs are provided in Appendix A.6.

We further assess the applicability of SCALEARN in a few-shot setting, where we assume that only $k = \{4,16,32,100\}$ training samples are available for a given target task. For two-stage MTL methods, for a given benchmark, we use the source adapters of all tasks except the one corresponding to the target task, where we use a source adapter trained on only $k$ samples. On the basis of this set of source adapters, we then train a transfer layer on the target task using $k$ data points.

Table 3 shows the performance of ADAPTER, ADAPTERFUSION, and SCALEARN on the GLUE, SuperGLUE, and HumSet benchmarks, averaged over 5 runs. We observe that SCALEARN consistently outperforms ADAPTER and ADAPTERFUSION in all benchmarks and values of $k$ (except for $k = 4$ on HumSet) pointing to the strength of our method for data-lean settings. We provide the full results, including per-dataset ones, other variations of SCALEARN, and on RoBERTa$_{\text{LARGE}}$ in A.6.

## 7 RELATED WORK

**Parameter-efficient task learning in NLP.** Various parameter-efficient methods have emerged as a more sustainable alternative to full fine-tuning, enabling modularization, efficient sharing, and reusability of knowledge. A common modularization approach is to introduce a small number of additional parameters into a PLM, realized by various methods such as Adapters (Rebuffi et al., 2017; Houlsby et al., 2019), Compacter (Mahabadi et al., 2021a), and ProPETL-Adapter (Zeng et al., 2023). Similarly, LoRA (Hu et al., 2022) injects trainable low-rank matrices into each transformer layer, and BitFit (Ben Zaken et al., 2022) updates only the bias terms. Another line of research identifies sparse subnetworks within the model to tune (Ansell et al., 2022; Guo et al., 2021; Hauzenberger et al., 2023), while He et al. (2022) and Mao et al. (2022) propose to merge various distinct modules. We refer to Pfeiffer et al. (2023) for a full survey on this topic.

**Learning by scaling.** Besides the common approach of learning a feed-forward layer for a (non–) linear transformation of an input vector, several recent methods explore the merit of learning a scaling vector applied to the input vector in various scenarios. Liu et al. (2022) learn a modular network for STL that rescales PLM vectors through element-wise multiplication. Ilharco et al. (2023) and Ortiz-Jiménez et al. (2023) introduce task arithmetic to control PLM behavior by extracting task vectors from pre- and post-fine-tuning model weights, then scaling and combining them to improve MTL performance. Masoudian et al. (2023) learn a gating adapter that adjusts the scaling of representations to control the behavior of the model at inference time. Finally, Lian et al. (2022) learn to shift and scale the output vectors of a vision transformer in an STL setting. Our work contributes to this line of research by leveraging scaling for highly parameter-efficient and effective MTL.

**Joint MTL.** Interference and imbalance between tasks have been shown to impede performance in joint MTL (Kirkpatrick et al., 2017; Kendall et al., 2018; Pfeiffer et al., 2023). Several studies have aimed to address these issues and improve generalization. For example, Liu et al. (2019a) learn representations across multiple NLU tasks using context from a semantic similarity model, and Pilault et al. (2021) introduce a parameter-efficient model that uses modules facilitating weight sharing. Moreover, Stickland & Murray (2019) use an adapter for each task while also updating the PLM parameters. Zhang et al. (2022) further focus on modularity by only activating a subset of task-specific modules at once; however, tasks must be mapped a priori to a given high-level skill. Ponti et al. (2023) and Caccia et al. (2022) loosen this constraint by learning a task-skill allocation matrix for cross-task generalization, but rely on a multi-task pre-training stage. Finally, Mahabadi et al. (2021b) leverage a hypernetwork (Ha et al., 2017) that generates modular task-specific parameters.

**Two-stage MTL.** Various methods have been proposed to extract task-specific information and compose this knowledge. Chronopoulou et al. (2023) studies transfer learning in generative PLMs by first selecting source adapters based on different heuristics and merging their weights to create a new combined adapter. Huang et al. (2023) introduce LoraHub with the aim of composing LoRA (Hu et al., 2022) modules for cross-task generalization using black-box optimization and an additional pre-filtering stage. Asai et al. (2022) and Wang et al. (2023) leverage continuous prompts learned on large-scale source tasks, leading to competitive performance in MTL benchmarks, although both methods depend on the selection of typically high-resource source tasks. In contrast to the mentioned methods that highly depend on the selection of tasks and/or apply the combination to the weights, Pfeiffer et al. (2021) combines the output representations of several independent source adapters through an attention mechanism. Our work is directly related to this line of research and introduces a novel highly parameter-efficient transfer layer applied to the output representation.

## 8 CONCLUSION

We propose SCALEARN, a highly parameter-efficient and effective two-stage MTL method leveraging simple scaling of output vectors. Based on an initial analysis of the effect of scaling adapter output representations, our proposed approach directly learns the coefficients that scale the representations of source adapters and combines them by simply taking the sum. We conduct extensive transfer learning experiments using encoder PLMs on the three benchmarks of GLUE, SuperGLUE, and HumSet, consisting of a diverse set of tasks, domains, and languages. Our evaluation results show that SCALEARN and even its extremely parameter-efficient variants, such as SCALEARNUNI-FORM++, obtain strong improvement over existing MTL methods without any negative cross-task effects. We further show that these improvements are also present in few-shot transfer learning.

ETHICS STATEMENT

The nature of our work is manifold, and so are the ethical aspects touched by our research. First, we acknowledge the potential of NLP datasets and models for encoding unfair stereotypical (Blodgett et al., 2020) and exclusive (Dev et al., 2021) biases that may lead to representational and allocational harms (Barocas et al., 2017). This potential is a general property of PLMs, and the models and datasets we use in this research are no exception to this danger. We thus strongly advise practitioners to carefully consider the sociotechnical context before deploying any models (with or without SCALEARN), and, aligned with the specific deployment scenario, to take measures against unfair discrimination. Examples of such measures include the use of bias measurement (Nangia et al., 2020) and mitigation (Bordia & Bowman, 2019) approaches. Second, the core of this work deals with efficiency aspects. On the one hand, given the well-known relationship between model training (and inference) effort and potential $CO_2$ emissions (Strubell et al., 2019), our work directly contributes to reaching the goals of Green AI by making parameter-efficient MTL more environmentally sustainable. On the other hand, since PLM training often comes with high infrastructure requirements exclusive to certain user groups (Bender et al., 2021), we hope that our work also contributes to the ongoing democratization of language technology by reducing resource-related usage barriers.

REPRODUCIBILITY STATEMENT

For all our experiments, we use PLM configurations that are publicly available and can be downloaded from the Huggingface `transformers` library (Wolf et al., 2020). Sufficient details to reproduce our results, including hyperparameter settings and seeds used in training, and information about the datasets we use for training, including splits, can be found in Section 5 and in Appendix A.1. All datasets we use in our experiments are commonly used in the MTL literature and publicly available to ensure comparability and reproducibility. We also release our code under the MIT License, ensuring open access to the community for further development.

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

# A APPENDIX

## A.1 COMPLETE EXPERIMENT DETAILS

| Name | Category | Task | Domain | Metric |
|------|----------|------|--------|--------|
| MNLI | GLUE | NLI | various | accuracy |
| QQP | GLUE | paraphrase detection | social QA | accuracy & F1 |
| QNLI | GLUE | NLI | Wikipedia | accuracy |
| SST-2 | GLUE | sentiment analysis | Movie Reviews | accuracy |
| STS-B | GLUE | sentence similarity | various | Pearson & Spearman corr. |
| MRPC | GLUE | paraphrase detection | news | accuracy & F1 |
| RTE | GLUE | NLI | News, Wikipedia | accuracy |
| CoLA | GLUE | acceptability | various | Matthews' corr. |
| ReCoRD | SuperGLUE | cloze-style QA | news (CNN, Daily Mail) | F1 & EM |
| MultiRC | SuperGLUE | QA | various | F1 & EM |
| BoolQ | SuperGLUE | boolean QA | Wikipedia | accuracy |
| WiC | SuperGLUE | word sense disambiguation | lexical databases | accuracy |
| WSC | SuperGLUE | coreference / commonsense | fiction books | accuracy |
| COPA | SuperGLUE | commonsense reasoning | various | accuracy |
| CB | SuperGLUE | NLI | various | accuracy |
| Sectors | HumSet | classification | humanitarian crisis response | F1 & precision |
| Pillars 1D | HumSet | classification | humanitarian crisis response | F1 & precision |
| Subpillars 1D | HumSet | classification | humanitarian crisis response | F1 & precision |
| Pillars 2D | HumSet | classification | humanitarian crisis response | F1 & precision |
| Subpillars 2D | HumSet | classification | humanitarian crisis response | F1 & precision |

**Table 5:** Details of all datasets. Lexical databases for WiC include WordNet, VerbNet, Wiktionary. For datasets where two metrics are officially used, we use the underlined metric as our main metric. (Top) GLUE tasks. (Middle) SuperGLUE tasks. (Bottom) HumSet tasks.

**Dataset Details.** As has been mentioned, we are using the GLUE, SuperGLUE, and HumSet benchmarks for our experiments. Table 5 summarizes the tasks contained in each of the datasets. We use the `datasets` library (Lhoest et al., 2021) to load each dataset for our experiments. We set the maximum length of the input sequence to 128 tokens for all tasks in GLUE, SuperGLUE, and HumSet. However, for MultiRC and ReCoRD, we set the maximum length to 324 and 256, respectively, due to their significantly longer context lengths. Note that we treat HumSet as five separate tasks, following (Fekih et al., 2022). The GLUE and SuperGLUE benchmarks only contain the training and validation split publicly, so we follow Chen et al. (2022) and use 10% of the training samples from the training split as the validation set and the remaining 90% for training. We split the datasets with the `datasets` library (Lhoest et al., 2021) using seed 42 and shuffle the samples. Then, the original validation split is taken as the test set on which we report the performance of all models. For HumSet, we use the original train/validation/test splits, as all of them are publicly available, including labels. Details about the train/validation/test splits can be found in Table 6.

**Computing Infrastructure.** We run all experiments with RoBERTa$_{\text{BASE}}$ and XLM-R$_{\text{BASE}}$ on a single Nvidia GTX1080Ti GPU and Intel Xeon CPU E5-2640 v4 CPUs, and the experiments with RoBERTa$_{\text{LARGE}}$ and XLM-R$_{\text{LARGE}}$ on a single Nvidia RTX5000 GPU and Intel Xeon Silver 4216 CPUs.

**Implementation Details.** We use `PyTorch` (Paszke et al., 2019) for all experiments. For the joint multi-task learning methods, we adapt the codebase of Karimi Mahabadi et al. (2021) and Zeng et al. (2023), both of which rely on the `transformers` (Wolf et al., 2020) library. For all other models, we make use of the `adapter-transformers` library (Pfeiffer et al., 2020) library, a wrapper around the `transformers` library.

**Training and optimization.** We train all methods with a batch size of 32. All STL and two-stage MTL methods are trained for a maximum of 30 epochs with early stopping and patience of 5. [4] We use 10 seeds for low-resource and 3 seeds for high-resource tasks when using RoBERTa$_{\text{BASE}}$, and on 5 and 2 seeds for low- and high-resource tasks, respectively, when using RoBERTa$_{\text{LARGE}}$. We define tasks with more than $10k$ training samples as high-resource and as low-resource otherwise. All joint MTL models are trained on 3 seeds. We report the mean and standard deviations across all

---

[4]The exception is ReCoRD, which we train on 3 epochs due to its size.

| Name | \|Train\| | \|Validation\| | \|Test\| |
|---|---|---|---|
| MNLI | 353,431 | 39,270 | 9,815 |
| QQP | 327,461 | 36,384 | 40,430 |
| QNLI | 94,268 | 10,474 | 5,463 |
| SST-2 | 60,614 | 6,734 | 872 |
| STS-B | 5,174 | 574 | 1,500 |
| MRPC | 3,301 | 366 | 408 |
| RTE | 2,241 | 249 | 277 |
| CoLA | 7,695 | 855 | 1,043 |
| ReCoRD | 100,730 | 10,000 | 10,000 |
| MultiRC | 24,518 | 2,724 | 4,848 |
| BoolQ | 8,484 | 942 | 3,270 |
| WiC | 4,885 | 542 | 638 |
| WSC | 498 | 55 | 104 |
| COPA | 360 | 40 | 100 |
| CB | 225 | 25 | 56 |
| Sectors | 117,435 | 16,039 | 15,147 |
| Pillars 1D | 117,435 | 16,039 | 15,147 |
| Subpillars 1D | 117,435 | 16,039 | 15,147 |
| Pillars 2D | 117,435 | 16,039 | 15,147 |
| Subpillars 2D | 117,435 | 16,039 | 15,147 |

**Table 6:** Number of used samples for each dataset and used split. (Top) GLUE tasks. (Middle) SuperGLUE tasks. (Bottom) HumSet tasks.

runs. We use the AdamW (Kingma & Ba, 2015; Loshchilov & Hutter, 2019) optimizer with default PyTorch hyperparameters (weight decay $= 0.01$, $\beta_1 = 0.9$, $\beta_2 = 0.99$, $\epsilon = 1 \cdot 10^{-6}$). We use seeds $\{0,1\}$ for instances with two seeds, $\{0,1,2\}$ for instances with three seeds, seeds $\{0,1,2,3,4\}$ for instances with five seeds, and $\{0,1,2,3,4,5,6,7,8,9\}$ for instances with ten seeds.

**Single-task learning hyperparameters.** We train FINETUNE with a learning rate of 2e-5, ADAPTER with a learning rate of 3e-4, COMPACTER++ with a learning rate of 3e-3, and PROPETL with a learning rate of 1e-3, a mask learning rate of 5e-3, a sparsity rate of $0.5$, and a weight decay of $0.1$, which we found to be the most suitable for our setup. Moreover, we train $(IA)^3$ with a learning rate of 5e-3. Each of them is trained with a linear learning rate decay. For RoBERTa$_{\text{LARGE}}$, we add a linear learning rate warmup for the first 10% of training, as we notice it improves stability. For early stopping, we use the loss on the validation set, except for HumSet, where we use the F1-score, and in the few-shot setting, where we use the main metric for the respective dataset, as shown in Table 5. In the few-shot setting, we train for a maximum of 1,000 steps, apply an early stopping patience of 20, and use a maximum of 5,000 samples for validation. Note that, while the PLM layer normalization parameters have also been updated (Mahabadi et al., 2021a;b), following Pfeiffer et al. (2021), we keep them frozen. This approach improves modularity, while still allowing PLMs to efficiently adapt to new tasks. Note that the same hyperparameters as outlined here are also used for ADAPTER in our probing analyses (cf. Section 3).

**Joint MTL hyperparameters.** In all joint multi-task learning methods, we sample tasks with conventional temperature-based sampling with temperature $\tau = 10$, following Mahabadi et al. (2021b) and Zeng et al. (2023). Specifically, a task is sampled with probability $p_t^{1/\tau}$, where $p_t = \frac{N_t}{\sum_{i=1}^{\tau} N_t}$, $N_t$ the number of training samples of task $t$, and $\tau = 10$. Using this sampling strategy, we train each model for a total of 375,000 steps to ensure convergence and evaluate every 7,500 steps. We train each model with early stopping and patience of 10. In the end, the model checkpoint with the lowest average validation loss is loaded and evaluated on the test set. We train FINETUNE-M with a learning rate of 2e-5, ADAPTER-M, HYPERFORMER, and HYPERFORMER++ with a learning rate of 3e-4, and PROPETL-M with a learning rate of 3e-4 and a mask learning rate of 3e-3, a sparsity rate of 0.3, and no weight decay. We train each of them with a linear learning rate warmup for the first 10% of training, followed by a linear learning rate decay. For the remaining hyperparameters of PROPETL-M, HYPERFORMER, and HYPERFORMER++, we follow the respective original implementations, but always use a reduction factor of 16 for a fair comparison.

**Two-stage MTL hyperparameters**. We train each variant of SCALEARN * with a learning rate of 6e-3 and train ADAPTERFUSION with a learning rate of 5e-5, following Pfeiffer et al. (2021). Both SCALEARN * and ADAPTERFUSION are trained with a linear learning rate decay and no warmup. Early stopping is the same as in the single-task learning setting. We initialize the parameters of SCALEARN* with $\mathcal{N}\left(\frac{2}{T}, 0.001\right)$,[5] and apply a dropout rate of $0.3$ to increase robustness for SCALEARN and SCALEARN++. For AdapterSoup, we first calculate the cosine similarity of sentence embeddings for each task from the training set using the `sentence-transformers` (Reimers & Gurevych, 2019) library and the `all-mpnet-base-v2` model. In contrast to Chronopoulou et al. (2023), who only select 100 samples for each domain, we select 10000 samples for each task, as our sequences corresponding to tasks are meaningfully shorter than the sequences corresponding to domains. Using these similarities, we select the top 5 most similar tasks to the target task, normalize the similarity scores to obtain the weights, and perform weight-space averaging of the adapter parameters, following Chronopoulou et al. (2023). Note that we also include the corpus of the target task when calculating the similarities for weight-space averaging, and hence also the target adapter during weight-space averaging, and train a new task head on the target task to allow a more fair comparison to other two-stage MTL methods. We use a learning rate of 3e-4 when training the target task head with ADAPTERSOUP.

## A.2 ADDITIONAL PROBING ANALYSES

We show the single-task probing results using the remaining GLUE and SuperGLUE source tasks not shown in Section 3 in Figure 4. For the probing experiments when using two task adapters (the target task $t$ and MNLI), we show the remaining tasks from GLUE and SuperGLUE with fewer than 10k samples as target tasks in Figure 5.

---

[5]We also test out $\{\mathcal{N}\left(\frac{1}{T}, 0.001\right), \mathcal{N}\left(\frac{3}{T}, 0.001\right), \mathcal{N}\left(1, 0.001\right)\}$.

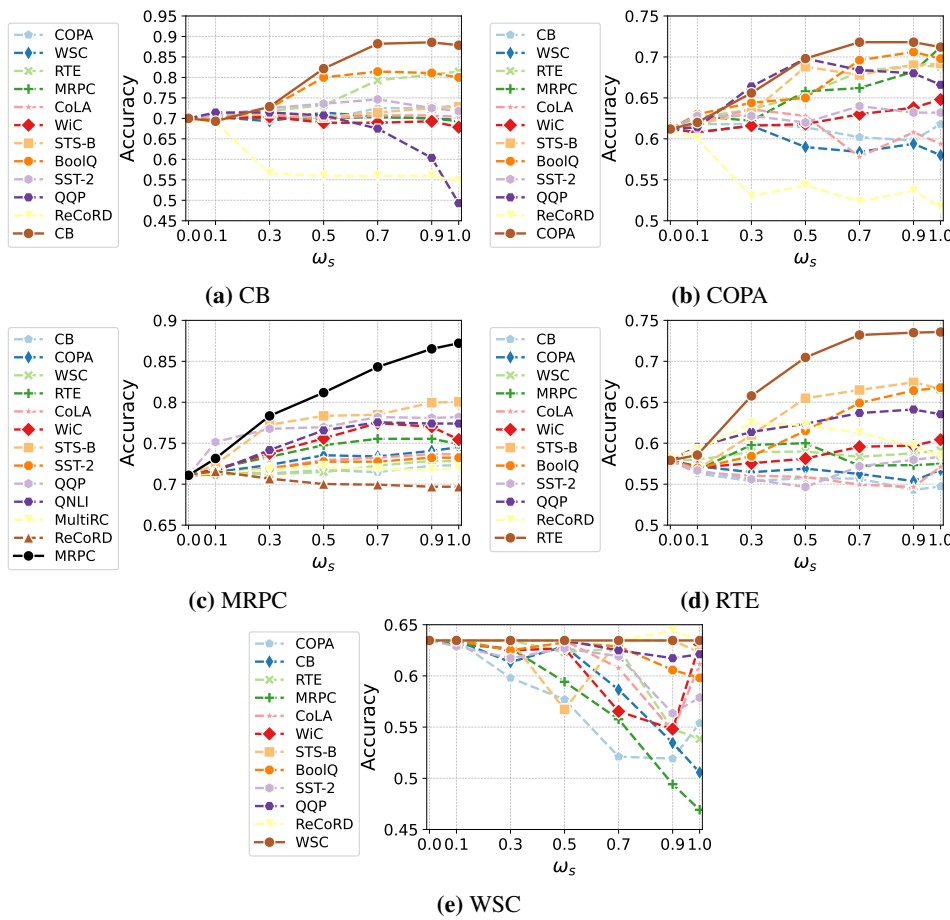

**Figure 4:** Effect of scaling the output representations $o_s^l$ of adapters by weight $\omega_s$ using different source adapters from all other tasks from GLUE and SuperGLUE. Each point shows the mean over 5 seeds.

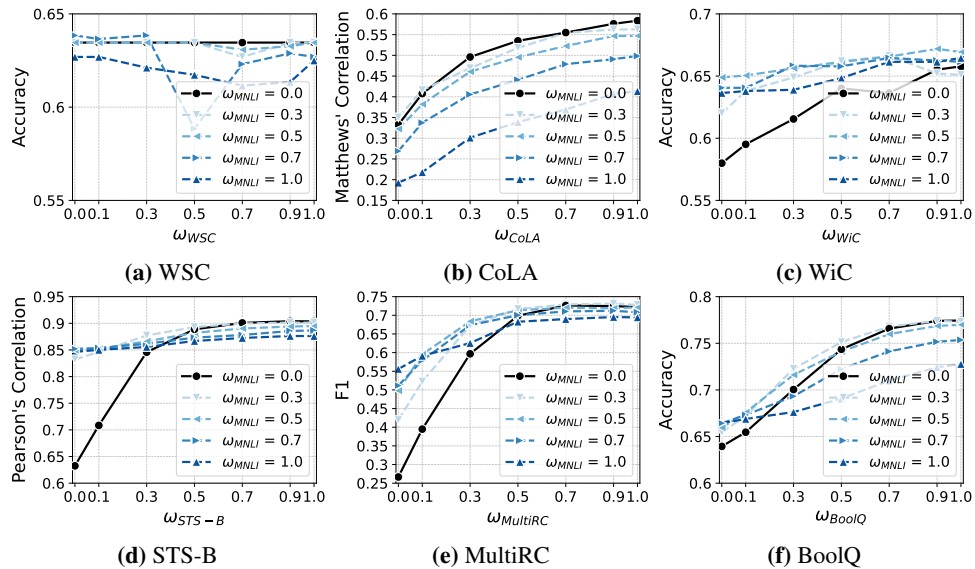

**Figure 5:** Effect of combining independently scaled output representations of two adapters trained on the target task and MNLI, respectively, on additional tasks from GLUE and SuperGLUE. Each point shows the mean over 5 seeds.

## A.3 ABLATION STUDY

Table 7 shows the effect of adding constraints on the distributional values of scaling coefficient in SCALEARN, evaluated on GLUE using RoBERTa$_{\text{BASE}}$. In particular, we change the original SCALEARN model by adding the constraints *mean* and *softmax* over the source task dimension, thus enforcing $\sum_{s=1}^{|S|} \omega_s^l = 1$. The results indicate that both constraints reduce average performance compared to those having no constraints, confirming our choice of directly learning the scaling coefficients without imposing any restrictions.

**Table 7:** Effect of adding various constraints to the scaling values of SCALEARN, evaluated on GLUE using RoBERTa$_{\text{BASE}}$. The constraints *mean* and *softmax* are applied over the task dimension, enforcing $\sum_{s=1}^{|S|} \omega_s^l = 1$. The best results are shown in **bold**.

| Model | Constraint | MNLI | QQP | QNLI | SST-2 | STS-B | MRPC | RTE | CoLA | Avg. |
|---|---|---|---|---|---|---|---|---|---|---|
| SCALEARN | None (original) | $86.97_{0.09}$ | $90.32_{0.10}$ | $\mathbf{92.51_{0.17}}$ | $93.88_{0.18}$ | $\mathbf{90.96_{0.16}}$ | $\mathbf{87.75_{0.58}}$ | $\mathbf{82.06_{1.37}}$ | $58.47_{1.76}$ | $\mathbf{85.36_{0.55}}$ |
| SCALEARN | Mean | $\mathbf{87.03_{0.01}}$ | $90.36_{0.30}$ | $92.34_{0.09}$ | $92.60_{1.38}$ | $90.62_{0.25}$ | $87.11_{0.79}$ | $79.21_{1.82}$ | $\mathbf{59.87_{2.95}}$ | $84.89_{0.95}$ |
| SCALEARN | Softmax | $86.85_{0.05}$ | $\mathbf{90.60_{0.05}}$ | $92.74_{0.22}$ | $93.75_{0.08}$ | $90.66_{0.10}$ | $85.83_{1.09}$ | $79.28_{1.04}$ | $58.43_{1.98}$ | $84.77_{0.58}$ |

## A.4 ADDITIONAL RESULTS

**More results using RoBERTa$_{\text{BASE}}$.** Table 11 shows the results when training on the combination of all GLUE and SuperGLUE tasks, resulting in a total of 15 tasks.

**Results using RoBERTa$_{\text{LARGE}}$.** We further validate our method and its variations on the encoder-based PLM RoBERTa$_{\text{LARGE}}$. Table 8 shows the corresponding results, including all baselines, on the GLUE benchmark. Table 9 shows the results on SuperGLUE. Table 10 shows the results on HumSet. Finally, Table 12 shows the results when training on the combination of all GLUE and SuperGLUE tasks, resulting in a total of 15 tasks.

**Table 8:** Evaluation results on GLUE using RoBERTa$_{\text{LARGE}}$. (Top) STL models, only learning a single task at a time. (Middle) Joint MTL methods, learning all tasks simultaneously. (Bottom) Two-stage MTL methods, composing the knowledge of several source adapters. The overall best results are underlined, and the best results among the two-stage MTL models are shown in **bold**.

| Model | MNLI | QQP | QNLI | SST-2 | STS-B | MRPC | RTE | CoLA | Avg. |
|---|---|---|---|---|---|---|---|---|---|
| FINETUNE | $89.57_{0.36}$ | $89.75_{1.03}$ | $93.91_{0.43}$ | $95.30_{0.65}$ | $91.89_{0.35}$ | $86.27_{1.15}$ | $81.52_{3.19}$ | $60.15_{2.89}$ | $86.04_{1.26}$ |
| ADAPTER | $89.62_{0.18}$ | $89.87_{0.67}$ | $94.13_{0.06}$ | $95.24_{0.08}$ | $91.81_{0.29}$ | $87.82_{2.11}$ | $81.23_{2.92}$ | $64.07_{1.97}$ | $86.72_{1.04}$ |
| PROPETL | $89.78_{0.24}$ | $89.23_{0.77}$ | $\underline{94.32}_{0.09}$ | $95.41_{0.00}$ | $91.45_{0.39}$ | $87.65_{0.73}$ | $84.55_{2.14}$ | $65.85_{2.10}$ | $87.28_{0.81}$ |
| COMPACTER++ | $89.15_{0.67}$ | $87.33_{2.39}$ | $92.93_{1.42}$ | $95.41_{0.00}$ | $91.46_{0.35}$ | $87.84_{1.23}$ | $79.71_{4.58}$ | $65.66_{2.08}$ | $86.19_{1.59}$ |
| $(IA)^3$ | $88.69_{0.61}$ | $87.79_{0.72}$ | $91.72_{0.79}$ | $94.95_{0.16}$ | $91.39_{0.45}$ | $86.37_{1.65}$ | $80.79_{3.16}$ | $64.70_{3.20}$ | $85.80_{1.34}$ |
| FINETUNE-M | $87.95_{0.39}$ | $89.82_{0.77}$ | $92.58_{0.32}$ | $94.88_{0.94}$ | $87.04_{0.68}$ | $81.37_{1.00}$ | $84.36_{1.19}$ | $55.32_{0.78}$ | $84.16_{0.76}$ |
| ADAPTER-M | $89.10_{0.36}$ | $89.35_{0.09}$ | $93.64_{0.05}$ | $94.90_{0.17}$ | $88.40_{0.32}$ | $83.09_{0.25}$ | $86.64_{0.00}$ | $56.38_{0.79}$ | $85.19_{0.25}$ |
| PROPETL-M | $88.98_{0.33}$ | $89.03_{0.15}$ | $94.14_{0.11}$ | $95.15_{0.05}$ | $91.56_{0.23}$ | $87.83_{1.10}$ | $\underline{88.45}_{0.29}$ | $60.99_{1.03}$ | $87.01_{0.41}$ |
| HYPERFORMER | $89.66_{0.40}$ | $90.15_{0.63}$ | $93.95_{0.13}$ | $95.80_{0.62}$ | $91.68_{0.35}$ | $86.60_{1.22}$ | $86.28_{0.29}$ | $61.18_{4.76}$ | $86.91_{1.05}$ |
| HYPERFORMER++ | $89.79_{0.21}$ | $89.54_{0.43}$ | $93.95_{0.54}$ | $95.22_{0.11}$ | $91.62_{0.29}$ | $88.07_{1.86}$ | $86.28_{1.06}$ | $65.16_{0.61}$ | $87.45_{0.64}$ |
| ADAPTERFUSION | $89.57_{0.17}$ | $\mathbf{90.88}_{0.06}$ | $94.15_{0.04}$ | $95.87_{0.00}$ | $91.86_{0.15}$ | $88.97_{0.78}$ | $85.70_{1.13}$ | $66.39_{1.83}$ | $87.93_{0.52}$ |
| ADAPTERSOUP | $65.83_{0.51}$ | $82.37_{0.00}$ | $74.06_{1.01}$ | $93.98_{0.24}$ | $81.67_{1.63}$ | $73.37_{0.51}$ | $67.27_{1.63}$ | $43.70_{1.62}$ | $72.78_{0.89}$ |
| SCALEARN | $90.09_{0.09}$ | $90.51_{0.26}$ | $94.18_{0.03}$ | $95.41_{0.16}$ | $92.32_{0.15}$ | $88.09_{0.82}$ | $\mathbf{87.08}_{0.54}$ | $65.40_{2.62}$ | $87.91_{0.55}$ |
| SCALEARNUNIFORM | $90.11_{0.04}$ | $90.05_{0.28}$ | $\mathbf{94.23}_{0.08}$ | $95.41_{0.16}$ | $92.11_{0.06}$ | $88.63_{1.72}$ | $84.40_{3.93}$ | $66.98_{0.58}$ | $87.74_{0.86}$ |
| SCALEARN++ | $\underline{90.31}_{0.10}$ | $90.59_{0.03}$ | $94.05_{0.03}$ | $\mathbf{95.93}_{0.24}$ | $\mathbf{92.48}_{0.15}$ | $88.48_{1.26}$ | $86.28_{1.05}$ | $\underline{67.13}_{0.59}$ | $\underline{88.16}_{0.43}$ |
| SCALEARNUNIFORM++ | $90.08_{0.01}$ | $90.49_{0.02}$ | $94.12_{0.16}$ | $95.18_{0.16}$ | $92.12_{0.09}$ | $\mathbf{90.05}_{0.54}$ | $84.98_{1.32}$ | $64.97_{0.85}$ | $87.75_{0.39}$ |

**Table 9:** Evaluation results on SuperGLUE using RoBERTa$_{\text{LARGE}}$.

| Model | ReCoRD | MultiRC | BoolQ | WiC | WSC | COPA | CB | RTE | Avg. |
|---|---|---|---|---|---|---|---|---|---|
| FINETUNE | $81.60_{1.25}$ | $79.03_{0.02}$ | $81.65_{0.30}$ | $69.72_{2.16}$ | $\underline{63.46}_{0.00}$ | $52.00_{8.28}$ | $90.36_{2.99}$ | $81.52_{3.19}$ | $74.92_{2.27}$ |
| ADAPTER | $88.52_{0.09}$ | $80.73_{0.69}$ | $82.36_{0.72}$ | $69.16_{1.31}$ | $63.25_{0.64}$ | $71.90_{13.63}$ | $92.68_{1.78}$ | $81.23_{2.92}$ | $78.73_{2.72}$ |
| PROPETL | $87.86_{2.59}$ | $\underline{81.19}_{0.99}$ | $81.61_{0.86}$ | $69.62_{2.16}$ | $\underline{63.46}_{0.00}$ | $69.00_{18.96}$ | $94.11_{4.04}$ | $84.55_{2.14}$ | $78.92_{3.97}$ |
| COMPACTER++ | $88.34_{0.97}$ | $79.18_{0.29}$ | $79.53_{6.13}$ | $69.26_{1.51}$ | $62.26_{1.43}$ | $79.00_{9.74}$ | $87.50_{7.48}$ | $79.71_{4.58}$ | $78.10_{4.02}$ |
| $(IA)^3$ | $87.47_{0.21}$ | $77.91_{0.43}$ | $80.97_{0.75}$ | $68.65_{2.55}$ | $60.58_{0.00}$ | $77.00_{0.00}$ | $90.00_{3.91}$ | $80.79_{3.16}$ | $77.93_{1.35}$ |
| FINETUNE-M | $83.57_{0.81}$ | $78.08_{0.55}$ | $81.70_{0.65}$ | $53.03_{0.37}$ | $49.36_{9.50}$ | $86.67_{2.36}$ | $82.14_{2.92}$ | $83.87_{2.01}$ | $74.80_{2.39}$ |
| ADAPTER-M | $86.76_{0.32}$ | $75.15_{0.24}$ | $77.18_{2.22}$ | $51.57_{1.12}$ | $53.21_{9.75}$ | $67.67_{1.25}$ | $80.95_{1.68}$ | $77.38_{1.36}$ | $71.23_{2.24}$ |
| PROPETL-M | $84.83_{0.40}$ | $79.60_{0.37}$ | $82.02_{1.11}$ | $55.33_{0.46}$ | $59.62_{9.05}$ | $86.67_{4.03}$ | $88.10_{2.23}$ | $85.56_{0.29}$ | $77.71_{2.24}$ |
| HYPERFORMER | $84.38_{1.00}$ | $79.68_{0.97}$ | $81.87_{0.97}$ | $53.81_{2.48}$ | $\underline{63.46}_{8.64}$ | $82.33_{6.94}$ | $83.93_{2.53}$ | $\underline{86.88}_{0.90}$ | $77.04_{3.05}$ |
| HYPERFORMER++ | $13.66_{0.00}$ | $40.21_{40.21}$ | $71.50_{9.33}$ | $49.14_{0.86}$ | $62.98_{0.48}$ | $54.00_{3.00}$ | $67.86_{17.86}$ | $66.97_{19.68}$ | $53.29_{11.43}$ |
| ADAPTERFUSION | $\underline{89.21}_{0.17}$ | $80.52_{0.24}$ | $82.21_{0.30}$ | $69.09_{1.68}$ | $\underline{63.46}_{0.68}$ | $81.20_{16.07}$ | $\mathbf{95.71}_{0.98}$ | $86.06_{1.07}$ | $80.93_{2.65}$ |
| ADAPTERSOUP | $70.33_{0.28}$ | $38.42_{12.42}$ | $73.20_{0.16}$ | $62.23_{1.17}$ | $\underline{63.46}_{0.00}$ | $54.50_{5.74}$ | $68.75_{1.03}$ | $61.37_{3.97}$ | $61.53_{3.06}$ |
| SCALEARN | $87.85_{0.01}$ | $78.40_{0.70}$ | $80.29_{2.52}$ | $68.56_{1.68}$ | $62.98_{0.68}$ | $85.40_{3.78}$ | $92.86_{1.79}$ | $84.91_{0.59}$ | $80.16_{1.47}$ |
| SCALEARNUNIFORM | $88.85_{0.22}$ | $80.42_{0.06}$ | $81.85_{0.21}$ | $69.91_{1.15}$ | $61.54_{0.00}$ | $82.00_{3.08}$ | $90.00_{1.60}$ | $84.04_{1.66}$ | $79.83_{1.00}$ |
| SCALEARN++ | $88.28_{0.23}$ | $\mathbf{80.76}_{0.58}$ | $\mathbf{83.08}_{0.31}$ | $69.59_{1.89}$ | $62.98_{0.68}$ | $\mathbf{87.80}_{1.10}$ | $91.07_{1.79}$ | $85.70_{0.32}$ | $\underline{81.16}_{0.86}$ |
| SCALEARNUNIFORM++ | $88.85_{0.22}$ | $80.70_{0.04}$ | $82.13_{0.21}$ | $\mathbf{70.19}_{0.26}$ | $62.98_{0.68}$ | $83.60_{2.88}$ | $91.07_{2.82}$ | $84.84_{1.02}$ | $80.54_{1.02}$ |

**Table 10:** Evluation results on HumSet using XLM-R$_{\text{LARGE}}$.

| Model | Sectors | Pillars 1D | Subpillars 1D | Pillars 2D | Subpillars 2D | Avg. |
|---|---|---|---|---|---|---|
| FINETUNE | $72.99_{0.17}$ | $51.38_{0.39}$ | $44.84_{0.89}$ | $61.90_{0.20}$ | $43.49_{0.86}$ | $54.92_{0.50}$ |
| ADAPTER | $72.29_{0.59}$ | $49.31_{1.27}$ | $\underline{45.25}_{0.03}$ | $62.58_{0.67}$ | $44.36_{0.66}$ | $54.76_{0.65}$ |
| PROPETL | $73.20_{0.32}$ | $51.58_{0.40}$ | $45.10_{0.92}$ | $61.52_{2.29}$ | $41.98_{0.70}$ | $54.68_{0.92}$ |
| COMPACTER++ | $61.77_{12.63}$ | $8.17_{5.92}$ | $6.37_{11.00}$ | $20.39_{24.91}$ | $15.36_{2.71}$ | $22.41_{11.43}$ |
| $(IA)^3$ | $64.72_{1.83}$ | $38.26_{7.27}$ | $26.77_{2.79}$ | $55.57_{1.48}$ | $31.11_{2.53}$ | $43.29_{3.18}$ |
| FINETUNE-M | $59.04_{7.86}$ | $22.95_{12.78}$ | $10.75_{5.31}$ | $29.76_{21.25}$ | $9.65_{1.25}$ | $26.43_{9.69}$ |
| ADAPTER-M | $65.66_{7.13}$ | $37.65_{11.25}$ | $28.51_{7.80}$ | $43.40_{16.06}$ | $27.44_{1.68}$ | $40.53_{8.78}$ |
| PROPETL-M | $70.56_{1.06}$ | $41.58_{6.27}$ | $35.91_{3.46}$ | $42.20_{14.55}$ | $29.67_{6.92}$ | $43.98_{6.45}$ |
| HYPERFORMER | $47.74_{20.72}$ | $29.06_{11.76}$ | $22.16_{8.44}$ | $35.92_{17.37}$ | $22.58_{10.58}$ | $31.49_{13.77}$ |
| HYPERFORMER++ | $0.00_{0.00}$ | $0.00_{0.00}$ | $0.00_{0.00}$ | $0.00_{0.00}$ | $0.00_{0.00}$ | $0.00_{0.00}$ |
| ADAPTERFUSION | $72.53_{0.45}$ | $51.33_{0.23}$ | $43.75_{0.52}$ | $62.31_{0.25}$ | $42.78_{2.11}$ | $54.54_{0.71}$ |
| ADAPTERSOUP | $52.54_{1.61}$ | $24.07_{2.18}$ | $20.62_{0.28}$ | $31.16_{1.40}$ | $12.84_{0.49}$ | $28.25_{1.19}$ |
| SCALEARN | $\underline{73.32}_{0.08}$ | $\underline{53.94}_{0.13}$ | $44.14_{0.75}$ | $\underline{63.89}_{0.16}$ | $44.75_{0.47}$ | $\underline{56.01}_{0.32}$ |
| SCALEARNUNIFORM | $72.56_{0.20}$ | $50.59_{0.10}$ | $44.62_{0.00}$ | $62.66_{0.00}$ | $45.16_{0.00}$ | $55.12_{0.06}$ |
| SCALEARN++ | $73.18_{0.04}$ | $51.41_{0.36}$ | $44.10_{0.09}$ | $63.37_{0.02}$ | $\underline{45.43}_{0.24}$ | $55.50_{0.15}$ |
| SCALEARNUNIFORM++ | $73.02_{0.20}$ | $50.84_{0.30}$ | $\mathbf{44.88}_{0.39}$ | $62.87_{0.01}$ | $44.45_{0.02}$ | $55.21_{0.18}$ |

**Table 11:** Evaluation results on the combination of all GLUE and SuperGLUE tasks using RoBERTa$_{BASE}$. (Top) STL models, only learning a single task at a time. (Middle) Joint MTL methods, learning all tasks simultaneously. (Bottom) Two-stage MTL methods, composing the knowledge of several source adapters. The overall best results are underlined, and the best results among the two-stage MTL models are shown in **bold**.

| Model | MNLI | QQP | QNLI | SST-2 | STS-B | MRPC | RTE | CoLA | ReCoRD | MultiRC | BoolQ | WiC | WSC | COPA | CB | Avg. |
|---|---|---|---|---|---|---|---|---|---|---|---|---|---|---|---|---|
| FINETUNE | $86.61_{0.51}$ | $90.32_{0.15}$ | $91.78_{0.28}$ | $93.33_{0.48}$ | $90.53_{0.22}$ | $86.94_{1.52}$ | $73.47_{2.05}$ | $58.46_{4.03}$ | $71.61_{0.85}$ | $71.64_{1.15}$ | $76.80_{1.34}$ | $66.38_{2.08}$ | $63.46_{0.00}$ | $68.60_{6.74}$ | $81.96_{1.33}$ | $78.12_{1.72}$ |
| ADAPTER | $86.50_{0.33}$ | $90.18_{0.11}$ | $92.25_{0.19}$ | $93.65_{0.71}$ | $90.23_{0.41}$ | $86.64_{1.07}$ | $72.89_{2.54}$ | $58.28_{2.50}$ | $79.02_{0.62}$ | $72.84_{0.48}$ | $76.71_{1.38}$ | $65.58_{1.56}$ | $63.46_{0.00}$ | $70.20_{4.13}$ | $84.82_{3.18}$ | $78.88_{1.28}$ |
| PROPETL | $86.19_{0.25}$ | $88.88_{0.48}$ | $92.05_{0.80}$ | $93.81_{0.72}$ | $90.03_{0.35}$ | $85.93_{1.22}$ | $74.19_{2.03}$ | $59.29_{2.07}$ | $80.29_{0.24}$ | $73.07_{0.49}$ | $76.55_{0.78}$ | $66.60_{1.65}$ | $63.46_{0.00}$ | $70.60_{3.44}$ | $84.46_{3.86}$ | $79.05_{1.21}$ |
| COMPACTER++ | $85.62_{0.42}$ | $88.84_{0.70}$ | $91.79_{0.39}$ | $93.58_{0.34}$ | $89.67_{0.54}$ | $87.21_{0.61}$ | $72.02_{2.21}$ | $58.49_{2.58}$ | $77.69_{2.67}$ | $70.44_{0.57}$ | $75.88_{0.96}$ | $66.44_{1.63}$ | $63.46_{0.00}$ | $68.30_{4.00}$ | $87.68_{3.62}$ | $78.48_{1.42}$ |
| (IA)³ | $84.24_{1.01}$ | $88.37_{0.20}$ | $90.57_{0.38}$ | $93.35_{0.30}$ | $89.93_{0.30}$ | $87.11_{1.14}$ | $72.56_{2.23}$ | $56.57_{5.39}$ | $75.27_{0.23}$ | $70.32_{0.49}$ | $76.31_{0.79}$ | $67.07_{1.68}$ | $63.35_{0.32}$ | $69.30_{8.37}$ | $87.32_{4.57}$ | $78.11_{1.49}$ |
| FINETUNE-M | $85.82_{0.22}$ | $90.18_{0.26}$ | $91.34_{0.28}$ | $93.46_{0.83}$ | $86.52_{0.86}$ | $83.66_{1.01}$ | $80.99_{1.62}$ | $49.23_{4.09}$ | $64.01_{0.91}$ | $71.94_{0.21}$ | $76.86_{0.40}$ | $52.30_{1.24}$ | $57.05_{7.86}$ | $73.00_{1.63}$ | $82.14_{2.53}$ | $75.90_{1.60}$ |
| ADAPTER-M | $86.14_{0.21}$ | $89.70_{0.17}$ | $91.38_{0.22}$ | $93.27_{0.48}$ | $88.67_{0.31}$ | $85.95_{0.76}$ | $80.87_{1.53}$ | $48.88_{1.38}$ | $69.79_{0.34}$ | $72.10_{0.29}$ | $75.65_{0.20}$ | $53.92_{0.78}$ | $62.18_{1.20}$ | $73.33_{1.70}$ | $82.14_{2.53}$ | $76.93_{0.81}$ |
| PROPETL-M | $85.40_{0.55}$ | $88.11_{0.35}$ | $91.56_{0.27}$ | $93.50_{0.38}$ | $90.31_{0.48}$ | $86.85_{0.58}$ | $79.78_{2.84}$ | $49.53_{3.63}$ | $70.33_{0.87}$ | $72.35_{0.63}$ | $75.78_{0.85}$ | $54.18_{2.26}$ | $52.56_{11.44}$ | $77.35_{0.47}$ | $87.50_{1.46}$ | $77.00_{1.80}$ |
| HYPERFORMER | $86.05_{0.45}$ | $88.82_{0.46}$ | $91.87_{0.25}$ | $93.81_{0.34}$ | $89.99_{0.97}$ | $86.85_{0.81}$ | $81.83_{2.09}$ | $56.11_{1.71}$ | $71.18_{0.73}$ | $71.59_{1.97}$ | $76.30_{1.63}$ | $51.83_{1.53}$ | $58.97_{6.35}$ | $64.67_{10.14}$ | $85.71_{1.46}$ | $77.04_{2.06}$ |
| HYPERFORMER++ | $86.32_{0.17}$ | $89.04_{0.20}$ | $91.71_{0.15}$ | $93.31_{0.59}$ | $90.80_{0.48}$ | $88.81_{1.21}$ | $83.15_{0.95}$ | $55.40_{3.13}$ | $71.83_{0.40}$ | $71.18_{0.69}$ | $76.90_{0.25}$ | $55.02_{1.26}$ | $60.90_{2.27}$ | $76.00_{0.82}$ | $85.71_{1.46}$ | $78.41_{0.93}$ |
| ADAPTERFUSION | $86.52_{0.20}$ | $90.18_{0.11}$ | $92.35_{0.16}$ | $93.62_{0.69}$ | $90.46_{0.31}$ | $87.89_{1.00}$ | $78.84_{1.63}$ | $58.67_{1.42}$ | $78.66_{0.94}$ | $72.71_{0.71}$ | $76.63_{0.71}$ | $66.36_{1.34}$ | $\underline{63.46}_{0.00}$ | $74.30_{3.02}$ | $83.57_{5.70}$ | $79.62_{1.20}$ |
| ADAPTERSOUP | $58.04_{0.62}$ | $83.02_{0.03}$ | $77.06_{0.26}$ | $91.06_{0.16}$ | $65.93_{0.27}$ | $71.57_{0.49}$ | $59.03_{0.93}$ | $36.83_{2.71}$ | $62.02_{0.12}$ | $35.06_{1.66}$ | $68.54_{0.32}$ | $58.86_{0.80}$ | $62.50_{1.61}$ | $52.50_{4.18}$ | $71.13_{0.73}$ | $63.54_{0.99}$ |
| SCALEARN | $86.93_{0.03}$ | $89.78_{0.09}$ | $92.78_{0.03}$ | $94.65_{0.35}$ | $\underline{90.97}_{0.09}$ | $88.21_{0.72}$ | $81.59_{1.69}$ | $59.32_{1.81}$ | $78.50_{0.48}$ | $72.67_{0.42}$ | $\underline{78.59}_{0.28}$ | $66.76_{1.70}$ | $\underline{63.46}_{0.51}$ | $\underline{80.60}_{3.27}$ | $\underline{96.07}_{1.41}$ | $\underline{81.39}_{0.86}$ |
| SCALEARNUNIFORM | $\underline{87.02}_{0.07}$ | $\underline{90.26}_{0.10}$ | $92.01_{0.92}$ | $94.38_{0.30}$ | $90.16_{0.11}$ | $87.97_{0.99}$ | $80.87_{1.09}$ | $58.97_{0.83}$ | $\underline{80.05}_{0.18}$ | $71.90_{0.29}$ | $76.42_{0.65}$ | $\underline{68.07}_{0.77}$ | $63.22_{0.85}$ | $73.30_{2.16}$ | $93.93_{1.73}$ | $80.57_{0.74}$ |
| SCALEARN++ | $86.94_{0.01}$ | $89.56_{0.27}$ | $\underline{92.80}_{0.08}$ | $94.04_{0.30}$ | $90.75_{0.16}$ | $\underline{88.21}_{1.05}$ | $80.40_{0.90}$ | $\underline{59.65}_{1.06}$ | $79.98_{0.21}$ | $71.16_{0.37}$ | $77.34_{0.37}$ | $67.43_{1.58}$ | $\underline{63.46}_{0.00}$ | $79.90_{1.66}$ | $94.29_{2.35}$ | $81.06_{0.76}$ |
| SCALEARNUNIFORM++ | $86.82_{0.17}$ | $90.16_{0.34}$ | $92.35_{0.31}$ | $94.61_{0.11}$ | $90.32_{0.14}$ | $87.97_{0.86}$ | $80.79_{0.96}$ | $59.33_{0.90}$ | $79.80_{0.56}$ | $\underline{72.76}_{0.51}$ | $76.22_{0.69}$ | $67.95_{1.04}$ | $61.78_{1.98}$ | $74.20_{1.75}$ | $93.21_{0.75}$ | $80.55_{0.74}$ |

**Table 12:** Evaluation results on the combination of all GLUE and SuperGLUE tasks using RoBERTa$_{LARGE}$.

| Model | MNLI | QQP | QNLI | SST-2 | STS-B | MRPC | RTE | CoLA | ReCoRD | MultiRC | BoolQ | WiC | WSC | COPA | CB | Avg. |
|---|---|---|---|---|---|---|---|---|---|---|---|---|---|---|---|---|
| FINETUNE | $89.57_{0.36}$ | $89.75_{1.03}$ | $93.91_{0.43}$ | $95.30_{0.65}$ | $91.89_{0.35}$ | $86.27_{1.15}$ | $81.52_{3.19}$ | $60.15_{2.89}$ | $81.60_{1.25}$ | $79.03_{0.02}$ | $81.65_{0.30}$ | $69.72_{2.16}$ | $63.46_{0.00}$ | $52.00_{8.28}$ | $90.36_{2.99}$ | $80.41_{1.67}$ |
| ADAPTER | $89.62_{0.18}$ | $89.87_{0.67}$ | $94.13_{0.06}$ | $95.24_{0.08}$ | $91.81_{0.29}$ | $87.82_{2.11}$ | $81.23_{2.92}$ | $64.07_{1.97}$ | $88.52_{0.09}$ | $80.73_{0.69}$ | $82.36_{0.72}$ | $69.16_{1.31}$ | $63.25_{0.64}$ | $71.90_{13.63}$ | $92.68_{1.78}$ | $82.83_{1.81}$ |
| PROPETL | $89.78_{0.24}$ | $89.23_{0.77}$ | $94.32_{0.09}$ | $95.41_{0.00}$ | $91.45_{0.39}$ | $87.65_{0.73}$ | $84.55_{2.14}$ | $65.85_{2.10}$ | $87.86_{2.59}$ | $81.19_{0.99}$ | $81.61_{0.86}$ | $69.62_{2.16}$ | $63.46_{0.00}$ | $69.00_{18.96}$ | $94.11_{4.04}$ | $83.01_{2.40}$ |
| COMPACTER++ | $89.15_{0.67}$ | $87.33_{2.39}$ | $92.93_{1.42}$ | $95.41_{0.00}$ | $91.46_{0.35}$ | $87.84_{1.23}$ | $79.71_{4.58}$ | $65.66_{2.08}$ | $88.34_{0.97}$ | $79.53_{6.13}$ | $79.18_{0.29}$ | $69.26_{1.51}$ | $62.26_{1.43}$ | $79.00_{0.74}$ | $87.50_{7.48}$ | $82.30_{2.69}$ |
| (IA)³ | $88.68_{0.61}$ | $87.79_{0.72}$ | $91.72_{0.79}$ | $94.95_{0.16}$ | $91.39_{0.45}$ | $86.37_{1.65}$ | $80.79_{3.16}$ | $64.70_{3.20}$ | $87.47_{0.21}$ | $77.91_{0.43}$ | $80.97_{0.75}$ | $68.65_{2.55}$ | $60.58_{0.90}$ | $74.00_{11.79}$ | $90.00_{3.91}$ | $81.73_{2.01}$ |
| FINETUNE-M | $88.23_{0.10}$ | $89.81_{0.08}$ | $92.48_{0.28}$ | $93.20_{0.76}$ | $85.41_{0.75}$ | $79.25_{3.06}$ | $84.12_{0.51}$ | $51.48_{3.50}$ | $74.48_{0.36}$ | $75.07_{0.39}$ | $78.99_{0.73}$ | $52.40_{0.70}$ | $58.01_{7.71}$ | $77.67_{3.40}$ | $81.55_{3.67}$ | $77.48_{1.73}$ |
| ADAPTER-M | $89.30_{0.31}$ | $90.04_{0.30}$ | $93.90_{0.14}$ | $95.18_{0.34}$ | $89.41_{0.31}$ | $85.46_{1.86}$ | $86.52_{1.19}$ | $57.36_{1.22}$ | $81.81_{1.12}$ | $77.81_{0.52}$ | $80.53_{0.63}$ | $55.69_{2.07}$ | $59.29_{2.52}$ | $83.33_{0.94}$ | $80.95_{2.23}$ | $80.44_{1.05}$ |
| PROPETL-M | $88.93_{0.30}$ | $88.25_{0.05}$ | $93.69_{0.26}$ | $95.03_{0.19}$ | $90.44_{0.60}$ | $86.11_{0.42}$ | $86.40_{0.74}$ | $57.86_{3.83}$ | $83.85_{0.17}$ | $77.86_{0.19}$ | $78.65_{1.39}$ | $51.20_{0.96}$ | $45.83_{12.47}$ | $90.33_{3.09}$ | $87.50_{1.46}$ | $80.13_{1.74}$ |
| HYPERFORMER | $89.75_{0.34}$ | $90.05_{0.22}$ | $94.37_{0.11}$ | $95.57_{0.30}$ | $91.80_{0.31}$ | $86.76_{1.44}$ | $88.69_{0.74}$ | $62.34_{1.85}$ | $85.16_{0.14}$ | $79.78_{0.37}$ | $82.16_{0.81}$ | $51.93_{2.20}$ | $61.86_{2.27}$ | $89.00_{4.90}$ | $84.52_{1.68}$ | $82.25_{1.18}$ |
| ADAPTERFUSION | $89.79_{0.12}$ | $\underline{90.83}_{0.27}$ | $94.14_{0.05}$ | $95.64_{0.00}$ | $92.08_{0.14}$ | $89.12_{0.22}$ | $85.85_{2.48}$ | $66.52_{1.31}$ | $\underline{89.26}_{0.00}$ | $79.25_{0.80}$ | $82.40_{0.54}$ | $69.50_{1.12}$ | $62.69_{1.72}$ | $88.60_{3.36}$ | $90.36_{2.99}$ | $84.40_{1.01}$ |
| ADAPTERSOUP | $57.29_{0.00}$ | $84.01_{0.00}$ | $73.71_{0.00}$ | $94.04_{0.00}$ | $71.16_{0.86}$ | $71.51_{0.23}$ | $63.54_{3.90}$ | $42.22_{3.27}$ | $68.09_{0.00}$ | $31.32_{0.51}$ | $72.48_{0.22}$ | $62.54_{1.06}$ | $62.98_{0.68}$ | $54.50_{3.87}$ | $71.88_{1.71}$ | $65.42_{0.09}$ |
| SCALEARN | $89.67_{0.13}$ | $89.70_{0.58}$ | $93.98_{0.31}$ | $95.36_{0.57}$ | $92.29_{0.13}$ | $88.28_{1.37}$ | $85.78_{1.16}$ | $67.20_{1.33}$ | $85.43_{0.44}$ | $80.08_{0.69}$ | $82.43_{0.79}$ | $70.16_{2.08}$ | $\underline{66.73}_{4.79}$ | $\underline{91.00}_{1.22}$ | $93.93_{2.04}$ | $\underline{84.80}_{0.18}$ |
| SCALEARNUNIFORM | $90.09_{0.04}$ | $90.54_{0.68}$ | $93.84_{0.69}$ | $\underline{95.70}_{0.08}$ | $92.13_{0.05}$ | $88.33_{1.12}$ | $85.85_{2.03}$ | $66.85_{1.05}$ | $88.24_{0.00}$ | $\underline{80.50}_{0.07}$ | $82.04_{0.16}$ | $70.28_{2.47}$ | $59.62_{0.00}$ | $90.40_{1.22}$ | $95.00_{0.33}$ | $84.63_{0.80}$ |
| SCALEARN++ | $\underline{90.13}_{0.27}$ | $90.22_{0.93}$ | $\underline{94.49}_{0.23}$ | $94.61_{2.11}$ | $\underline{92.35}_{0.10}$ | $87.70_{0.96}$ | $\underline{86.21}_{1.00}$ | $\underline{67.22}_{1.28}$ | $87.53_{0.13}$ | $80.14_{0.29}$ | $\underline{82.51}_{1.95}$ | $69.40_{1.63}$ | $62.82_{1.11}$ | $89.80_{1.10}$ | $94.29_{0.80}$ | $84.63_{0.93}$ |
| SCALEARNUNIFORM++ | $90.10_{0.14}$ | $90.45_{0.10}$ | $93.91_{0.09}$ | $95.30_{0.00}$ | $92.12_{0.12}$ | $\underline{88.97}_{0.88}$ | $84.77_{1.34}$ | $65.83_{1.49}$ | $88.28_{0.56}$ | $80.46_{0.23}$ | $82.23_{0.28}$ | $70.09_{0.36}$ | $60.10_{0.68}$ | $89.20_{1.30}$ | $\underline{95.36}_{0.98}$ | $84.48_{0.57}$ |

## A.5 SCALING COEFFICIENT VISUALIZATIONS

SCALEARNUNIFORM and SCALEARNUNIFORM++ utilize uniform scaling and learn coefficients that are directly used to scale the output representations of the source adapters. In the following, we leverage this characteristic to provide an analysis of the potential degrees of effects of source tasks on target tasks. We present the adapter weights learned using RoBERTa$_{\text{BASE}}$ for GLUE and SuperGLUE, and using XLM-R$_{\text{BASE}}$ for HumSet with the random seed set to 0.

The learned coefficients of each PLM layer on GLUE, SuperGLUE, and HumSet of SCALEAR-NUNIFORM are shown in Figure 6, Figure 7, and Figure 8, respectively. The weights reveal that in most cases, the actual target task adapter is activated most strongly across the layers. Among the source tasks, most weights are close to 0, while some source tasks also show high values, particularly in some of the higher layers of the PLM. Interestingly, some of the scaling coefficients go beyond or even below 1, which would not have been possible in the traditional paradigm where scaling coefficients combining multiple vectors are restricted to sum up to 1.

The learned weights on GLUE, SuperGLUE, and HumSet of SCALEARNUNIFORM++ are shown in Figure 9. SCALEARNUNIFORM++ also mostly activates the actual target task adapter, whereas this effect is comparatively weaker in SuperGLUE and stronger in HumSet. As is the case with SCALEARNUNIFORM, many scaling coefficients exceed or go below 1.

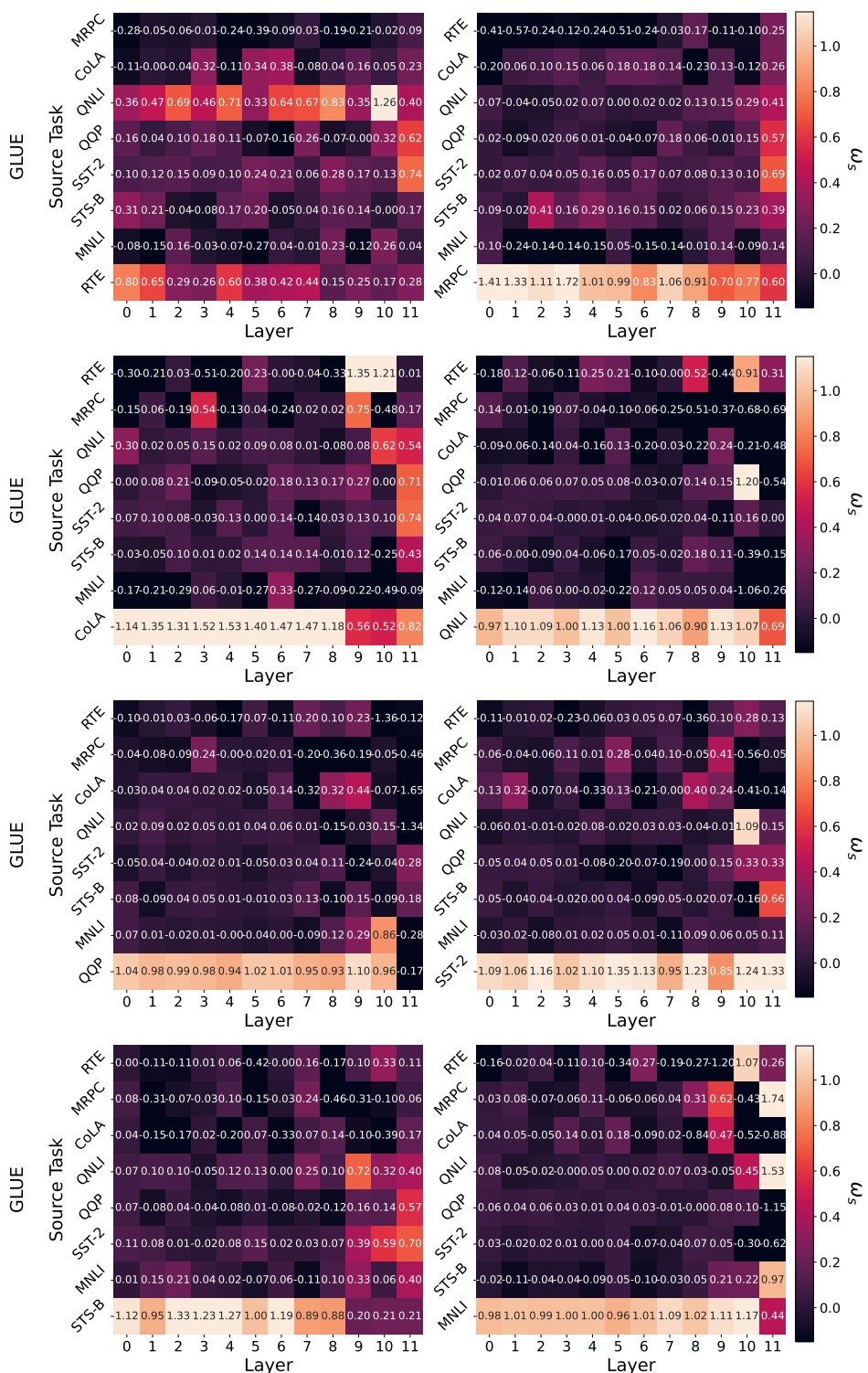

**Figure 6:** SCALEARNUNIFORM scaling coefficients on GLUE using RoBERTa$_{\text{BASE}}$ on seed 0. Target tasks are shown in the last index of each heatmap.

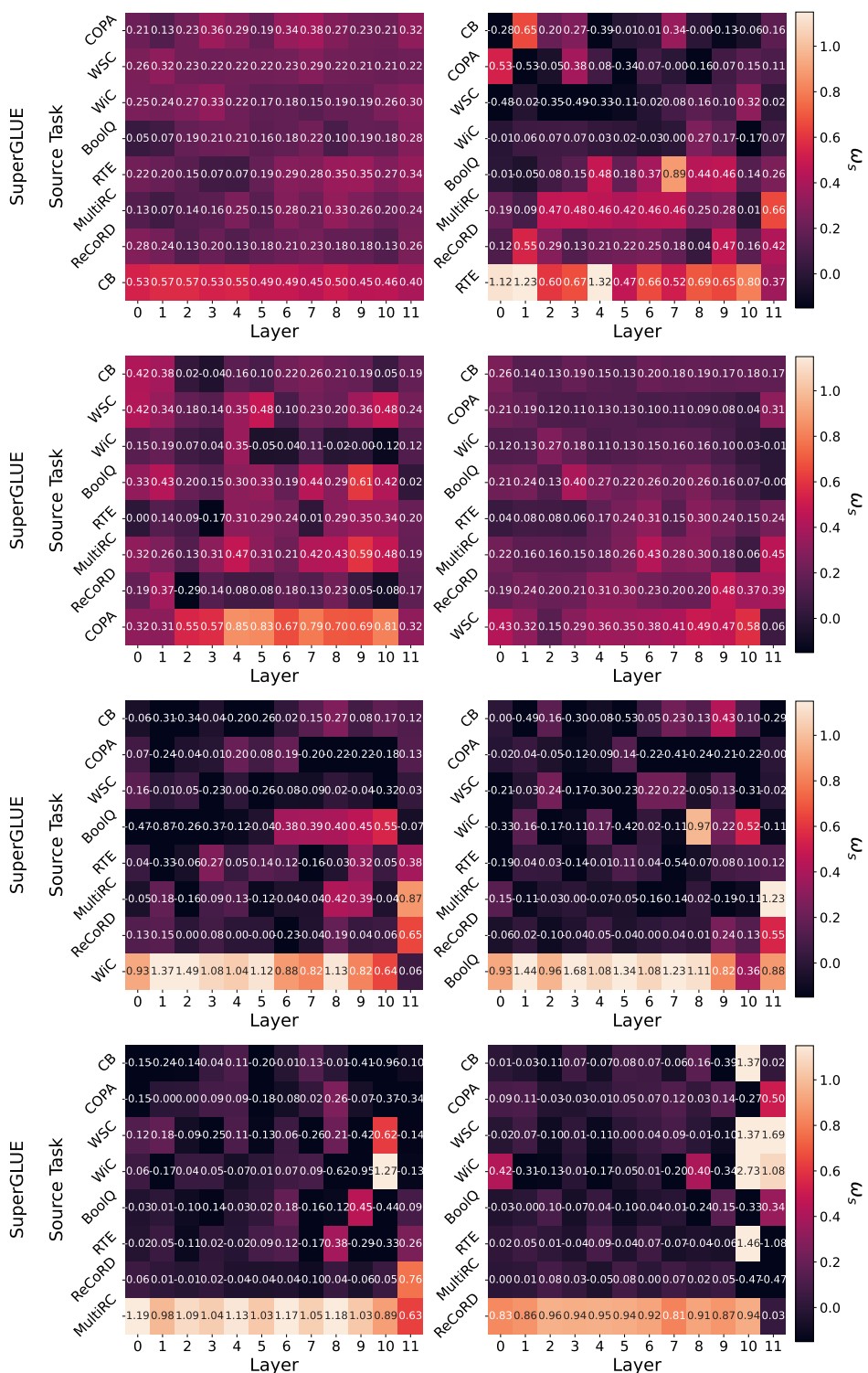

**Figure 7:** SCALEARNUNIFORM scaling coefficients on SuperGLUE using RoBERTa$_{BASE}$ on seed 0. Target tasks are shown in the last index of each heatmap.

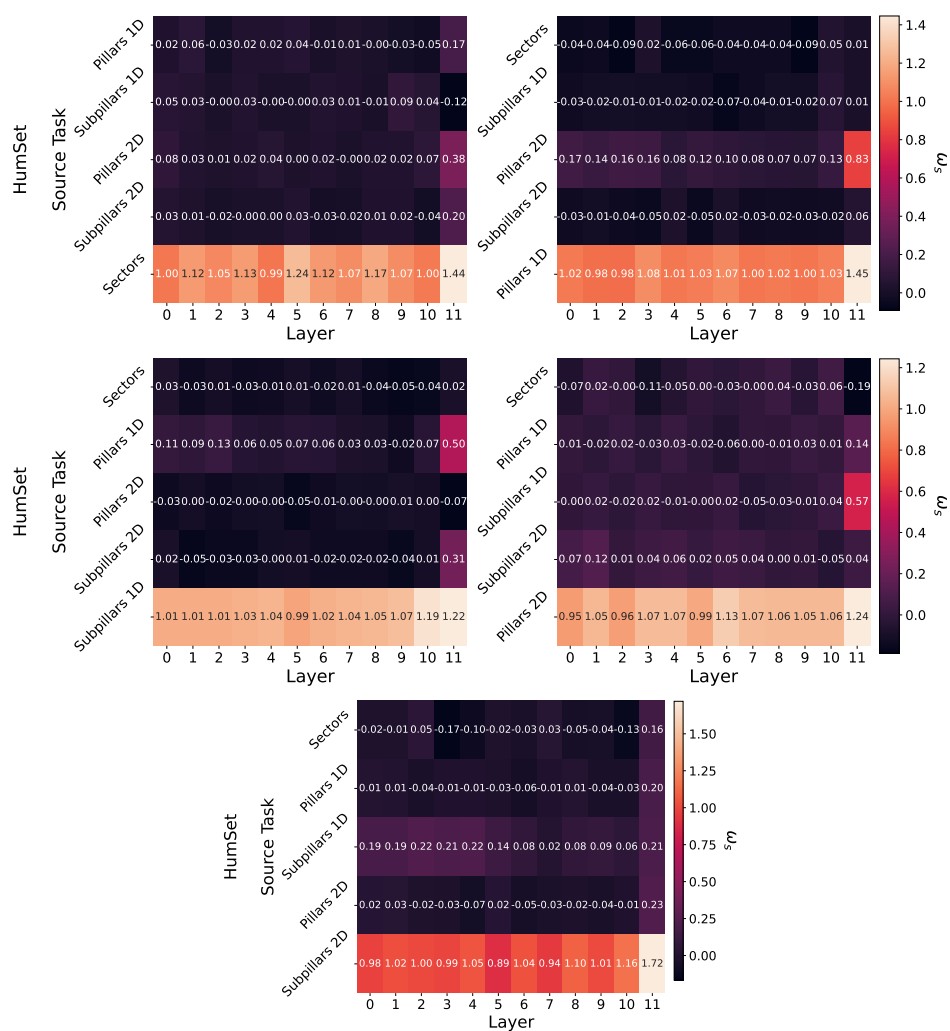

**Figure 8:** SCALEARNUNIFORM scaling coefficients on HumSet using XLM-R_BASE on seed 0. Target tasks are shown in the last index of each heatmap.

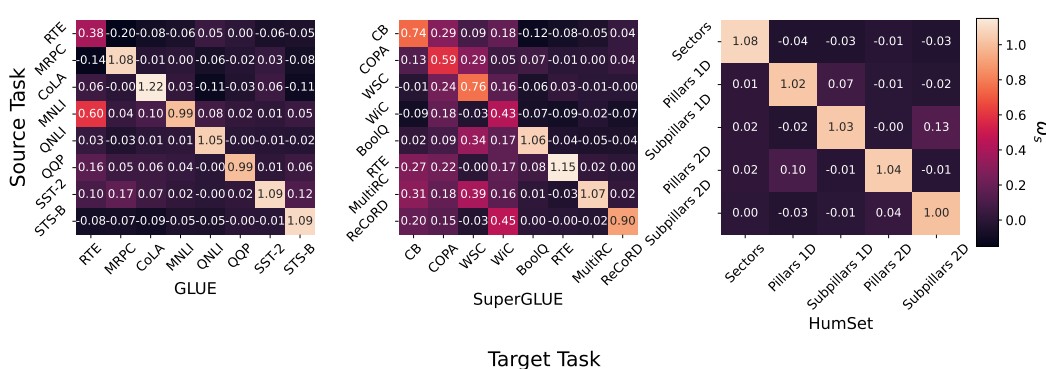

**Figure 9:** SCALEARNUNIFORM++ scaling coefficients on GLUE, SuperGLUE, and HumSet using RoBERTa_BASE for GLUE and SuperGLUE and XLM-R_BASE for HumSet on seed 0.

A.6   COMPLETE FEW-SHOT RESULTS

To obtain a more complete understanding of the few-shot capabilities of ADAPTER, ADAPTERFU-SION, and SCALEARN, we show few-shot transfer learning results for each dataset, as well as for every variant of SCALEARN (cf. Section 6.3).

**Few-shot results using RoBERTa$_{BASE}$.** Table 13 shows the few-shot transfer learning performance of the methods on the GLUE benchmark using $k = \{4,16,32,100\}$ samples. Table 14 shows the performance of the methods on SuperGLUE. Table 15 shows the performance of the methods on HumSet (on XLM-R)$_{BASE}$. Finally, Table 16 shows the results when training on the combination of all GLUE and SuperGLUE tasks, resulting in $|S| = 15$ source tasks.

**Few-shot results using RoBERTa$_{LARGE}$.** Figure 10 provides an overview, comparing the few-shot learning capabilities of ADAPTER, ADAPTERFUSION, and SCALEARN when using RoBERTa$_{LARGE}$. Moreover, Table 17 shows the few-shot learning performance of the methods on the GLUE benchmark using $k = \{4,16,32,100\}$ samples. Table 18 shows the performance of the methods on SuperGLUE. Table 19 shows the performance of the methods on HumSet (on XLM-R$_{LARGE}$). Finally, Table 20 shows the results when training on the combination of all GLUE and SuperGLUE tasks.

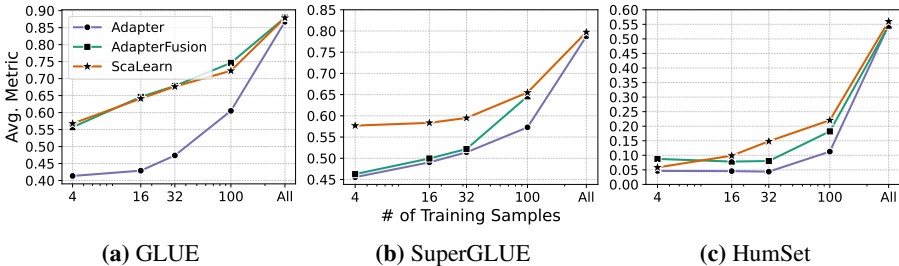

| (a) GLUE | (b) SuperGLUE | (c) HumSet |

**Figure 10:** Few-shot learning results ($k = \{4,16,32,100\}$) comparing ADAPTER, ADAPTERFUSION, and SCALEARN using RoBERTa$_{LARGE}$ on three benchmarks. We show the mean across 5 seeds. For ADAPTERFUSION and SCALEARN, we assume that there is a Pfeiffer adapter trained on the target task on $k$ samples and a Pfeiffer adapter trained on all samples for all other tasks available.

**Table 13:** Complete few-shot transfer learning results on GLUE with $k = \{4,16,32,100\}$ training samples for each target task using RoBERTa$_{\text{BASE}}$.

| Model | Samples | MNLI | QQP | QNLI | SST-2 | STS-B | MRPC | RTE | CoLA | Avg. |
|---|---|---|---|---|---|---|---|---|---|---|
| ADAPTER | 4 | $33.65_{1.39}$ | $63.27_{0.11}$ | $50.53_{0.04}$ | $50.92_{0.00}$ | $32.12_{9.28}$ | $68.38_{0.00}$ | $52.71_{0.00}$ | $2.93_{3.88}$ | $44.31_{1.84}$ |
| ADAPTER | 16 | $34.78_{0.58}$ | $63.18_{0.00}$ | $50.46_{0.20}$ | $57.18_{1.23}$ | $55.53_{10.12}$ | $68.38_{0.00}$ | $53.72_{1.29}$ | $0.25_{0.56}$ | $47.94_{1.75}$ |
| ADAPTER | 32 | $33.56_{0.66}$ | $63.18_{0.00}$ | $51.86_{0.33}$ | $70.46_{2.25}$ | $73.78_{1.30}$ | $68.38_{0.00}$ | $54.58_{1.81}$ | $0.00_{0.00}$ | $51.98_{0.80}$ |
| ADAPTER | 100 | $40.71_{2.67}$ | $71.74_{0.50}$ | $58.77_{4.13}$ | $85.00_{2.25}$ | $82.51_{1.21}$ | $73.09_{1.27}$ | $56.17_{1.95}$ | $21.69_{3.94}$ | $61.21_{2.24}$ |
| ADAPTER | All | $86.50_{0.33}$ | $90.18_{0.11}$ | $92.25_{0.19}$ | $93.65_{0.71}$ | $90.23_{0.41}$ | $86.64_{1.07}$ | $72.89_{2.54}$ | $58.28_{2.50}$ | $83.83_{0.98}$ |
| ADAPTERFUSION | 4 | $33.94_{2.09}$ | $72.01_{5.39}$ | $52.36_{2.75}$ | $50.92_{0.00}$ | $77.17_{2.44}$ | $72.99_{4.28}$ | $52.78_{0.16}$ | $2.79_{3.54}$ | $51.87_{2.58}$ |
| ADAPTERFUSION | 16 | $49.12_{2.76}$ | $76.26_{1.20}$ | $61.95_{11.04}$ | $59.29_{6.12}$ | $83.51_{1.79}$ | $78.28_{0.37}$ | $60.65_{2.27}$ | $0.92_{1.82}$ | $58.75_{3.42}$ |
| ADAPTERFUSION | 32 | $43.89_{3.17}$ | $76.45_{0.83}$ | $78.35_{0.75}$ | $68.26_{5.11}$ | $70.72_{30.12}$ | $78.87_{1.63}$ | $60.87_{4.48}$ | $1.91_{4.27}$ | $59.91_{6.30}$ |
| ADAPTERFUSION | 100 | $47.22_{5.48}$ | $77.23_{1.74}$ | $77.80_{5.43}$ | $85.28_{2.42}$ | $85.81_{1.64}$ | $78.43_{1.34}$ | $70.04_{1.17}$ | $13.95_{7.80}$ | $66.97_{3.38}$ |
| ADAPTERFUSION | All | $86.82_{0.04}$ | $90.23_{0.01}$ | $92.48_{0.15}$ | $93.23_{0.95}$ | $90.37_{0.20}$ | $88.41_{0.49}$ | $79.49_{2.21}$ | $59.04_{1.69}$ | $85.01_{0.72}$ |
| SCALEARN | 4 | $35.59_{2.13}$ | $76.24_{0.38}$ | $62.30_{4.58}$ | $52.68_{0.66}$ | $85.34_{0.98}$ | $75.00_{1.59}$ | $52.71_{0.00}$ | $4.25_{0.83}$ | $55.51_{1.39}$ |
| SCALEARN | 16 | $51.21_{0.84}$ | $76.85_{0.19}$ | $65.03_{1.37}$ | $64.01_{0.90}$ | $86.18_{0.38}$ | $79.07_{0.68}$ | $62.74_{1.74}$ | $7.51_{2.36}$ | $61.58_{1.06}$ |
| SCALEARN | 32 | $51.91_{0.36}$ | $76.19_{0.18}$ | $73.63_{0.46}$ | $69.56_{3.25}$ | $86.34_{0.44}$ | $75.98_{0.39}$ | $65.42_{1.50}$ | $8.56_{1.70}$ | $63.45_{1.03}$ |
| SCALEARN | 100 | $57.88_{0.34}$ | $77.25_{0.39}$ | $73.97_{0.73}$ | $83.97_{1.76}$ | $87.81_{0.28}$ | $78.38_{1.36}$ | $69.17_{1.70}$ | $13.31_{1.71}$ | $67.72_{1.03}$ |
| SCALEARN | All | $86.97_{0.09}$ | $90.32_{0.10}$ | $92.51_{0.17}$ | $93.88_{0.18}$ | $90.96_{0.16}$ | $87.75_{0.58}$ | $82.06_{1.37}$ | $58.47_{1.76}$ | $85.36_{0.55}$ |
| SCALEARN++ | 4 | $34.05_{1.78}$ | $75.50_{0.56}$ | $59.88_{4.74}$ | $52.25_{0.70}$ | $85.20_{0.80}$ | $72.99_{1.46}$ | $52.71_{0.00}$ | $3.87_{2.20}$ | $54.55_{1.53}$ |
| SCALEARN++ | 16 | $50.52_{1.42}$ | $76.30_{0.60}$ | $60.40_{3.04}$ | $62.20_{1.99}$ | $85.96_{0.30}$ | $78.04_{1.58}$ | $61.59_{0.98}$ | $9.00_{2.05}$ | $60.50_{1.49}$ |
| SCALEARN++ | 32 | $52.30_{1.35}$ | $75.71_{0.65}$ | $72.01_{2.62}$ | $71.90_{2.37}$ | $86.04_{0.37}$ | $76.18_{1.07}$ | $63.68_{0.94}$ | $7.54_{3.03}$ | $63.17_{1.55}$ |
| SCALEARN++ | 100 | $56.16_{0.83}$ | $76.60_{0.76}$ | $61.66_{5.15}$ | $83.07_{1.92}$ | $87.24_{0.20}$ | $77.89_{1.19}$ | $65.05_{2.95}$ | $11.50_{1.47}$ | $64.90_{1.81}$ |
| SCALEARN++ | All | $87.06_{0.03}$ | $90.04_{0.12}$ | $92.03_{1.10}$ | $94.15_{0.30}$ | $90.87_{0.19}$ | $87.43_{0.58}$ | $80.87_{1.05}$ | $59.82_{0.78}$ | $85.35_{0.52}$ |
| SCALEARNUNIFORM | 4 | $34.17_{1.67}$ | $76.62_{0.62}$ | $55.25_{2.01}$ | $52.48_{1.37}$ | $84.47_{0.97}$ | $75.44_{1.75}$ | $52.71_{0.00}$ | $5.09_{1.50}$ | $54.53_{1.24}$ |
| SCALEARNUNIFORM | 16 | $49.55_{1.21}$ | $76.60_{0.32}$ | $66.69_{1.07}$ | $65.05_{2.42}$ | $85.83_{0.40}$ | $77.65_{1.09}$ | $61.81_{1.95}$ | $10.96_{2.45}$ | $61.77_{1.36}$ |
| SCALEARNUNIFORM | 32 | $51.50_{1.92}$ | $76.28_{0.56}$ | $72.84_{0.54}$ | $71.49_{2.38}$ | $86.01_{0.43}$ | $75.88_{1.03}$ | $63.75_{1.16}$ | $11.15_{2.18}$ | $63.61_{1.28}$ |
| SCALEARNUNIFORM | 100 | $55.06_{1.23}$ | $76.94_{0.38}$ | $70.42_{2.28}$ | $81.63_{0.90}$ | $86.22_{0.45}$ | $75.93_{1.54}$ | $64.62_{1.02}$ | $15.54_{2.95}$ | $65.79_{1.35}$ |
| SCALEARNUNIFORM | All | $86.93_{0.10}$ | $90.37_{0.11}$ | $92.43_{0.36}$ | $93.58_{0.20}$ | $90.08_{0.07}$ | $87.57_{0.86}$ | $80.07_{1.18}$ | $59.04_{1.05}$ | $85.01_{0.49}$ |
| SCALEARNUNIFORM++ | 4 | $34.86_{2.18}$ | $76.08_{0.38}$ | $53.36_{3.84}$ | $51.79_{1.09}$ | $83.12_{1.63}$ | $74.80_{1.05}$ | $52.71_{0.00}$ | $4.34_{2.15}$ | $53.88_{1.54}$ |
| SCALEARNUNIFORM++ | 16 | $50.09_{0.81}$ | $76.13_{0.25}$ | $61.35_{3.09}$ | $62.59_{1.52}$ | $85.55_{0.40}$ | $76.42_{0.72}$ | $62.60_{0.70}$ | $11.94_{3.04}$ | $60.83_{1.32}$ |
| SCALEARNUNIFORM++ | 32 | $50.96_{1.64}$ | $76.15_{0.47}$ | $70.24_{0.96}$ | $71.97_{2.06}$ | $85.67_{0.41}$ | $74.41_{0.66}$ | $62.24_{0.66}$ | $12.85_{2.49}$ | $63.06_{1.17}$ |
| SCALEARNUNIFORM++ | 100 | $48.96_{1.99}$ | $76.77_{0.34}$ | $60.64_{3.67}$ | $81.90_{0.67}$ | $85.66_{0.63}$ | $75.69_{1.17}$ | $63.54_{1.53}$ | $15.90_{2.99}$ | $63.63_{1.62}$ |
| SCALEARNUNIFORM++ | All | $86.98_{0.17}$ | $90.38_{0.01}$ | $92.53_{0.28}$ | $94.11_{0.07}$ | $90.18_{0.19}$ | $87.43_{0.63}$ | $80.04_{0.99}$ | $59.45_{0.67}$ | $85.14_{0.38}$ |

**Table 14:** Complete few-shot transfer learning results on SuperGLUE with $k = \{4,16,32,100\}$ training samples for each target task using RoBERTa$_{\text{BASE}}$.

| Model | Samples | ReCoRD | Multi | BoolQ | WiC | WSC | COPA | CB | RTE | Avg. |
|---|---|---|---|---|---|---|---|---|---|---|
| ADAPTER | 4 | $9.65_{2.79}$ | $24.92_{6.71}$ | $62.05_{0.27}$ | $49.44_{1.26}$ | $41.92_{12.04}$ | $50.20_{3.63}$ | $62.14_{8.12}$ | $52.71_{0.00}$ | $44.13_{4.35}$ |
| ADAPTER | 16 | $13.82_{6.06}$ | $37.48_{8.48}$ | $62.17_{0.00}$ | $50.53_{1.18}$ | $42.50_{5.46}$ | $53.00_{5.48}$ | $69.29_{2.93}$ | $53.72_{1.29}$ | $47.81_{3.86}$ |
| ADAPTER | 32 | $17.64_{12.76}$ | $38.55_{3.74}$ | $62.16_{0.03}$ | $52.26_{1.01}$ | $36.54_{0.00}$ | $51.20_{2.39}$ | $70.71_{1.60}$ | $54.58_{1.81}$ | $47.95_{3.01}$ |
| ADAPTER | 100 | $37.69_{2.61}$ | $51.56_{3.89}$ | $61.51_{1.27}$ | $54.04_{1.01}$ | $50.38_{10.12}$ | $58.40_{5.18}$ | $73.93_{4.11}$ | $56.17_{1.95}$ | $55.46_{3.77}$ |
| ADAPTER | All | $79.02_{0.62}$ | $72.84_{0.48}$ | $76.71_{1.38}$ | $65.58_{1.56}$ | $63.46_{0.00}$ | $70.20_{4.13}$ | $84.82_{3.18}$ | $72.89_{2.54}$ | $73.19_{1.74}$ |
| ADAPTERFUSION | 4 | $8.51_{2.73}$ | $44.50_{24.40}$ | $62.16_{0.03}$ | $50.31_{1.04}$ | $38.08_{3.44}$ | $50.40_{2.19}$ | $51.07_{2.40}$ | $52.64_{1.31}$ | $44.71_{4.69}$ |
| ADAPTERFUSION | 16 | $13.71_{10.75}$ | $48.86_{14.98}$ | $62.12_{0.27}$ | $50.16_{1.84}$ | $38.46_{4.30}$ | $56.80_{7.22}$ | $52.92_{3.71}$ | $52.92_{3.71}$ | $48.86_{5.88}$ |
| ADAPTERFUSION | 32 | $26.79_{14.35}$ | $46.39_{16.63}$ | $62.03_{0.34}$ | $52.23_{0.87}$ | $37.12_{1.29}$ | $59.60_{5.86}$ | $68.93_{2.71}$ | $54.66_{2.35}$ | $50.97_{5.55}$ |
| ADAPTERFUSION | 100 | $34.02_{13.55}$ | $43.52_{4.01}$ | $61.83_{1.45}$ | $54.61_{1.07}$ | $43.85_{8.78}$ | $64.20_{3.83}$ | $74.64_{3.43}$ | $59.71_{1.63}$ | $54.55_{4.72}$ |
| ADAPTERFUSION | All | $78.82_{0.49}$ | $71.79_{1.67}$ | $76.72_{0.55}$ | $66.57_{1.24}$ | $63.46_{0.00}$ | $73.10_{4.51}$ | $82.32_{2.85}$ | $76.03_{2.38}$ | $73.60_{1.71}$ |
| SCALEARN | 4 | $28.37_{6.53}$ | $31.53_{11.93}$ | $61.63_{0.22}$ | $49.72_{0.39}$ | $49.62_{5.34}$ | $71.80_{4.49}$ | $66.79_{11.48}$ | $52.71_{0.00}$ | $51.52_{5.05}$ |
| SCALEARN | 16 | $31.07_{6.24}$ | $49.97_{7.42}$ | $60.92_{1.21}$ | $51.50_{0.49}$ | $51.35_{5.25}$ | $69.00_{5.24}$ | $72.86_{2.33}$ | $54.22_{1.31}$ | $55.11_{3.69}$ |
| SCALEARN | 32 | $34.80_{6.48}$ | $44.28_{3.71}$ | $61.70_{0.22}$ | $50.53_{0.94}$ | $48.08_{8.68}$ | $69.00_{6.34}$ | $76.07_{2.04}$ | $56.75_{1.18}$ | $55.10_{4.07}$ |
| SCALEARN | 100 | $40.82_{1.25}$ | $58.92_{2.28}$ | $62.11_{1.16}$ | $53.89_{0.99}$ | $61.92_{2.21}$ | $69.00_{2.74}$ | $86.79_{1.60}$ | $61.37_{1.71}$ | $61.85_{1.74}$ |
| SCALEARN | All | $79.52_{0.06}$ | $73.22_{0.44}$ | $77.27_{0.68}$ | $66.35_{1.20}$ | $63.46_{0.00}$ | $74.80_{2.15}$ | $90.89_{2.59}$ | $78.88_{2.14}$ | $75.55_{1.16}$ |
| SCALEARNUNIFORM | 4 | $22.64_{6.41}$ | $29.69_{6.54}$ | $61.72_{0.25}$ | $49.84_{0.86}$ | $44.62_{5.71}$ | $70.60_{2.30}$ | $70.36_{4.48}$ | $52.71_{0.00}$ | $50.27_{3.32}$ |
| SCALEARNUNIFORM | 16 | $30.01_{1.08}$ | $50.32_{7.20}$ | $61.72_{1.03}$ | $52.48_{0.70}$ | $49.81_{7.24}$ | $66.80_{2.17}$ | $73.93_{0.90}$ | $54.51_{2.75}$ | $54.95_{3.23}$ |
| SCALEARNUNIFORM | 32 | $30.84_{5.74}$ | $45.75_{5.47}$ | $61.41_{0.32}$ | $51.57_{0.73}$ | $48.27_{6.61}$ | $71.40_{2.30}$ | $75.71_{0.98}$ | $55.38_{0.75}$ | $55.04_{2.86}$ |
| SCALEARNUNIFORM | 100 | $35.50_{1.94}$ | $58.74_{2.59}$ | $61.36_{0.99}$ | $52.79_{0.58}$ | $56.97_{7.98}$ | $65.00_{2.00}$ | $82.86_{3.24}$ | $59.21_{1.28}$ | $59.05_{2.58}$ |
| SCALEARNUNIFORM | All | $80.13_{0.38}$ | $71.91_{0.60}$ | $76.06_{0.41}$ | $67.37_{1.22}$ | $62.50_{1.27}$ | $71.20_{1.23}$ | $89.11_{1.97}$ | $75.31_{0.90}$ | $74.20_{1.00}$ |
| SCALEARN++ | 4 | $27.53_{4.00}$ | $11.16_{6.18}$ | $60.92_{1.59}$ | $49.94_{0.50}$ | $44.62_{5.71}$ | $70.00_{2.24}$ | $62.50_{8.28}$ | $52.71_{0.00}$ | $47.42_{3.56}$ |
| SCALEARN++ | 16 | $25.78_{2.80}$ | $49.43_{10.93}$ | $59.86_{2.01}$ | $52.01_{0.62}$ | $49.42_{8.62}$ | $71.80_{1.10}$ | $74.64_{3.43}$ | $56.68_{1.17}$ | $54.95_{3.83}$ |
| SCALEARN++ | 32 | $34.00_{2.31}$ | $39.99_{5.10}$ | $59.80_{0.63}$ | $52.04_{0.53}$ | $42.50_{3.99}$ | $73.60_{4.56}$ | $75.71_{1.60}$ | $56.39_{0.86}$ | $54.25_{2.45}$ |
| SCALEARN++ | 100 | $37.32_{3.39}$ | $58.72_{1.28}$ | $60.43_{2.22}$ | $53.23_{0.61}$ | $62.12_{1.87}$ | $68.20_{1.83}$ | $85.71_{2.19}$ | $59.06_{1.89}$ | $60.35_{1.84}$ |
| SCALEARN++ | All | $80.13_{0.09}$ | $72.71_{0.57}$ | $76.44_{0.53}$ | $67.13_{1.24}$ | $62.26_{2.28}$ | $75.20_{1.93}$ | $93.04_{2.14}$ | $79.03_{0.95}$ | $75.74_{1.22}$ |
| SCALEARNUNIFORM++ | 4 | $23.04_{8.12}$ | $29.11_{2.02}$ | $61.02_{0.41}$ | $49.62_{1.41}$ | $46.73_{4.54}$ | $67.60_{5.68}$ | $66.43_{8.60}$ | $52.71_{0.00}$ | $49.53_{3.85}$ |
| SCALEARNUNIFORM++ | 16 | $26.67_{4.91}$ | $53.00_{8.69}$ | $61.06_{1.41}$ | $52.16_{0.67}$ | $50.96_{7.10}$ | $67.40_{2.97}$ | $74.29_{4.66}$ | $54.80_{2.74}$ | $55.04_{4.14}$ |
| SCALEARNUNIFORM++ | 32 | $30.62_{1.27}$ | $49.46_{6.35}$ | $59.88_{1.47}$ | $51.69_{0.70}$ | $44.62_{3.70}$ | $67.20_{1.66}$ | $78.21_{0.80}$ | $56.90_{1.07}$ | $54.82_{2.13}$ |
| SCALEARNUNIFORM++ | 100 | $29.77_{9.96}$ | $58.40_{2.35}$ | $60.77_{0.91}$ | $53.26_{1.87}$ | $61.15_{3.76}$ | $63.20_{2.77}$ | $80.00_{0.80}$ | $57.18_{1.74}$ | $57.97_{3.02}$ |
| SCALEARNUNIFORM++ | All | $79.79_{0.14}$ | $71.75_{0.38}$ | $76.13_{0.52}$ | $67.87_{0.89}$ | $63.46_{0.00}$ | $74.00_{1.70}$ | $91.61_{2.53}$ | $74.84_{1.58}$ | $74.93_{0.97}$ |

**Table 15:** Complete few-shot transfer learning results on HumSet with $k = \{4,16,32,100\}$ training samples for each target task using XLM-R$_{\text{BASE}}$.

| Model | Samples | Sectors | Pillars 1D | Subpillars 1D | Pillars 2D | Subpillars 2D | Avg. |
|---|---|---|---|---|---|---|---|
| ADAPTER | 4 | $5.78_{2.05}$ | $4.21_{1.16}$ | $0.69_{0.34}$ | $11.07_{2.07}$ | $3.58_{0.49}$ | $5.07_{1.22}$ |
| ADAPTER | 16 | $8.22_{6.21}$ | $2.59_{2.28}$ | $0.78_{0.42}$ | $8.42_{4.12}$ | $2.59_{1.34}$ | $4.52_{2.87}$ |
| ADAPTER | 32 | $4.65_{1.88}$ | $2.30_{2.71}$ | $0.82_{0.15}$ | $5.96_{7.43}$ | $2.97_{1.52}$ | $3.34_{2.74}$ |
| ADAPTER | 100 | $44.26_{1.22}$ | $10.59_{9.70}$ | $0.00_{0.00}$ | $25.26_{1.36}$ | $0.01_{0.02}$ | $16.02_{2.46}$ |
| ADAPTER | All | $71.38_{0.28}$ | $51.02_{1.23}$ | $43.26_{0.82}$ | $61.43_{0.91}$ | $42.46_{0.51}$ | $53.91_{0.75}$ |
| ADAPTERFUSION | 4 | $13.60_{1.29}$ | $7.20_{2.19}$ | $2.45_{0.37}$ | $16.24_{2.77}$ | $8.16_{1.00}$ | $9.53_{1.53}$ |
| ADAPTERFUSION | 16 | $13.27_{1.27}$ | $8.38_{0.99}$ | $2.17_{0.67}$ | $15.98_{2.41}$ | $7.63_{0.73}$ | $9.48_{1.21}$ |
| ADAPTERFUSION | 32 | $12.59_{1.91}$ | $6.41_{1.79}$ | $2.24_{0.25}$ | $13.67_{3.94}$ | $7.12_{1.00}$ | $8.40_{1.78}$ |
| ADAPTERFUSION | 100 | $8.03_{1.36}$ | $4.23_{2.75}$ | $1.77_{0.54}$ | $32.02_{4.30}$ | $5.07_{1.32}$ | $10.22_{2.05}$ |
| ADAPTERFUSION | All | $72.05_{0.12}$ | $49.63_{0.53}$ | $43.15_{0.38}$ | $60.68_{0.23}$ | $42.14_{0.46}$ | $53.53_{0.35}$ |
| SCALEARN | 4 | $5.56_{1.27}$ | $4.54_{0.57}$ | $1.12_{0.23}$ | $12.99_{0.26}$ | $3.95_{0.85}$ | $5.63_{0.64}$ |
| SCALEARN | 16 | $13.21_{0.74}$ | $8.90_{0.41}$ | $3.68_{0.16}$ | $18.30_{0.60}$ | $7.40_{0.53}$ | $10.30_{0.49}$ |
| SCALEARN | 32 | $16.64_{0.43}$ | $16.48_{0.74}$ | $7.23_{0.37}$ | $26.39_{0.34}$ | $11.11_{0.47}$ | $15.57_{0.47}$ |
| SCALEARN | 100 | $34.04_{1.36}$ | $26.31_{0.67}$ | $13.27_{1.06}$ | $30.68_{1.20}$ | $14.43_{0.39}$ | $23.75_{0.94}$ |
| SCALEARN | All | $72.36_{0.05}$ | $51.63_{0.61}$ | $44.06_{0.37}$ | $61.52_{0.11}$ | $42.81_{0.63}$ | $54.48_{0.35}$ |
| SCALEARNUNIFORM | 4 | $5.35_{1.09}$ | $4.32_{0.17}$ | $1.03_{0.20}$ | $13.24_{0.43}$ | $3.78_{0.64}$ | $5.54_{0.50}$ |
| SCALEARNUNIFORM | 16 | $13.65_{0.47}$ | $8.69_{0.59}$ | $3.64_{0.13}$ | $17.51_{1.23}$ | $7.59_{0.13}$ | $10.22_{0.51}$ |
| SCALEARNUNIFORM | 32 | $15.34_{0.52}$ | $16.72_{1.09}$ | $6.98_{0.34}$ | $25.75_{0.48}$ | $10.58_{0.19}$ | $15.07_{0.52}$ |
| SCALEARNUNIFORM | 100 | $33.40_{0.63}$ | $25.48_{0.71}$ | $13.43_{0.64}$ | $29.44_{0.78}$ | $14.92_{0.62}$ | $23.33_{0.68}$ |
| SCALEARNUNIFORM | All | $72.20_{0.14}$ | $50.08_{0.79}$ | $42.97_{0.70}$ | $60.62_{0.16}$ | $41.95_{0.60}$ | $53.56_{0.48}$ |
| SCALEARN++ | 4 | $5.42_{1.47}$ | $4.66_{0.45}$ | $1.16_{0.33}$ | $13.17_{0.17}$ | $3.62_{1.24}$ | $5.61_{0.73}$ |
| SCALEARN++ | 16 | $13.55_{0.71}$ | $8.89_{0.16}$ | $3.62_{0.09}$ | $18.62_{1.10}$ | $7.73_{0.28}$ | $10.48_{0.47}$ |
| SCALEARN++ | 32 | $16.27_{0.82}$ | $16.35_{1.62}$ | $7.27_{0.13}$ | $26.08_{0.51}$ | $10.70_{0.28}$ | $15.33_{0.67}$ |
| SCALEARN++ | 100 | $33.76_{0.49}$ | $25.83_{0.74}$ | $13.27_{0.66}$ | $30.11_{0.51}$ | $14.37_{0.61}$ | $23.47_{0.60}$ |
| SCALEARN++ | All | $72.38_{0.27}$ | $51.66_{0.27}$ | $44.23_{0.50}$ | $61.66_{0.13}$ | $42.21_{0.21}$ | $54.43_{0.28}$ |
| SCALEARNUNIFORM++ | 4 | $5.27_{1.18}$ | $4.37_{0.14}$ | $1.08_{0.09}$ | $13.20_{0.50}$ | $3.56_{1.15}$ | $5.50_{0.61}$ |
| SCALEARNUNIFORM++ | 16 | $13.47_{0.77}$ | $9.04_{0.58}$ | $3.60_{0.10}$ | $17.41_{0.59}$ | $7.50_{0.33}$ | $10.20_{0.47}$ |
| SCALEARNUNIFORM++ | 32 | $15.24_{0.35}$ | $16.75_{0.72}$ | $7.31_{0.28}$ | $26.23_{0.83}$ | $10.61_{0.27}$ | $15.23_{0.49}$ |
| SCALEARNUNIFORM++ | 100 | $39.22_{2.98}$ | $26.22_{0.74}$ | $13.76_{1.11}$ | $30.34_{0.63}$ | $14.56_{0.59}$ | $24.82_{1.21}$ |
| SCALEARNUNIFORM++ | All | $72.02_{0.32}$ | $50.78_{0.41}$ | $42.60_{0.85}$ | $60.82_{0.14}$ | $42.14_{0.72}$ | $53.67_{0.49}$ |

**Table 16:** Complete few-shot transfer learning results on the combination of all GLUE and SuperGLUE tasks with $k = \{4,16,32,100\}$ training samples for each target task using RoBERTa$_{\text{BASE}}$.

| Model | Samples | MNLI | QQP | QNLI | SST-2 | STS-B | MRPC | RTE | CoLA | ReCoRD | Multi | BoolQ | WiC | WSC | COPA | CB | Avg. |
|---|---|---|---|---|---|---|---|---|---|---|---|---|---|---|---|---|---|
| ADAPTER | 4 | $33.65_{1.39}$ | $63.27_{0.11}$ | $50.53_{0.04}$ | $50.92_{0.00}$ | $32.12_{9.28}$ | $68.38_{0.00}$ | $52.71_{0.00}$ | $2.93_{3.88}$ | $9.65_{2.79}$ | $24.92_{6.71}$ | $62.05_{0.27}$ | $49.44_{1.26}$ | $41.92_{12.04}$ | $50.20_{3.63}$ | $62.14_{8.12}$ | $43.66_{3.3}$ |
| ADAPTER | 16 | $34.78_{0.58}$ | $63.18_{0.00}$ | $50.46_{0.20}$ | $57.18_{1.23}$ | $55.53_{10.12}$ | $68.38_{0.00}$ | $53.72_{1.29}$ | $0.25_{0.56}$ | $13.82_{6.06}$ | $37.48_{8.48}$ | $62.17_{0.00}$ | $50.53_{1.18}$ | $42.50_{5.46}$ | $53.00_{5.48}$ | $69.29_{2.93}$ | $47.48_{2.9}$ |
| ADAPTER | 32 | $33.56_{0.66}$ | $63.18_{0.00}$ | $51.86_{0.33}$ | $70.46_{0.25}$ | $73.78_{1.30}$ | $68.38_{0.00}$ | $54.58_{1.81}$ | $0.00_{0.00}$ | $17.64_{12.76}$ | $38.55_{3.74}$ | $62.16_{0.03}$ | $52.26_{1.78}$ | $36.54_{0.00}$ | $51.20_{2.39}$ | $70.71_{1.60}$ | $49.66_{1.91}$ |
| ADAPTER | 100 | $40.71_{2.67}$ | $71.74_{0.50}$ | $58.77_{4.13}$ | $85.00_{2.25}$ | $82.51_{1.21}$ | $73.09_{0.27}$ | $56.17_{1.95}$ | $21.69_{3.94}$ | $37.69_{2.61}$ | $51.56_{3.89}$ | $61.51_{1.27}$ | $54.04_{1.01}$ | $50.38_{10.12}$ | $58.40_{6.18}$ | $73.93_{4.11}$ | $58.48_{3.07}$ |
| ADAPTER | All | $86.50_{0.33}$ | $90.18_{0.11}$ | $92.25_{0.19}$ | $93.65_{0.71}$ | $90.46_{0.31}$ | $86.64_{1.07}$ | $72.89_{2.54}$ | $58.28_{2.50}$ | $79.02_{0.62}$ | $72.89_{0.48}$ | $76.71_{1.38}$ | $65.58_{1.56}$ | $63.46_{0.00}$ | $70.20_{1.13}$ | $84.82_{3.18}$ | $78.88_{1.28}$ |
| ADAPTERFUSION | 4 | $33.03_{1.19}$ | $64.07_{12.23}$ | $51.46_{2.13}$ | $51.03_{0.26}$ | $77.54_{5.71}$ | $69.75_{2.15}$ | $53.72_{2.06}$ | $4.37_{4.11}$ | $17.13_{3.59}$ | $51.01_{19.87}$ | $62.17_{0.00}$ | $52.13_{3.45}$ | $38.85_{5.16}$ | $66.80_{8.01}$ | $60.00_{8.62}$ | $50.2_{5.17}$ |
| ADAPTERFUSION | 16 | $46.55_{3.27}$ | $73.63_{4.79}$ | $55.28_{7.92}$ | $58.05_{1.96}$ | $83.74_{0.93}$ | $71.23_{3.91}$ | $59.86_{4.17}$ | $4.01_{3.69}$ | $14.67_{6.72}$ | $53.06_{9.62}$ | $62.07_{0.44}$ | $54.29_{4.89}$ | $41.54_{6.88}$ | $68.80_{6.50}$ | $67.50_{10.37}$ | $54.28_{5.14}$ |
| ADAPTERFUSION | 32 | $43.42_{4.77}$ | $70.63_{6.80}$ | $70.48_{11.30}$ | $68.12_{6.56}$ | $84.17_{1.79}$ | $72.65_{5.98}$ | $60.79_{5.72}$ | $1.95_{4.37}$ | $12.40_{4.73}$ | $51.64_{8.97}$ | $62.08_{0.77}$ | $54.26_{2.19}$ | $38.27_{5.01}$ | $65.20_{6.83}$ | $71.79_{5.84}$ | $55.19_{5.64}$ |
| ADAPTERFUSION | 100 | $46.10_{2.95}$ | $76.77_{1.27}$ | $78.02_{4.45}$ | $83.88_{6.65}$ | $86.45_{1.57}$ | $77.50_{1.43}$ | $68.74_{4.29}$ | $22.17_{2.73}$ | $22.91_{5.96}$ | $50.84_{4.44}$ | $62.46_{0.84}$ | $58.06_{2.04}$ | $52.69_{6.71}$ | $71.40_{4.51}$ | $78.93_{6.11}$ | $62.46_{3.33}$ |
| ADAPTERFUSION | All | $86.52_{0.20}$ | $90.18_{0.11}$ | $92.35_{0.16}$ | $93.62_{0.69}$ | $90.46_{0.31}$ | $87.89_{1.00}$ | $78.84_{1.63}$ | $58.67_{1.42}$ | $78.66_{0.94}$ | $72.71_{0.71}$ | $76.63_{0.71}$ | $66.36_{1.34}$ | $63.46_{0.00}$ | $74.30_{3.02}$ | $83.57_{5.70}$ | $79.62_{1.2}$ |
| SCALEARN | 4 | $37.57_{1.54}$ | $73.10_{2.37}$ | $60.18_{2.87}$ | $52.78_{1.73}$ | $73.36_{4.84}$ | $72.35_{2.08}$ | $51.70_{2.26}$ | $4.26_{5.41}$ | $34.46_{6.05}$ | $48.41_{4.21}$ | $61.64_{0.33}$ | $52.48_{0.79}$ | $45.38_{6.74}$ | $75.00_{3.54}$ | $73.57_{4.96}$ | $54.42_{3.31}$ |
| SCALEARN | 16 | $51.76_{0.97}$ | $76.33_{0.35}$ | $57.83_{1.33}$ | $66.33_{3.67}$ | $83.03_{1.10}$ | $74.51_{1.29}$ | $60.36_{3.24}$ | $11.31_{5.26}$ | $33.47_{3.83}$ | $47.42_{6.90}$ | $61.70_{0.90}$ | $52.48_{0.98}$ | $46.92_{6.13}$ | $72.80_{4.55}$ | $83.21_{2.40}$ | $58.63_{2.86}$ |
| SCALEARN | 32 | $53.37_{1.50}$ | $76.13_{0.24}$ | $68.44_{0.66}$ | $77.73_{2.61}$ | $83.63_{1.29}$ | $75.05_{1.13}$ | $63.10_{1.72}$ | $12.98_{2.77}$ | $34.59_{1.07}$ | $47.05_{5.32}$ | $63.16_{0.46}$ | $54.36_{1.29}$ | $51.73_{7.46}$ | $72.20_{1.79}$ | $85.00_{0.98}$ | $61.24_{2.02}$ |
| SCALEARN | 100 | $60.40_{1.25}$ | $77.46_{0.43}$ | $73.91_{1.07}$ | $86.40_{1.20}$ | $87.19_{0.73}$ | $76.32_{1.26}$ | $71.05_{1.08}$ | $18.99_{2.22}$ | $40.31_{1.63}$ | $59.71_{0.32}$ | $63.46_{1.54}$ | $57.34_{2.04}$ | $57.69_{6.38}$ | $72.40_{0.58}$ | $90.00_{2.71}$ | $66.17_{1.90}$ |
| SCALEARN | All | $86.93_{0.03}$ | $89.78_{0.09}$ | $92.78_{0.03}$ | $94.65_{0.35}$ | $90.97_{0.09}$ | $88.21_{0.72}$ | $81.59_{1.69}$ | $59.32_{1.81}$ | $78.50_{0.48}$ | $72.67_{0.42}$ | $78.59_{0.28}$ | $66.76_{1.70}$ | $63.46_{0.51}$ | $80.60_{3.27}$ | $96.07_{1.41}$ | $81.39_{0.86}$ |
| SCALEARNUNIFORM | 4 | $39.28_{4.15}$ | $74.64_{0.41}$ | $55.21_{2.97}$ | $51.63_{1.86}$ | $63.58_{9.93}$ | $70.29_{0.40}$ | $52.71_{0.00}$ | $7.19_{2.78}$ | $16.65_{7.46}$ | $39.82_{10.73}$ | $60.57_{0.82}$ | $52.32_{0.84}$ | $48.85_{11.61}$ | $70.80_{1.92}$ | $70.00_{3.66}$ | $51.57_{3.97}$ |
| SCALEARNUNIFORM | 16 | $50.97_{0.92}$ | $76.20_{0.33}$ | $55.84_{1.97}$ | $62.91_{3.33}$ | $77.14_{2.38}$ | $73.24_{1.89}$ | $60.51_{1.95}$ | $13.85_{1.57}$ | $26.25_{6.79}$ | $49.67_{7.11}$ | $61.15_{0.95}$ | $52.19_{0.67}$ | $53.46_{8.03}$ | $71.40_{2.70}$ | $78.93_{7.93}$ | $57.58_{3.23}$ |
| SCALEARNUNIFORM | 32 | $48.73_{1.45}$ | $76.14_{0.10}$ | $66.42_{0.96}$ | $74.54_{2.25}$ | $81.22_{0.78}$ | $74.41_{1.76}$ | $63.97_{0.40}$ | $13.31_{4.10}$ | $35.15_{3.98}$ | $55.50_{4.23}$ | $61.60_{0.55}$ | $54.17_{1.21}$ | $46.92_{2.39}$ | $68.60_{2.79}$ | $78.21_{3.87}$ | $59.93_{2.06}$ |
| SCALEARNUNIFORM | 100 | $57.81_{0.74}$ | $77.10_{0.46}$ | $67.14_{1.42}$ | $81.35_{1.95}$ | $84.99_{0.37}$ | $75.34_{0.71}$ | $65.92_{2.61}$ | $17.58_{4.34}$ | $38.38_{3.95}$ | $59.14_{1.42}$ | $62.20_{0.69}$ | $55.45_{1.71}$ | $52.50_{6.92}$ | $71.40_{6.89}$ | $90.00_{2.04}$ | $63.75_{1.95}$ |
| SCALEARNUNIFORM | All | $87.02_{0.07}$ | $90.26_{0.10}$ | $92.01_{0.92}$ | $94.38_{0.30}$ | $90.16_{0.11}$ | $87.97_{0.99}$ | $80.87_{1.09}$ | $58.97_{0.83}$ | $80.05_{0.18}$ | $71.90_{0.29}$ | $76.42_{0.65}$ | $68.07_{0.77}$ | $63.22_{0.85}$ | $73.30_{2.16}$ | $93.93_{1.73}$ | $80.57_{0.74}$ |
| SCALEARN++ | 4 | $36.43_{0.84}$ | $71.99_{3.17}$ | $52.47_{2.56}$ | $50.96_{0.31}$ | $70.73_{3.49}$ | $69.66_{1.77}$ | $52.71_{0.00}$ | $5.38_{3.00}$ | $29.01_{4.05}$ | $29.56_{10.20}$ | $62.02_{0.35}$ | $50.38_{1.07}$ | $43.27_{4.14}$ | $72.20_{1.64}$ | $73.57_{4.45}$ | $51.36_{2.74}$ |
| SCALEARN++ | 16 | $49.88_{1.29}$ | $75.59_{0.93}$ | $55.82_{1.57}$ | $59.20_{2.02}$ | $80.68_{1.14}$ | $73.14_{1.44}$ | $58.84_{1.17}$ | $12.36_{4.88}$ | $25.45_{4.19}$ | $30.50_{17.83}$ | $60.12_{1.37}$ | $52.29_{1.69}$ | $47.88_{9.46}$ | $73.40_{3.21}$ | $79.64_{2.71}$ | $55.65_{3.66}$ |
| SCALEARN++ | 32 | $49.02_{2.00}$ | $74.91_{1.21}$ | $67.10_{0.91}$ | $74.29_{0.74}$ | $82.91_{0.76}$ | $73.24_{1.17}$ | $62.89_{1.34}$ | $11.30_{2.31}$ | $34.01_{1.05}$ | $25.76_{8.19}$ | $61.50_{1.55}$ | $53.48_{0.46}$ | $40.77_{3.82}$ | $74.80_{2.28}$ | $81.79_{3.43}$ | $57.85_{2.22}$ |
| SCALEARN++ | 100 | $58.52_{1.21}$ | $76.49_{0.86}$ | $68.02_{1.65}$ | $82.98_{0.84}$ | $86.45_{0.41}$ | $75.64_{0.37}$ | $68.59_{2.18}$ | $13.34_{3.89}$ | $39.05_{3.79}$ | $57.37_{3.88}$ | $60.63_{1.26}$ | $56.11_{0.73}$ | $59.04_{3.30}$ | $75.20_{1.30}$ | $91.79_{2.04}$ | $64.62_{1.85}$ |
| SCALEARN++ | All | $86.94_{0.01}$ | $89.56_{0.27}$ | $92.80_{0.08}$ | $94.04_{0.30}$ | $90.75_{0.16}$ | $88.21_{1.05}$ | $80.40_{0.90}$ | $59.65_{1.06}$ | $79.98_{0.21}$ | $71.16_{0.37}$ | $77.34_{0.37}$ | $67.43_{1.58}$ | $63.46_{0.00}$ | $79.90_{1.66}$ | $94.29_{2.35}$ | $81.06_{0.76}$ |
| SCALEARNUNIFORM++ | 4 | $38.43_{1.63}$ | $73.40_{1.00}$ | $54.48_{2.33}$ | $51.93_{1.31}$ | $58.37_{14.72}$ | $70.49_{0.48}$ | $52.71_{0.00}$ | $4.87_{2.38}$ | $17.96_{8.79}$ | $36.31_{10.99}$ | $60.91_{1.15}$ | $51.82_{1.24}$ | $49.23_{8.00}$ | $71.20_{2.86}$ | $71.79_{3.43}$ | $50.95_{4.02}$ |
| SCALEARNUNIFORM++ | 16 | $52.24_{1.62}$ | $75.52_{0.69}$ | $53.65_{0.74}$ | $62.04_{2.23}$ | $76.27_{2.73}$ | $74.51_{1.46}$ | $60.79_{1.65}$ | $8.95_{5.77}$ | $26.04_{6.66}$ | $51.37_{8.49}$ | $61.43_{1.58}$ | $52.07_{1.03}$ | $49.23_{5.90}$ | $69.60_{2.88}$ | $73.21_{5.92}$ | $56.46_{4.02}$ |
| SCALEARNUNIFORM++ | 32 | $49.59_{1.30}$ | $75.70_{0.35}$ | $65.38_{1.04}$ | $74.50_{0.66}$ | $80.99_{0.96}$ | $73.68_{0.82}$ | $62.17_{0.93}$ | $11.80_{3.59}$ | $35.20_{1.93}$ | $57.29_{4.93}$ | $62.14_{0.52}$ | $53.76_{1.44}$ | $51.15_{5.90}$ | $70.40_{1.34}$ | $80.36_{5.05}$ | $60.27_{2.05}$ |
| SCALEARNUNIFORM++ | 100 | $53.47_{1.29}$ | $76.22_{0.58}$ | $60.70_{0.57}$ | $82.94_{1.03}$ | $83.60_{0.58}$ | $73.82_{0.21}$ | $63.83_{3.23}$ | $16.43_{3.97}$ | $33.27_{3.38}$ | $59.32_{0.99}$ | $62.70_{0.31}$ | $55.49_{2.21}$ | $58.46_{7.67}$ | $65.20_{2.86}$ | $86.07_{3.87}$ | $62.10_{3.05}$ |
| SCALEARNUNIFORM++ | All | $86.82_{0.17}$ | $90.16_{0.34}$ | $92.35_{0.31}$ | $94.61_{0.11}$ | $90.32_{0.14}$ | $87.97_{0.86}$ | $80.79_{0.96}$ | $59.33_{0.90}$ | $79.80_{0.56}$ | $72.76_{0.51}$ | $76.22_{0.69}$ | $67.95_{1.04}$ | $61.78_{1.98}$ | $74.20_{1.75}$ | $93.21_{0.75}$ | $80.55_{0.74}$ |

**Table 17:** Complete few-shot transfer learning results on GLUE with $k = \{4,16,32,100\}$ training samples for each target task using RoBERTa$_{\text{LARGE}}$.

| Model | Samples | MNLI | QQP | QNLI | SST-2 | STS-B | MRPC | RTE | CoLA | Avg. |
|---|---|---|---|---|---|---|---|---|---|---|
| ADAPTER | 4 | $34.09_{0.48}$ | $62.00_{2.54}$ | $50.46_{1.12}$ | $50.92_{0.00}$ | $10.02_{2.34}$ | $68.33_{0.11}$ | $51.48_{2.74}$ | $3.47_{3.01}$ | $41.35_{1.54}$ |
| ADAPTER | 16 | $35.12_{1.00}$ | $63.11_{0.18}$ | $49.59_{0.24}$ | $59.38_{3.42}$ | $12.41_{5.51}$ | $68.38_{0.00}$ | $52.64_{1.06}$ | $2.55_{3.07}$ | $42.90_{1.81}$ |
| ADAPTER | 32 | $34.05_{0.94}$ | $63.88_{1.40}$ | $51.30_{0.98}$ | $74.70_{2.59}$ | $27.16_{13.89}$ | $68.77_{0.71}$ | $51.62_{1.75}$ | $7.47_{10.36}$ | $47.37_{4.08}$ |
| ADAPTER | 100 | $41.39_{2.59}$ | $71.35_{0.81}$ | $53.75_{1.18}$ | $83.67_{2.22}$ | $76.84_{4.07}$ | $69.07_{1.48}$ | $56.97_{2.30}$ | $30.96_{5.72}$ | $60.50_{2.55}$ |
| ADAPTER | All | $89.62_{0.18}$ | $89.87_{0.67}$ | $94.13_{0.06}$ | $95.24_{0.08}$ | $91.81_{0.29}$ | $87.82_{2.11}$ | $81.23_{2.92}$ | $64.07_{1.97}$ | $86.72_{1.04}$ |
| ADAPTERFUSION | 4 | $39.26_{6.48}$ | $79.28_{0.71}$ | $65.13_{11.67}$ | $51.03_{0.23}$ | $76.40_{12.07}$ | $69.95_{2.76}$ | $54.08_{3.07}$ | $4.93_{1.85}$ | $55.01_{4.85}$ |
| ADAPTERFUSION | 16 | $49.94_{8.89}$ | $80.37_{0.13}$ | $78.85_{3.67}$ | $56.65_{3.82}$ | $83.96_{0.85}$ | $77.50_{1.62}$ | $70.47_{4.04}$ | $16.08_{3.34}$ | $64.23_{3.29}$ |
| ADAPTERFUSION | 32 | $56.12_{10.53}$ | $80.01_{0.25}$ | $80.55_{1.30}$ | $75.29_{7.71}$ | $85.36_{0.87}$ | $77.11_{4.44}$ | $78.70_{3.54}$ | $6.77_{8.63}$ | $67.49_{4.66}$ |
| ADAPTERFUSION | 100 | $60.84_{13.22}$ | $78.86_{3.07}$ | $85.09_{0.80}$ | $85.44_{1.87}$ | $88.09_{0.39}$ | $81.86_{1.63}$ | $84.40_{2.62}$ | $34.69_{2.72}$ | $74.91_{3.29}$ |
| ADAPTERFUSION | All | $89.57_{0.17}$ | $90.88_{0.06}$ | $94.15_{0.04}$ | $95.87_{0.00}$ | $91.86_{0.15}$ | $88.97_{0.78}$ | $85.70_{1.13}$ | $66.39_{1.83}$ | $87.93_{0.52}$ |
| SCALEARN | 4 | $45.65_{4.75}$ | $79.59_{0.24}$ | $66.97_{3.83}$ | $52.06_{1.12}$ | $81.94_{2.17}$ | $72.06_{2.37}$ | $52.71_{0.00}$ | $3.14_{1.31}$ | $56.77_{1.97}$ |
| SCALEARN | 16 | $57.54_{1.50}$ | $80.04_{0.58}$ | $77.24_{0.85}$ | $62.59_{2.91}$ | $85.08_{1.83}$ | $76.42_{2.70}$ | $69.75_{2.56}$ | $4.23_{3.10}$ | $64.11_{2.00}$ |
| SCALEARN | 32 | $60.95_{1.59}$ | $79.95_{0.34}$ | $77.72_{0.94}$ | $74.13_{1.58}$ | $88.50_{0.27}$ | $76.91_{1.69}$ | $77.91_{1.83}$ | $5.14_{2.00}$ | $67.65_{1.28}$ |
| SCALEARN | 100 | $69.18_{1.32}$ | $80.80_{0.21}$ | $83.64_{2.26}$ | $84.20_{0.98}$ | $89.25_{0.40}$ | $77.60_{1.78}$ | $82.96_{0.93}$ | $10.80_{1.43}$ | $72.30_{1.17}$ |
| SCALEARN | All | $90.09_{0.09}$ | $90.51_{0.26}$ | $94.18_{0.03}$ | $95.41_{0.16}$ | $92.32_{0.15}$ | $88.09_{0.82}$ | $87.08_{0.54}$ | $65.40_{2.62}$ | $87.91_{0.55}$ |
| SCALEARNUNIFORM | 4 | $45.73_{5.20}$ | $79.74_{0.34}$ | $67.95_{3.57}$ | $52.41_{1.39}$ | $81.59_{1.89}$ | $72.21_{2.26}$ | $52.71_{0.00}$ | $3.25_{1.02}$ | $56.95_{1.96}$ |
| SCALEARNUNIFORM | 16 | $57.61_{1.01}$ | $79.81_{0.31}$ | $74.55_{1.75}$ | $57.43_{2.44}$ | $85.32_{0.85}$ | $75.34_{1.10}$ | $68.81_{1.21}$ | $1.92_{2.57}$ | $62.60_{1.41}$ |
| SCALEARNUNIFORM | 32 | $58.86_{1.71}$ | $80.06_{0.14}$ | $75.86_{1.12}$ | $73.60_{1.06}$ | $86.61_{0.33}$ | $74.66_{1.16}$ | $77.91_{1.12}$ | $5.66_{4.15}$ | $66.65_{1.35}$ |
| SCALEARNUNIFORM | 100 | $63.51_{1.39}$ | $80.34_{0.21}$ | $74.98_{2.50}$ | $81.44_{1.48}$ | $87.36_{0.24}$ | $76.47_{1.26}$ | $81.37_{1.87}$ | $14.98_{1.27}$ | $70.06_{1.28}$ |
| SCALEARNUNIFORM | All | $90.11_{0.04}$ | $90.05_{0.28}$ | $94.23_{0.08}$ | $95.41_{0.16}$ | $92.11_{0.06}$ | $88.63_{1.72}$ | $84.40_{3.93}$ | $66.98_{0.58}$ | $87.74_{0.86}$ |
| SCALEARN++ | 4 | $44.54_{4.16}$ | $79.58_{0.41}$ | $66.90_{2.38}$ | $51.70_{0.75}$ | $80.80_{3.59}$ | $71.86_{1.54}$ | $52.71_{0.00}$ | $3.78_{0.89}$ | $56.48_{1.72}$ |
| SCALEARN++ | 16 | $56.71_{1.57}$ | $80.11_{0.37}$ | $73.80_{1.36}$ | $60.16_{3.41}$ | $85.17_{1.14}$ | $75.20_{3.15}$ | $69.82_{2.07}$ | $2.85_{3.64}$ | $62.98_{2.09}$ |
| SCALEARN++ | 32 | $58.87_{1.51}$ | $79.09_{0.49}$ | $75.92_{0.89}$ | $73.12_{3.27}$ | $87.45_{0.32}$ | $75.69_{1.18}$ | $77.33_{0.90}$ | $5.47_{4.01}$ | $66.61_{1.57}$ |
| SCALEARN++ | 100 | $65.07_{1.14}$ | $80.23_{0.33}$ | $78.82_{0.81}$ | $82.00_{1.89}$ | $88.01_{0.42}$ | $76.62_{1.16}$ | $81.81_{2.60}$ | $12.11_{2.78}$ | $70.58_{1.44}$ |
| SCALEARN++ | All | $90.31_{0.10}$ | $90.59_{0.03}$ | $94.05_{0.03}$ | $95.93_{0.24}$ | $92.48_{0.15}$ | $88.48_{1.26}$ | $86.28_{1.05}$ | $67.13_{0.59}$ | $88.16_{0.43}$ |
| SCALEARNUNIFORM++ | 4 | $44.48_{4.38}$ | $79.42_{0.58}$ | $66.59_{4.06}$ | $51.46_{0.57}$ | $82.15_{1.17}$ | $73.22_{1.12}$ | $52.71_{0.00}$ | $2.34_{0.52}$ | $56.55_{1.55}$ |
| SCALEARNUNIFORM++ | 16 | $56.63_{1.44}$ | $79.53_{0.45}$ | $72.95_{2.27}$ | $56.94_{1.01}$ | $85.14_{0.66}$ | $75.21_{2.09}$ | $68.86_{1.85}$ | $0.80_{2.46}$ | $62.06_{1.53}$ |
| SCALEARNUNIFORM++ | 32 | $57.68_{3.31}$ | $79.47_{0.42}$ | $73.78_{1.89}$ | $75.15_{0.96}$ | $86.64_{0.56}$ | $76.65_{1.49}$ | $78.34_{0.66}$ | $1.78_{2.84}$ | $66.19_{1.52}$ |
| SCALEARNUNIFORM++ | 100 | $56.72_{1.49}$ | $78.91_{0.82}$ | $66.11_{2.51}$ | $83.75_{0.58}$ | $85.53_{0.82}$ | $74.33_{2.49}$ | $81.68_{2.51}$ | $20.84_{3.14}$ | $68.48_{1.79}$ |
| SCALEARNUNIFORM++ | All | $90.08_{0.01}$ | $90.49_{0.02}$ | $94.12_{0.16}$ | $95.18_{0.16}$ | $92.12_{0.09}$ | $90.05_{0.54}$ | $84.98_{1.32}$ | $64.97_{0.85}$ | $87.75_{0.39}$ |

**Table 18:** Complete few-shot transfer learning results on SuperGLUE with $k = \{4,16,32,100\}$ training samples for each target task using RoBERTa$_{\text{LARGE}}$.

| Model | Samples | ReCoRD | Multi | BoolQ | WiC | WSC | COPA | CB | RTE | Avg. |
|---|---|---|---|---|---|---|---|---|---|---|
| ADAPTER | 4 | $15.58_{3.93}$ | $31.78_{15.80}$ | $61.83_{0.58}$ | $49.75_{0.56}$ | $50.38_{8.93}$ | $49.60_{5.59}$ | $53.93_{4.96}$ | $51.48_{2.74}$ | $45.54_{5.39}$ |
| ADAPTER | 16 | $17.42_{7.21}$ | $40.46_{3.08}$ | $61.64_{0.54}$ | $51.38_{1.37}$ | $54.04_{2.30}$ | $53.60_{5.46}$ | $61.07_{3.19}$ | $52.64_{1.06}$ | $49.03_{3.03}$ |
| ADAPTER | 32 | $22.04_{14.70}$ | $41.11_{5.21}$ | $62.17_{0.01}$ | $52.88_{1.91}$ | $47.69_{3.76}$ | $66.20_{7.60}$ | $67.50_{2.33}$ | $51.62_{1.75}$ | $51.40_{4.66}$ |
| ADAPTER | 100 | $31.01_{19.22}$ | $51.93_{4.94}$ | $62.17_{0.00}$ | $55.96_{2.03}$ | $52.88_{5.81}$ | $65.20_{13.86}$ | $82.14_{5.65}$ | $56.97_{2.30}$ | $57.28_{6.73}$ |
| ADAPTER | All | $88.52_{0.09}$ | $80.73_{0.69}$ | $82.36_{0.72}$ | $69.16_{1.31}$ | $63.25_{0.64}$ | $71.90_{13.63}$ | $92.68_{1.78}$ | $81.23_{2.92}$ | $78.73_{2.72}$ |
| ADAPTERFUSION | 4 | $19.21_{4.17}$ | $24.07_{20.35}$ | $61.77_{0.18}$ | $50.63_{1.49}$ | $43.27_{12.03}$ | $57.00_{7.42}$ | $61.43_{11.75}$ | $52.71_{0.00}$ | $46.26_{7.17}$ |
| ADAPTERFUSION | 16 | $14.28_{5.34}$ | $28.09_{4.31}$ | $61.51_{0.35}$ | $51.10_{3.13}$ | $47.31_{9.26}$ | $66.20_{12.44}$ | $77.86_{4.48}$ | $53.21_{1.37}$ | $49.95_{5.09}$ |
| ADAPTERFUSION | 32 | $18.82_{11.93}$ | $37.68_{10.93}$ | $64.97_{3.64}$ | $52.82_{1.39}$ | $44.42_{3.36}$ | $62.40_{10.24}$ | $78.21_{4.45}$ | $58.05_{4.21}$ | $52.17_{6.27}$ |
| ADAPTERFUSION | 100 | $55.42_{1.38}$ | $59.98_{0.03}$ | $71.06_{2.02}$ | $56.02_{1.25}$ | $55.58_{5.33}$ | $76.40_{13.22}$ | $84.64_{4.11}$ | $57.62_{2.71}$ | $64.59_{3.75}$ |
| ADAPTERFUSION | All | $89.21_{0.17}$ | $80.52_{0.24}$ | $82.21_{0.30}$ | $69.09_{1.68}$ | $63.46_{0.68}$ | $81.20_{16.30}$ | $95.71_{0.98}$ | $86.06_{1.07}$ | $80.93_{2.65}$ |
| SCALEARN | 4 | $32.72_{3.66}$ | $58.49_{1.59}$ | $61.90_{0.30}$ | $51.66_{1.61}$ | $55.58_{8.66}$ | $71.00_{6.36}$ | $77.50_{2.04}$ | $52.71_{0.00}$ | $57.69_{3.03}$ |
| SCALEARN | 16 | $36.71_{3.11}$ | $53.37_{3.76}$ | $61.82_{0.56}$ | $53.51_{1.09}$ | $50.19_{5.54}$ | $77.40_{7.13}$ | $77.86_{4.11}$ | $55.88_{3.01}$ | $58.34_{3.54}$ |
| SCALEARN | 32 | $36.72_{3.37}$ | $57.30_{4.03}$ | $61.47_{0.75}$ | $53.26_{2.28}$ | $49.04_{5.73}$ | $80.60_{3.05}$ | $80.00_{1.49}$ | $57.62_{5.12}$ | $59.50_{3.23}$ |
| SCALEARN | 100 | $54.21_{12.46}$ | $59.79_{0.30}$ | $68.78_{3.12}$ | $51.88_{1.84}$ | $57.12_{1.87}$ | $81.80_{5.97}$ | $85.00_{2.04}$ | $65.34_{3.44}$ | $65.49_{3.88}$ |
| SCALEARN | All | $87.85_{0.01}$ | $78.40_{0.70}$ | $80.29_{2.52}$ | $68.56_{1.68}$ | $62.98_{0.68}$ | $85.40_{3.78}$ | $92.86_{1.79}$ | $84.91_{0.59}$ | $80.16_{1.47}$ |
| SCALEARNUNIFORM | 4 | $33.12_{5.16}$ | $59.47_{0.94}$ | $61.51_{1.01}$ | $50.91_{1.64}$ | $63.46_{0.00}$ | $68.00_{3.08}$ | $78.93_{2.33}$ | $52.71_{0.00}$ | $58.51_{1.77}$ |
| SCALEARNUNIFORM | 16 | $32.75_{2.12}$ | $54.65_{7.16}$ | $62.11_{0.15}$ | $52.12_{3.49}$ | $51.26_{0.85}$ | $76.20_{2.65}$ | $81.79_{2.65}$ | $54.44_{3.40}$ | $57.76_{2.71}$ |
| SCALEARNUNIFORM | 32 | $35.30_{3.67}$ | $58.22_{3.85}$ | $61.76_{0.61}$ | $54.67_{2.40}$ | $51.92_{6.04}$ | $76.40_{2.97}$ | $80.00_{2.93}$ | $58.92_{5.58}$ | $59.65_{3.51}$ |
| SCALEARNUNIFORM | 100 | $41.50_{5.85}$ | $60.01_{0.10}$ | $61.96_{0.76}$ | $51.85_{1.21}$ | $58.27_{1.75}$ | $72.40_{5.37}$ | $85.00_{2.04}$ | $60.65_{1.05}$ | $61.45_{2.27}$ |
| SCALEARNUNIFORM | All | $88.85_{0.22}$ | $80.42_{0.06}$ | $81.85_{0.21}$ | $69.91_{1.15}$ | $61.54_{0.00}$ | $82.00_{3.08}$ | $90.00_{1.60}$ | $84.04_{1.66}$ | $79.83_{1.00}$ |
| SCALEARN++ | 4 | $33.87_{1.90}$ | $56.11_{3.47}$ | $61.75_{0.21}$ | $51.32_{1.66}$ | $60.58_{3.96}$ | $68.00_{6.04}$ | $78.21_{2.33}$ | $52.71_{0.00}$ | $57.82_{2.45}$ |
| SCALEARN++ | 16 | $35.36_{0.48}$ | $53.71_{5.41}$ | $61.93_{0.39}$ | $52.79_{0.17}$ | $50.77_{2.99}$ | $71.40_{3.78}$ | $80.00_{4.07}$ | $55.23_{2.75}$ | $57.65_{2.51}$ |
| SCALEARN++ | 32 | $38.87_{1.77}$ | $59.95_{0.00}$ | $61.94_{0.81}$ | $54.61_{2.06}$ | $46.92_{3.22}$ | $78.60_{2.30}$ | $79.64_{2.71}$ | $53.14_{3.49}$ | $59.21_{2.05}$ |
| SCALEARN++ | 100 | $43.15_{4.43}$ | $59.95_{0.00}$ | $63.36_{0.98}$ | $52.01_{0.73}$ | $57.12_{3.23}$ | $75.20_{4.15}$ | $86.79_{2.04}$ | $62.24_{2.68}$ | $62.48_{2.28}$ |
| SCALEARN++ | All | $88.28_{0.23}$ | $80.76_{0.58}$ | $83.08_{0.31}$ | $69.59_{1.89}$ | $62.98_{0.68}$ | $87.80_{1.10}$ | $91.07_{1.79}$ | $85.70_{0.32}$ | $81.16_{0.86}$ |
| SCALEARNUNIFORM++ | 4 | $33.87_{1.90}$ | $56.11_{3.47}$ | $61.75_{0.21}$ | $51.32_{1.66}$ | $60.58_{3.96}$ | $68.00_{6.04}$ | $78.21_{2.33}$ | $52.71_{0.00}$ | $57.82_{2.45}$ |
| SCALEARNUNIFORM++ | 16 | $35.36_{0.48}$ | $53.71_{5.41}$ | $61.93_{0.39}$ | $52.79_{0.17}$ | $50.77_{2.99}$ | $71.40_{3.78}$ | $80.00_{4.07}$ | $55.23_{2.75}$ | $57.65_{2.51}$ |
| SCALEARNUNIFORM++ | 32 | $38.87_{1.77}$ | $59.95_{0.00}$ | $61.94_{0.81}$ | $54.61_{2.06}$ | $46.92_{3.22}$ | $78.60_{2.30}$ | $79.64_{2.71}$ | $53.14_{3.49}$ | $59.21_{2.05}$ |
| SCALEARNUNIFORM++ | 100 | $43.15_{4.43}$ | $59.95_{0.00}$ | $63.36_{0.98}$ | $52.01_{0.73}$ | $57.12_{3.23}$ | $75.20_{4.15}$ | $86.79_{2.04}$ | $62.24_{2.68}$ | $62.48_{2.28}$ |
| SCALEARNUNIFORM++ | All | $88.85_{0.22}$ | $80.70_{0.04}$ | $82.13_{0.21}$ | $70.19_{0.26}$ | $62.98_{0.68}$ | $83.60_{2.88}$ | $91.07_{2.82}$ | $84.84_{1.02}$ | $80.54_{1.02}$ |

**Table 19:** Complete few-shot transfer learning results on HumSet with $k = \{4,16,32,100\}$ training samples for each target task using XLM-R$_{\text{LARGE}}$.

| Model | Samples | Sectors | Pillars 1D | Subpillars 1D | Pillars 2D | Subpillars 2D | Avg. |
|---|---|---|---|---|---|---|---|
| ADAPTER | 4 | $4.80_{0.60}$ | $4.33_{0.18}$ | $0.60_{0.08}$ | $10.87_{1.72}$ | $2.56_{0.56}$ | $4.63_{0.63}$ |
| ADAPTER | 16 | $7.12_{2.11}$ | $1.35_{1.85}$ | $0.45_{0.32}$ | $11.08_{0.59}$ | $2.82_{0.82}$ | $4.56_{1.14}$ |
| ADAPTER | 32 | $6.60_{3.21}$ | $0.58_{0.54}$ | $0.52_{0.24}$ | $11.82_{1.44}$ | $2.40_{0.92}$ | $4.39_{1.27}$ |
| ADAPTER | 100 | $24.66_{13.33}$ | $12.38_{3.57}$ | $0.00_{0.00}$ | $16.21_{1.14}$ | $3.13_{2.91}$ | $11.27_{4.19}$ |
| ADAPTER | All | $72.29_{0.59}$ | $49.31_{1.27}$ | $45.25_{0.03}$ | $62.58_{0.67}$ | $44.36_{0.66}$ | $54.76_{0.65}$ |
| ADAPTERFUSION | 4 | $12.43_{2.84}$ | $7.58_{0.95}$ | $2.11_{0.12}$ | $14.59_{0.57}$ | $7.10_{1.13}$ | $8.76_{1.12}$ |
| ADAPTERFUSION | 16 | $11.06_{2.41}$ | $6.49_{2.35}$ | $2.30_{0.26}$ | $13.08_{1.04}$ | $6.33_{1.79}$ | $7.85_{1.57}$ |
| ADAPTERFUSION | 32 | $11.90_{3.19}$ | $6.40_{2.61}$ | $2.50_{0.60}$ | $13.23_{0.90}$ | $6.16_{1.54}$ | $8.04_{1.77}$ |
| ADAPTERFUSION | 100 | $31.92_{5.40}$ | $17.74_{2.59}$ | $1.94_{0.42}$ | $31.44_{2.30}$ | $8.08_{3.78}$ | $18.22_{2.90}$ |
| ADAPTERFUSION | All | $72.53_{0.45}$ | $51.33_{0.23}$ | $43.75_{0.52}$ | $62.31_{0.25}$ | $42.78_{2.11}$ | $54.54_{0.71}$ |
| SCALEARN | 4 | $5.52_{0.93}$ | $4.94_{0.21}$ | $1.30_{0.26}$ | $13.59_{0.46}$ | $3.81_{0.90}$ | $5.83_{0.55}$ |
| SCALEARN | 16 | $12.05_{0.80}$ | $7.78_{0.31}$ | $3.24_{0.09}$ | $20.10_{1.33}$ | $6.19_{0.30}$ | $9.87_{0.57}$ |
| SCALEARN | 32 | $16.34_{0.63}$ | $15.74_{0.95}$ | $6.54_{0.29}$ | $24.92_{0.40}$ | $10.54_{0.33}$ | $14.82_{0.52}$ |
| SCALEARN | 100 | $24.60_{0.97}$ | $24.36_{1.80}$ | $11.37_{0.40}$ | $34.26_{2.54}$ | $15.63_{0.64}$ | $22.05_{1.27}$ |
| SCALEARN | All | $73.32_{0.08}$ | $53.94_{0.13}$ | $44.14_{0.75}$ | $63.89_{0.16}$ | $44.75_{0.47}$ | $56.01_{0.32}$ |
| SCALEARNUNIFORM | 4 | $4.92_{0.61}$ | $4.84_{0.26}$ | $1.25_{0.30}$ | $13.05_{0.48}$ | $3.41_{0.11}$ | $5.49_{0.35}$ |
| SCALEARNUNIFORM | 16 | $11.58_{0.45}$ | $7.78_{0.53}$ | $3.15_{0.19}$ | $20.11_{0.32}$ | $5.79_{0.16}$ | $9.68_{0.33}$ |
| SCALEARNUNIFORM | 32 | $15.45_{0.00}$ | $15.48_{0.64}$ | $6.54_{0.52}$ | $24.22_{0.16}$ | $9.70_{0.17}$ | $14.28_{0.30}$ |
| SCALEARNUNIFORM | 100 | $21.91_{0.00}$ | $23.31_{2.49}$ | $10.60_{0.22}$ | $36.44_{2.05}$ | $15.27_{0.13}$ | $21.51_{0.98}$ |
| SCALEARNUNIFORM | All | $72.56_{0.20}$ | $50.59_{0.10}$ | $44.62_{0.00}$ | $62.66_{0.00}$ | $45.16_{0.00}$ | $55.12_{0.06}$ |
| SCALEARN++ | 4 | $4.90_{0.40}$ | $4.95_{0.20}$ | $1.45_{0.26}$ | $13.48_{0.52}$ | $3.37_{0.50}$ | $5.63_{0.38}$ |
| SCALEARN++ | 16 | $12.45_{0.65}$ | $8.47_{0.77}$ | $3.29_{0.13}$ | $21.01_{1.12}$ | $6.55_{0.37}$ | $10.35_{0.61}$ |
| SCALEARN++ | 32 | $16.61_{0.57}$ | $15.80_{1.00}$ | $6.71_{0.29}$ | $24.76_{0.32}$ | $10.31_{0.36}$ | $14.84_{0.51}$ |
| SCALEARN++ | 100 | $24.44_{0.95}$ | $23.95_{0.40}$ | $11.36_{0.65}$ | $35.18_{1.28}$ | $15.77_{0.77}$ | $22.14_{0.81}$ |
| SCALEARN++ | All | $73.18_{0.04}$ | $51.41_{0.36}$ | $44.10_{0.09}$ | $63.37_{0.02}$ | $45.43_{0.24}$ | $55.50_{0.15}$ |
| SCALEARNUNIFORM++ | 4 | $4.92_{0.61}$ | $4.84_{0.26}$ | $1.25_{0.30}$ | $13.05_{0.48}$ | $3.41_{0.11}$ | $5.49_{0.35}$ |
| SCALEARNUNIFORM++ | 16 | $11.58_{0.45}$ | $7.78_{0.53}$ | $3.15_{0.19}$ | $20.11_{0.32}$ | $5.79_{0.16}$ | $9.68_{0.33}$ |
| SCALEARNUNIFORM++ | 32 | $15.45_{0.00}$ | $15.48_{0.64}$ | $6.54_{0.52}$ | $24.22_{0.16}$ | $9.70_{0.17}$ | $14.28_{0.30}$ |
| SCALEARNUNIFORM++ | 100 | $21.91_{0.00}$ | $23.31_{2.49}$ | $10.60_{0.22}$ | $36.44_{2.05}$ | $15.27_{0.13}$ | $21.51_{0.98}$ |
| SCALEARNUNIFORM++ | All | $73.02_{0.20}$ | $50.84_{0.30}$ | $44.88_{0.39}$ | $62.87_{0.01}$ | $44.45_{0.02}$ | $55.21_{0.18}$ |

**Table 20:** Complete few-shot transfer learning results on the combination of all GLUE and SuperGLUE tasks with $k = \{4, 16, 32, 100\}$ training samples for each target task using RoBERTa$_{LARGE}$.

| Model | Samples | MNLI | QQP | QNLI | SST-2 | STS-B | MRPC | RTE | CoLA | ReCoRD | Multi | BoolQ | WiC | WSC | COPA | CB | Avg. |
|---|---|---|---|---|---|---|---|---|---|---|---|---|---|---|---|---|---|
| ADAPTER | 4 | $34.09_{0.48}$ | $62.00_{2.54}$ | $50.46_{1.12}$ | $50.92_{0.00}$ | $10.02_{2.34}$ | $68.33_{0.11}$ | $51.48_{2.74}$ | $3.47_{3.01}$ | $15.58_{5.93}$ | $31.78_{15.80}$ | $61.83_{0.58}$ | $49.75_{0.56}$ | $50.38_{8.93}$ | $49.60_{5.59}$ | $53.93_{4.96}$ | $42.91_{3.51}$ |
| ADAPTER | 16 | $35.12_{1.00}$ | $63.11_{0.18}$ | $49.59_{0.24}$ | $59.38_{3.42}$ | $12.41_{5.51}$ | $68.38_{0.00}$ | $52.64_{1.06}$ | $2.55_{3.07}$ | $17.42_{7.21}$ | $40.46_{3.08}$ | $61.64_{0.54}$ | $51.38_{1.37}$ | $54.04_{2.30}$ | $53.60_{5.46}$ | $61.07_{3.19}$ | $45.52_{2.51}$ |
| ADAPTER | 32 | $34.05_{0.94}$ | $63.88_{1.40}$ | $51.30_{0.98}$ | $74.70_{2.59}$ | $27.16_{13.89}$ | $68.77_{0.71}$ | $51.62_{1.75}$ | $7.47_{10.36}$ | $22.04_{14.70}$ | $41.11_{5.21}$ | $62.17_{0.01}$ | $52.88_{1.91}$ | $47.69_{3.76}$ | $66.20_{7.60}$ | $67.50_{2.33}$ | $49.24_{4.54}$ |
| ADAPTER | 100 | $41.39_{2.59}$ | $71.35_{0.81}$ | $53.75_{1.18}$ | $83.67_{2.22}$ | $76.84_{4.07}$ | $69.07_{1.48}$ | $56.97_{2.30}$ | $30.96_{5.72}$ | $31.01_{19.22}$ | $51.93_{4.94}$ | $62.17_{0.00}$ | $55.96_{2.03}$ | $52.88_{5.81}$ | $65.20_{13.86}$ | $82.14_{5.65}$ | $59.02_{4.79}$ |
| ADAPTER | All | $89.62_{0.18}$ | $89.87_{0.67}$ | $94.13_{0.06}$ | $95.24_{0.08}$ | $91.81_{0.29}$ | $87.82_{2.11}$ | $81.23_{2.92}$ | $64.07_{1.97}$ | $88.52_{0.09}$ | $80.73_{0.69}$ | $82.36_{0.72}$ | $69.16_{1.31}$ | $62.79_{1.57}$ | $71.90_{13.63}$ | $92.68_{1.78}$ | $82.80_{1.87}$ |
| ADAPTERFUSION | 4 | $39.28_{3.90}$ | $67.18_{6.80}$ | $51.03_{2.67}$ | $51.95_{1.79}$ | $80.27_{0.95}$ | $68.38_{0.00}$ | $52.71_{0.00}$ | $8.24_{3.12}$ | $22.33_{10.68}$ | $20.43_{4.01}$ | $61.13_{0.91}$ | $52.14_{0.65}$ | $58.65_{8.33}$ | $79.67_{6.03}$ | $73.81_{2.73}$ | $52.48_{3.50}$ |
| ADAPTERFUSION | 16 | $49.24_{4.09}$ | $78.34_{0.61}$ | $78.75_{0.80}$ | $58.14_{5.01}$ | $83.54_{1.17}$ | $74.59_{6.29}$ | $70.40_{7.61}$ | $19.35_{5.34}$ | $21.80_{15.81}$ | $43.89_{7.61}$ | $62.46_{5.01}$ | $53.61_{2.00}$ | $56.41_{12.21}$ | $80.67_{6.03}$ | $79.76_{3.72}$ | $60.73_{5.55}$ |
| ADAPTERFUSION | 32 | $52.62_{9.97}$ | $78.71_{1.16}$ | $79.27_{4.25}$ | $79.32_{2.61}$ | $84.15_{1.70}$ | $75.98_{2.34}$ | $76.77_{1.85}$ | $12.63_{3.82}$ | $30.11_{7.06}$ | $47.13_{4.01}$ | $66.61_{5.92}$ | $60.61_{2.33}$ | $57.69_{6.99}$ | $84.33_{1.15}$ | $82.74_{2.06}$ | $64.58_{4.01}$ |
| ADAPTERFUSION | 100 | $62.92_{6.87}$ | $78.32_{4.42}$ | $85.14_{1.00}$ | $86.81_{1.35}$ | $87.66_{0.39}$ | $79.98_{2.70}$ | $81.11_{4.57}$ | $35.73_{1.51}$ | $44.01_{4.57}$ | $66.08_{1.82}$ | $73.29_{0.89}$ | $59.30_{2.65}$ | $59.94_{0.56}$ | $83.00_{7.21}$ | $83.93_{3.57}$ | $71.15_{2.87}$ |
| ADAPTERFUSION | All | $89.79_{0.12}$ | $90.83_{0.27}$ | $94.14_{0.05}$ | $95.64_{0.00}$ | $92.08_{0.14}$ | $89.12_{0.22}$ | $85.85_{2.48}$ | $66.52_{1.31}$ | $89.26_{0.00}$ | $79.25_{0.80}$ | $82.40_{0.54}$ | $69.50_{1.12}$ | $62.69_{1.72}$ | $88.60_{3.36}$ | $90.36_{2.99}$ | $84.40_{1.01}$ |
| SCALEARN | 4 | $48.58_{1.93}$ | $71.81_{1.62}$ | $56.89_{0.79}$ | $52.75_{2.61}$ | $57.27_{7.57}$ | $68.30_{0.14}$ | $51.14_{2.71}$ | $0.98_{3.40}$ | $32.82_{4.98}$ | $57.55_{2.58}$ | $60.04_{3.10}$ | $51.46_{1.33}$ | $54.49_{15.54}$ | $72.33_{11.85}$ | $77.38_{14.43}$ | $54.25_{4.97}$ |
| SCALEARN | 16 | $52.40_{2.74}$ | $78.07_{0.81}$ | $65.65_{2.08}$ | $73.20_{2.72}$ | $74.23_{6.80}$ | $68.87_{6.25}$ | $69.80_{1.63}$ | $8.55_{4.93}$ | $34.99_{2.74}$ | $58.02_{2.10}$ | $61.63_{2.87}$ | $55.02_{2.52}$ | $54.17_{8.06}$ | $87.67_{2.52}$ | $87.50_{4.72}$ | $61.98_{3.17}$ |
| SCALEARN | 32 | $58.77_{0.93}$ | $80.03_{1.89}$ | $75.40_{0.39}$ | $75.65_{2.42}$ | $82.79_{0.88}$ | $70.02_{1.11}$ | $74.37_{2.53}$ | $4.62_{4.95}$ | $34.93_{1.67}$ | $55.81_{3.58}$ | $69.07_{1.01}$ | $57.11_{2.28}$ | $57.69_{5.00}$ | $84.33_{0.58}$ | $86.31_{4.12}$ | $64.46_{2.42}$ |
| SCALEARN | 100 | $68.52_{1.55}$ | $80.90_{0.52}$ | $85.49_{0.43}$ | $86.77_{0.93}$ | $88.43_{1.06}$ | $76.31_{0.14}$ | $80.75_{0.21}$ | $20.50_{4.11}$ | $56.03_{1.01}$ | $64.86_{0.82}$ | $74.49_{1.05}$ | $56.74_{1.59}$ | $58.97_{3.64}$ | $88.33_{2.08}$ | $91.07_{1.79}$ | $71.88_{1.59}$ |
| SCALEARN | All | $89.67_{0.13}$ | $89.70_{0.58}$ | $93.98_{0.31}$ | $95.36_{0.57}$ | $92.29_{0.13}$ | $88.28_{1.37}$ | $85.78_{1.16}$ | $67.20_{1.33}$ | $85.43_{0.44}$ | $80.08_{0.69}$ | $82.43_{0.79}$ | $70.16_{2.08}$ | $66.73_{4.79}$ | $91.00_{1.22}$ | $93.93_{2.04}$ | $84.80_{1.18}$ |
| SCALEARNUNIFORM | 4 | $49.39_{1.75}$ | $69.28_{2.31}$ | $61.28_{2.03}$ | $52.52_{2.58}$ | $45.14_{7.06}$ | $68.79_{0.28}$ | $51.14_{2.71}$ | $3.23_{4.99}$ | $36.25_{4.01}$ | $57.55_{2.26}$ | $60.52_{2.33}$ | $50.84_{0.96}$ | $58.01_{4.00}$ | $77.67_{1.53}$ | $77.38_{6.76}$ | $54.60_{3.04}$ |
| SCALEARNUNIFORM | 16 | $51.40_{0.97}$ | $77.92_{0.57}$ | $61.71_{2.49}$ | $63.65_{3.38}$ | $67.19_{2.57}$ | $69.20_{0.57}$ | $70.16_{3.79}$ | $5.98_{0.30}$ | $35.41_{7.20}$ | $57.18_{2.58}$ | $60.99_{1.76}$ | $55.43_{2.07}$ | $44.23_{4.41}$ | $80.33_{7.57}$ | $86.90_{3.72}$ | $59.18_{2.93}$ |
| SCALEARNUNIFORM | 32 | $55.41_{1.98}$ | $78.05_{0.92}$ | $66.22_{2.11}$ | $72.55_{0.93}$ | $76.26_{1.07}$ | $70.51_{0.62}$ | $76.41_{0.91}$ | $6.40_{4.69}$ | $35.38_{0.82}$ | $56.19_{3.35}$ | $64.64_{1.91}$ | $56.43_{0.41}$ | $52.88_{6.73}$ | $81.00_{8.89}$ | $86.31_{1.03}$ | $62.31_{2.42}$ |
| SCALEARNUNIFORM | 100 | $62.62_{1.38}$ | $79.81_{0.13}$ | $72.80_{1.78}$ | $82.53_{1.47}$ | $85.25_{0.75}$ | $73.94_{1.39}$ | $80.99_{1.16}$ | $19.37_{1.42}$ | $44.64_{10.32}$ | $59.98_{0.06}$ | $68.80_{1.24}$ | $54.18_{0.79}$ | $57.05_{0.00}$ | $86.33_{2.31}$ | $92.26_{1.03}$ | $68.04_{1.81}$ |
| SCALEARNUNIFORM | All | $90.09_{0.04}$ | $90.54_{0.08}$ | $93.84_{0.69}$ | $95.70_{0.08}$ | $92.13_{0.05}$ | $88.33_{1.12}$ | $85.85_{2.39}$ | $66.85_{1.05}$ | $88.24_{0.00}$ | $80.50_{0.07}$ | $82.04_{0.16}$ | $70.28_{2.47}$ | $59.62_{0.00}$ | $90.40_{1.52}$ | $95.00_{2.33}$ | $84.63_{0.80}$ |
| SCALEARN++ | 4 | $48.69_{4.16}$ | $71.65_{2.79}$ | $55.85_{3.70}$ | $50.96_{0.07}$ | $51.62_{8.06}$ | $68.38_{0.42}$ | $52.71_{0.00}$ | $5.29_{4.21}$ | $32.36_{5.52}$ | $39.12_{8.18}$ | $60.93_{1.34}$ | $51.25_{0.57}$ | $40.06_{2.94}$ | $78.00_{6.24}$ | $80.95_{4.49}$ | $52.52_{3.51}$ |
| SCALEARN++ | 16 | $50.01_{3.62}$ | $76.53_{0.51}$ | $62.43_{0.62}$ | $60.89_{2.58}$ | $67.50_{4.89}$ | $69.44_{1.02}$ | $67.87_{1.57}$ | $6.36_{2.91}$ | $29.36_{1.39}$ | $56.80_{2.88}$ | $60.24_{0.48}$ | $53.81_{3.63}$ | $50.00_{6.93}$ | $83.67_{1.15}$ | $88.10_{2.06}$ | $58.87_{2.42}$ |
| SCALEARN++ | 32 | $56.60_{1.73}$ | $78.04_{0.71}$ | $68.94_{4.68}$ | $73.09_{3.39}$ | $77.90_{1.60}$ | $70.83_{1.70}$ | $75.45_{1.25}$ | $7.49_{2.53}$ | $32.09_{0.95}$ | $59.95_{0.00}$ | $65.77_{0.94}$ | $56.22_{0.96}$ | $45.19_{3.47}$ | $86.00_{1.00}$ | $85.71_{0.00}$ | $62.62_{1.79}$ |
| SCALEARN++ | 100 | $65.59_{1.16}$ | $78.53_{1.78}$ | $76.61_{1.19}$ | $84.25_{1.67}$ | $85.79_{1.40}$ | $73.04_{0.49}$ | $79.90_{0.91}$ | $17.96_{1.29}$ | $53.04_{1.69}$ | $62.62_{0.85}$ | $73.22_{1.32}$ | $56.43_{1.70}$ | $55.45_{4.84}$ | $86.33_{0.58}$ | $88.69_{4.12}$ | $69.16_{1.67}$ |
| SCALEARN++ | All | $90.13_{0.27}$ | $90.22_{0.93}$ | $94.49_{0.23}$ | $94.61_{2.11}$ | $92.35_{0.10}$ | $87.70_{0.96}$ | $86.21_{1.00}$ | $67.23_{1.28}$ | $87.53_{0.13}$ | $80.14_{0.29}$ | $82.51_{1.95}$ | $69.40_{1.63}$ | $62.82_{2.11}$ | $89.80_{1.10}$ | $94.29_{0.80}$ | $84.63_{0.93}$ |
| SCALEARNUNIFORM++ | 4 | $47.67_{3.84}$ | $69.74_{1.51}$ | $61.64_{0.52}$ | $51.87_{0.69}$ | $49.56_{6.92}$ | $69.04_{0.93}$ | $50.54_{3.06}$ | $2.13_{7.37}$ | $32.78_{1.10}$ | $55.54_{1.54}$ | $59.29_{5.56}$ | $51.52_{1.31}$ | $60.26_{6.38}$ | $75.00_{2.00}$ | $82.14_{3.57}$ | $54.58_{5.69}$ |
| SCALEARNUNIFORM++ | 16 | $50.04_{1.94}$ | $77.01_{0.36}$ | $62.54_{2.72}$ | $60.89_{0.34}$ | $68.96_{4.79}$ | $69.04_{0.28}$ | $70.52_{2.80}$ | $5.23_{2.52}$ | $32.66_{6.34}$ | $57.51_{2.73}$ | $60.52_{1.84}$ | $52.61_{2.14}$ | $49.68_{6.01}$ | $79.67_{4.62}$ | $84.52_{4.12}$ | $58.76_{3.06}$ |
| SCALEARNUNIFORM++ | 32 | $54.75_{1.45}$ | $78.77_{0.55}$ | $66.14_{3.06}$ | $74.43_{3.40}$ | $76.21_{1.62}$ | $69.61_{1.27}$ | $77.14_{1.16}$ | $5.71_{1.07}$ | $34.22_{0.99}$ | $55.38_{4.22}$ | $64.24_{0.67}$ | $55.02_{0.72}$ | $45.51_{3.38}$ | $82.00_{5.57}$ | $86.90_{2.06}$ | $61.73_{2.01}$ |
| SCALEARNUNIFORM++ | 100 | $58.14_{1.35}$ | $78.26_{0.89}$ | $68.29_{1.51}$ | $83.18_{0.89}$ | $83.88_{1.89}$ | $74.10_{0.51}$ | $82.31_{1.25}$ | $20.12_{2.05}$ | $46.23_{1.82}$ | $60.40_{0.50}$ | $66.71_{2.50}$ | $53.40_{0.77}$ | $56.73_{2.54}$ | $80.67_{4.16}$ | $91.67_{1.03}$ | $66.94_{1.58}$ |
| SCALEARNUNIFORM++ | All | $90.10_{0.14}$ | $90.45_{0.10}$ | $93.91_{0.09}$ | $95.30_{0.00}$ | $92.12_{0.12}$ | $88.97_{0.88}$ | $84.77_{1.34}$ | $65.83_{1.49}$ | $88.28_{0.56}$ | $80.46_{0.23}$ | $82.23_{0.28}$ | $70.09_{0.36}$ | $60.10_{0.68}$ | $89.20_{1.30}$ | $95.36_{0.98}$ | $84.48_{0.57}$ |

