# OpenReview forum: "ScaLearn: Simple and Highly Parameter-Efficient Task Transfer by Learning to Scale"
_ICLR.cc/2024/Conference — Submitted to ICLR 2024_

### Official Review · Reviewer_6MN5 · 2023-10-28

**Soundness:** 3 good
**Presentation:** 3 good
**Contribution:** 2 fair
**Rating:** 5
**Confidence:** 4

**Summary:**

This paper proposes a new method for task transfer by aggregating the output representations of source task models with some weighted sum, where the weight is either applied on each coordinate or on the entire representation. They claim that this method is more parameter efficient than the previous method which trains a separate adapter for each task, and show that the performance is slightly better.

**Strengths:**

- The idea is simple yet seems to be something that people haven’t tried before.
- The algorithm both improves parameter efficiency and also the performance, which is quite nice.
- The paper is well written and the algorithm details are carefully explained.

**Weaknesses:**

- The improvement in the performance is quite marginal. In table 3, it looks like the proposed method is only slightly better than previous algorithms.
- I believe that the focus of the algorithm is the parameter efficiency. However, I’m not very convinced that this is an important improvement, given that the adapter anyways is already having much fewer parameters than the original models. Isn’t it true that the key bottleneck is the model size itself?
- It would be good if the authors can include experiments beyond NLP tasks to show the generality of their method.

**Questions:**

- In table 3, the version of the ScaLearn that achieves best performance varies a lot across tasks. Why is this the case? Can you provide some intuition around when would each version work better?

---

> ### Author Response · Authors · 2023-11-20
> **Answer to Reviewer 6MN5**
>
> We thank the reviewer for recognizing the novelty and performance of our method, as well as the clarity of our presentation. We appreciate the insightful feedback and address the reviewer's concerns below.
> - **[W1] Performance improvement of ScaLearn**: We would like to emphasize that the performance of ScaLearn is considerably better than the baselines at a much lower or similar parameter count. While the performance on high-resource tasks (QQP, MNLI) is similar to some other methods (since the source adapter can learn the task well), this is considerably different for many other tasks: This is especially the case on SuperGLUE, as shown in Table 3 - our ScaLearn beats the best baseline by ~2 percentage points (!), and this is especially visible on the low-resource tasks - notably CB (90.89 vs. 87.68 for the best baseline, Compacter++) and RTE (78.88 vs. 76.53 for the best baseline, Adapter-m). In addition, on the very different multilingual benchmark HumSet, ScaLearn is the only (!) method that clearly beats any of the single-task learning methods. We attribute this to the simplicity and power of scaling output representations. Furthermore, we also show considerably improved few-shot learning abilities (Section 6.3).
> - **[W2] Significance of parameter-efficiency**: We agree that our method provides strong parameter-efficiency and that the original backbone model typically has more parameters. However, especially since models tend to get larger and larger, it is important to be able to efficiently adapt the model to novel domains, tasks, and languages and mitigate catastrophic interference. The importance of our problem setup and efficiency is also highlighted by reviewer YAiw. This strongly motivates a modular design of such models. In this paradigm, the parameter-efficiency of PEFT (parameter-efficient fine-tuning) has become paramount, especially when scaling up the number of tasks and, thus, modules and corresponding parameter counts [1]. Therefore, this is a crucial aspect of any method within this paradigm. This is especially the case in two-stage MTL methods - and in this regime, ScaLearn, and especially its variations, is highly parameter-efficient while also clearly outperforming the baselines. For an overview of the parameter-efficiency of ScaLearn and all baselines, we refer the reviewer to Figure 1 on page 1 and the parameter-efficiency analysis (Section 6.1).
> - **[W3] Experiments beyond NLP tasks**: We understand the reviewer's interest in applying our method to other contexts beyond NLP. In this work, our focus is on showcasing its strong performance and efficiency in the context of NLP. In doing that, we follow numerous other papers that equally focus on NLP tasks [2, 3, 4]. We have shown the performance of our method on a diverse number of NLP tasks, domains, and languages. Yet, we thank the reviewer for the idea, and it comprises an interesting direction for future research.
> - **[Q1] Variability in ScaLearn\* performance**: Overall, the performance of the different variations of ScaLearn is rather similar. Moreover, the performance of ScaLearn and ScaLearn++ tends to be highly similar on most tasks (cf. Table 2, 3, and 4), with both performing the best overall. The performance of the variants relying on uniform scaling—ScaLearnUniform and ScaLearnUniform++—also tends to be highly competitive on most tasks and is generally a bit lower than the non-uniform variants, but still very strong. We attribute this to the difference in scaling the output representations element-wise or not (i.e., non-uniform or uniform scaling) and the corresponding difference in learning capacity. For the best performance, we advise using the vanilla ScaLearn or ScaLearn++, and if the parameter count is the main constraint, we advise using ScaLearnUniform++.
>
> [1] Modular Deep Learning. arxiv preprint, 2023.
>
> [2] Few-shot parameter-efficient fine-tuning is better and cheaper than in-context
> learning. NeurIPS, 2022.
>
> [3] Adapter-Fusion: Non-destructive task composition for transfer learning. EACL, 2021.
>
> [4] Conditionally adaptive multi-task learning: Improving transfer learning in NLP using fewer parameters & less data. ICLR, 2023.

---

> > ### Author Response · Authors · 2023-11-22
> >
> > Dear Reviewer 6MN5,
> >
> > We are wondering if our response and revision have resolved your concerns. In the revised version and rebuttal response, we have conducted additional experiments and revised the manuscript, improving clarity and further highlighting the benefits of our method.
> >
> > If our response has addressed the reviewer’s concerns, we would highly appreciate it if you could re-evaluate our work and consider raising the score.
> >
> >
> > Should there be any further questions or suggestions, please let us know. We are open to continued dialogue to enhance the quality of our research.
> >
> >
> > Kind regards,
> >
> > The Authors

---

### Official Review · Reviewer_YAiw · 2023-10-29

**Soundness:** 2 fair
**Presentation:** 2 fair
**Contribution:** 2 fair
**Rating:** 5
**Confidence:** 4

**Summary:**

The paper proposes a method for transferring previously learned knowledge (in the form of adaptors) from different source tasks to a given target task. The main idea of the paper is to propose a new method that is more parameter-efficient compared to the previous methods. Their method ScaLEARN learns a vector that scales the outputs of the adaptors of the source tasks and then adds these outputs together before passing them to the next layer. They propose some variants of this method that share these activation scaling parameters and across layers, or across the dimension to reduce the parameter count even further. They compare their work with AdaptorFusion methods from 2021.

**Strengths:**

S1. The problem being studied in the paper is becoming increasingly important in the scenarios of large language models.

S2. The method reduces the parameter requirement compared to the AdaptorFusion paper. Moreover, some variants of the method (ScaLearnUniform++) are extremely parameter efficient while they perform similarly to adaptor-fusion and the basic method ScaLearn.

**Weaknesses:**

W1. The experimental section is not very strong and is missing some very strong baselines, relevant settings, and analysis. Please refer to the questions below.

W2. The paper is slightly harder to read, the motivation and the analysis on scaling part was not very clear to me until I read section 4 about the method.

**Questions:**

**For Me to Improve My Score (Most to least important)**

Q1: The only modular composition baseline in this paper is AdaptorFusion from 2021, which is quite old now and there have been other works that tackle the same problem from different angles that should act as baselines here. [1] Combining Modular Skills in Multitask Learning, [2] AdapterSoup: Weight Averaging to Improve Generalization of Pretrained Language Models, and [3] Multi-Head Adapter Routing for Cross-Task Generalization. And the baseline methods used in these papers.

Q2: The experimental setup used in the paper can be significantly improved by studying this problem on good seq2seq zero-shot models like T0, or maybe LLaMA family models and then comparing the zero-shot performance, few-shot performance, and the performance obtained via this kind of composition of learned modules.

Q3: How is the classification layer for the target task finetuned? In Table 1 and everywhere in the paper, it seems like you do not count these parameters when counting for the number of trainable parameters. Can you clarify this if the classifier parameters are also learned on each source task then this needs to be clarified throughout the paper.



**Other Questions**

Q4: The experiments in the paper should add IA3 () as a baseline. IA3 paper shares very high similarity to the proposed method, the ScaLEARN method is like adapting source task adapter modules using IA3 on a downstream task. Hence, for all the experiments, IA3 would be a good baseline as it would learn a lower number of parameters than ScaLearn. However, I don't suspect it to perform better than the presented method as it is leveraging source adaptors.

Q5: At multiple places, the paper talks about how the scaling coefficient does not need to sum to 1 and I agree that for ScaLearn this might be the case however, I am not sure if there is enough evidence in the paper, to claim that this is how it should be for all other methods and this has been talked about at multiple places in the paper as a new finding. It can be that for the other methods having a constraint on summing to 1 might be better than not having it. Hence, this is a method-specific detail and should not be portrayed as a general finding that we should not have a summation constraint. I might have missed something here and I will willing to take this comment back in light of evidence.


I looking forward to the rebuttal and will update my score if some of my main concerns are addressed. I really like the simplicity of the method however I would like it to be more rigorously tested against other baselines and experimental setups.

---

> ### Author Response · Authors · 2023-11-20
> **Results on additional baselines (AdapterSoup and (IA)^3)**
>
> We thank the reviewer for acknowledging the importance, simplicity, and merits of our method. We appreciate the insightful feedback and address each of the concerns below.
> - **[W1] Experimental section and baselines**: We acknowledge the reviewer's emphasis on the importance of a diverse evaluation to demonstrate the merit of our method. We have already included a broad range of strong single-task learning (STL) and multi-task learning (MTL) baselines, and we appreciate the reviewer's perspective on this matter. In response, we have included the PEFT method (IA)^3 [4] and AdapterSoup [2] (adapted to the two-stage MTL framework) in our experiments.
>
> RTable 1: Performance on GLUE using RoBERTa-base.
> | Model                 | MNLI  | QQP   | QNLI  | SST-2 | STS-B | MRPC  | RTE   | CoLA  | Avg.  |
> |-----------------------|-------|-------|-------|-------|-------|-------|-------|-------|-------|
> | Adapter               | 86.50 | 90.18 | 92.25 | 93.65 | 90.23 | 86.64 | 72.89 | 58.28 | 83.83 |
> | ProPETL               | 86.19 | 88.88 | 92.05 | 93.81 | 90.03 | 85.93 | 74.19 | **59.29** | 83.80 |
> | Compacter++           | 85.62 | 88.84 | 91.79 | 93.58 | 89.67 | 87.21 | 72.02 | 58.49 | 83.40 |
> | *(IA)^3*               | 83.78 | 88.37 | 90.57 | 93.35 | 89.93 | 87.11 | 72.56 | 56.57 | 82.78
> | *AdapterSoup* | 63.47 | 81.63 | 78.00 | 90.75 | 80.17 | 75.00 | 62.09 | 41.06 | 71.52 |
> | ScaLearn              | **86.97** | **90.32** | **92.51** | **93.88** | **90.96** | **87.75** | **82.06** | 58.47 | **85.36** |
>
> RTable 2: Performance on SuperGLUE.
>
> | Model                      | ReCoRD | MultiRC | BoolQ  | WiC    | WSC    | COPA   | CB     | RTE    | Avg.   |
> |----------------------------|--------|---------|--------|--------|--------|--------|--------|--------|--------|
> | Adapter                    | 79.02  | 72.84   | 76.71  | 65.58  | **63.46**  | 70.20  | 84.82  | 72.89  | 73.19  |
> | ProPETL                    | **80.29** | 73.07   | 76.58  | 66.60  | **63.46**  | 70.60  | 84.46  | 74.19  | 73.69  |
> | Compacter++                | 77.69  | 70.44   | 75.88  | 66.46  | **63.46**  | 68.30  | 87.68  | 72.02  | 72.74  |
> | *(IA)^3*                     | 75.27  | 70.32   | 76.31  | **67.07**  | 63.35  | 69.30  | 87.32  | 72.56  | 72.69  |
> | *AdapterSoup* | 64.26 | 33.62 | 68.84 | 58.53 | 63.46 | 52.40 | 70.89 | 57.83 | 58.73 |
> | ScaLearn            | 79.52  | **73.22** | **77.27** | 66.35  | **63.46** | **74.80**  | **90.89**  | **78.88**  | **75.55**  |
>
> RTable 3: Performance on HumSet using XLM-R-base.
>
> | Model                   | Sectors | Pillars 1D | Subpillars 1D | Pillars 2D | Subpillars 2D | Avg.  |
> |-------------------------|---------|------------|---------------|------------|---------------|-------|
> | Adapter                 | 71.38   | 51.02      | 43.26         | 61.43      | 42.46         | 53.91 |
> | ProPETL                 | 71.69   | 49.69      | 41.63         | 60.58      | 39.85         | 52.69 |
> | Compacter++             | 69.97   | 37.37      | 37.76         | 58.13      | 33.10         | 47.26 |
> | *(IA)^3*                  | 70.22   | 45.55      | 40.05         | 58.54      | 39.27         | 50.73 |
> | *AdapterSoup*             | 56.81   | 30.09      | 21.84         | 40.71      | 17.89         | 33.47 |
> | ScaLearn         | **72.36** | **51.63**  | **44.06**     | **61.52**  | **42.81**     | **54.48** |
>
> RTable 1, 2, and 3 demonstrate that our ScaLearn model outperforms the newly introduced baselines.
>
> Regarding (IA)^3, it performs similarly but slightly worse than other STL methods in our setup. We attribute this difference to our different experimental setup, including tasks, dataset sizes, and PLMs, which vary meaningfully from the original (IA)^3 paper [4]. Aside from the original study, hardly any comparisons of (IA)^3 with other PEFT techniques exist; the closest is [5], which is aligned with our findings.
>
> Regarding AdapterSoup, we adapted it to the two-stage MTL framework by adding the target task corpus and, hence, target adapter to the similarity calculation, and also train a new task head on every target task. Still, its comparatively worse performance suggests that calculating weights using sentence similarity is not the best fit for our specific problem setup.
>
> We incorporated the results of both as an additional baseline in the revised draft. This addition, we believe, further strengthens our evaluation and addresses the reviewer's main concerns, and reflects our commitment to a thorough and inclusive comparative analysis.
>
> [1] Combining parameter-efficient modules for task-level generalisation. EACL, 2023.
>
> [2] AdapterSoup: Weight averaging to improve generalization of pretrained language models. EACL, 2023.
>
> [3] Multi-head adapter routing for data-efficient fine-tuning. arxiv preprint, 2023.
>
> [4] Few-Shot Parameter-Efficient Fine-Tuning is Better and Cheaper than In-Context Learning. NeurIPS, 2022.
>
> [5] https://adapterhub.ml/blog/2022/09/updates-in-adapter-transformers-v3-1/#ia3

---

> > ### Author Response · Authors · 2023-11-20
> > **General answer to Reviewer YAiw**
> >
> > - **[W2] Readability**: We thank the reviewer for raising this point. We have walked through the paper to ensure clarity and readability throughout. Based on this, we have made a number of alterations (highlighted in blue), including an introductory sentence to our analyses (Section 3).
> >
> > Concerning the questions:
> > - **[Q1] Modular composition baseline**: We acknowledge the inclusion of additional baselines. To address the reviewer's concerns, we have included two additional baselines (see first comment). Regarding the proposed baselines, neither of them tackles the very same problem; all the proposed works are generally related to knowledge composition and MTL, but neither of them shares the same characteristics as our method, which is what we term a "two-stage MTL" method.
> >     - [1] explicitly relies on a multi-task pre-training stage, aiming to learn modular, general skills and a task-skill matrix jointly with the aim of cross-task transfer.
> >     - [3] further expands and improves on this idea, otherwise sharing the general setup and characteristics of [1], relying on a multi-task pre-training stage.
> >     - Hence, we would rather consider both as "joint MTL" architectures. The conceptual similarity of [1] to HyperFormer is explicitly mentioned in [1] on page 4. We already include HyperFormer as a strong modular joint MTL baseline in our results, in addition to several other joint MTL baselines. Thus, [1] and [3] are of a different paradigm to our method and do not offer the benefits of the two-stage MTL methods that we raise throughout the paper, promoting reusability and addressing cases involving data privacy and societal concerns. Crucially, ScaLearn does not need multiple source corpora in the second stage. However, we acknowledge the relevance and impact of these works by mentioning them in our Related Work (Section 7).
> >     - Regarding AdapterSoup [2], the aim is to improve performance on unseen domains by weight-space averaging the adapter parameters. It does so by calculating similarities across target and source corpora but without additional training on the target task. Therefore, it is also conceptually different from our method, since there is no training involved in the composition phase. Nonetheless, we included results of an adapted version of AdapterSoup as a two-stage MTL method as another baseline in the revised draft and in the answer above.
> > - **[Q2] Experiments with seq2seq models**: We acknowledge the potential of expanding our method to seq2seq models. Our current focus is on encoder PLMs, a point that we make more explicit in the revised draft. This focus was chosen based on our initial analyses of scaling the output representations of adapters, and it aims to provide a clear and concise study within the scope of this paper, also in terms of computation - we are running each of the methods on multiple seeds and two PLMs to ensure reliability, leading to a very high number of experiments. While this is a valid limitation of our study, we leave the exploration of expanding our method to seq2seq models for future work.
> > - **[Q3] Clarification on the classification layer**: Indeed, the classifier (task head) parameters are learned on each target task (also for two-stage MTL methods), but we do not count them when considering the number of trainable parameters. We chose not to include them to focus on the efficiency of the main method, as the classifier layer (task head) represents a standard component across  setups. In response to the reviewer's feedback, we have clarified this in the revised draft. We thank the reviewer for raising this issue and believe it will enhance the clarity of our experiment setup.
> > - **[Q4] Adding (IA)^3 [4] as a baseline**: We thank the reviewer for raising this point. Indeed, the approach of scaling key and value as well as the intermediate feed-forward output representations as done in (IA)^3 is conceptually related to ScaLearn. As suggested by the reviewer, we also added (IA)^3 as an additional baseline. Since it is commonly applied in a single-task learning (STL) setting like other PEFT methods, we do the same in our experiments. We added the results to the revised draft and to the above answer.
> > - **[Q5] Scaling coefficient summation constraint**: The insights gained in our Analysis (Section 3) provide the motivation for learning scaling coefficients without imposing distributional constraints, and we do not think we claim that this is how it should be in other methods. Rather, our point is that enforcing the scaling coefficients to sum up to 1 can be restraining for ScaLearn, as shown in an ablation study (Appendix A.3, Table 7, page 21), but not necessarily so for other methods. We recognize that transfer learning methods vary greatly, and what holds true for ScaLearn may not be universally applicable. If there are instances where our claims appear overly general, we invite the reviewer to point them out so we can make the necessary corrections.

---

> > > ### Comment · Reviewer_YAiw · 2023-11-21
> > > **Updating my score**
> > >
> > > Hi, I have read the response and I am updating my score to 5. However I still feel experiments with bigger models, better baselines would help the method significantly.

---

> > > > ### Author Response · Authors · 2023-11-22
> > > >
> > > > We appreciate the feedback and thank the reviewer for updating their score.
> > > > Best regards,
> > > > The Authors

---

### Official Review · Reviewer_tpyz · 2023-10-30

**Soundness:** 2 fair
**Presentation:** 3 good
**Contribution:** 2 fair
**Rating:** 5
**Confidence:** 3

**Summary:**

This article presents a new two-stage multitask learning method called SCALEARN, which achieves knowledge transfer by learning the output representation of scaling the source task. The proposed approach achieves high parameter efficiency and strong performance, and can avoid problems such as task interference and data privacy. The authors conduct extensive experiments on three benchmark datasets, e.g., GLUE, SuperGLUE, and HumSet, using RoBERTa and XLM-R as pre-trained language models. The experimental results demonstrate that ScaLearn and its variants outperform strong baselines on various tasks, and also perform well in a few-shot setting.

**Strengths:**

- ScaLearn achieves transfer learning on downstream tasks by scaling and combining the output representations of the source task adapter. This approach has high scalability and low parameter overhead. The idea is simple and technically sound.
- In the third section of the paper, the authors analyze the effects of scaling output representations in transfer learning: scaling output vectors is an effective method for controlling the (partial or full) activation of the knowledge contained in an adapter module and the optimal weights do not necessarily sum up to 1. This provides a new approach for “how to leverage modular knowledge to learn new tasks”. The investigation effectively supports the paper’s claims.
- Experiments are very extensive and thorough, including a variety of tasks, architectures, datasets (GLUE, SuperGLUE, HumSet), and strong baselines beyond vanilla dropout. Both standard transfer learning and few-shot transfer learning are reported, accompanied by in-depth ablation studies. The experiment and analysis are well organized.

**Weaknesses:**

- Analysis limited to adapter-based methods. Unclear how well it will perform to other PEFT architectures (e.g. Prompt Tuning).
- To my best knowledge, IA3 [1] achieves stronger fine-tuning performance by scaling the weighted activations in the activation layer using learned vectors. This is similar to your method, but I did not find it in the PEFT baselines you compared.
- It seems that many source tasks are closely related to each other. I would suggest authors use benchmarks such as CrossFit [2] to do a more large-scale analysis, where the transferring is more challenging as some source tasks can be relatively less related to the target tasks.
- For the second stage during training, the output representations of multiple source adapters are scaled and combined, which reminds me of MoE (Mixture of Experts), where each source adapter corresponds to an expert. It is a well-known phenomenon that learnable MoE can lead to overfitting and collapse. However, in your method, it seems that this issue does not arise. Could the authors explain the specific reasons behind this?

---

[1] Few-Shot Parameter-Efficient Fine-Tuning is Better and Cheaper than In-Context Learning. NeurIPS 2022

[2] CrossFit: A Few-shot Learning Challenge for Cross-task Generalization in NLP. EMNLP2021

**Questions:**

Please respond to the concerns listed in weaknesses.

---

> ### Author Response · Authors · 2023-11-20
> **General answer to Reviewer tpyz**
>
> We thank the reviewer for their thoughtful feedback and for recognizing the technical soundness and thoroughness of the experiments. We appreciate the constructive feedback and address each of the concerns raised in the review below.
> - **[W1] Focus on adapter-based methods**: We acknowledge the reviewer's observation regarding the focus on adapter-based methods. This focus was intentional, following Pfeiffer et al. [1]. Our method is strongly motivated by our analyses of scaling the output representations of adapters (Section 3). Furthermore, we chose to concentrate on adapters due to the wide availability of pre-trained adapters via AdapterHub [2], which facilitates accessibility and reproducibility in the research community. To the best of our knowledge, such sharing platforms are currently not available for other PEFT architectures. However, these are crucial for two-stage MTL methods since they rely on readily available modules (in our case, adapters). Moreover, we justify the use of adapter modules to enable a fair comparison with the other two-stage MTL baseline - AdapterFusion. Additionally, as suggested by Reviewer YAiw, we added AdapterSoup as an additional adapter-based two-stage MTL baseline. In addition, we understand that other PEFT architectures like LoRA and prompt tuning operate differently, and, hence, these methods are not directly compatible with our method, which scales the output representations of the source modules. While this is a valid limitation of our study, we leave the exploration of expanding our method to other PEFT techniques for future work.
> - **[W3] Use of different benchmarks such as CrossFit [5]**: We agree with the importance of diversity in target and source tasks as well as large-scale analyses. To address diversity, we included SuperGLUE and HumSet in addition to the commonly used GLUE benchmark. SuperGLUE presents considerably more challenging tasks than GLUE, which are also more varied in terms of tasks and corpora sizes. We relied on GLUE and SuperGLUE as we wanted to use benchmarks that are most commonly used within the research community. To include a more distinct benchmark, we added additional experiments on the multilingual, multi-class classification benchmark HumSet, which spans the domain of human crisis response, making it very different from both GLUE and SuperGLUE. Using a different backbone model, XLM-R, the similarly strong results on HumSet showcase our method's applicability across different models, languages, and task formulations In addition, we evaluate our method on the combination of all GLUE and SuperGLUE tasks and show that it also outperforms the baselines at a fraction of the parameter cost when considering a higher number of source tasks. This shows that our method also works across more dissimilar tasks and when scaling up the number of source tasks, too. These challenging and different settings demonstrate the robustness and adaptability across different tasks, domains, and languages of our method. Still, we recognize the potential of including large-scale benchmarks like CrossFit in future work to further validate our method under even more different conditions.
> - **[W4] Comparison to MoEs and overfitting**: We appreciate the insightful comment and acknowledge the conceptual similarity between the scaling in our method and Mixture of Experts (MoEs) models. A key difference between our method and MoEs is that, in our setup, the backbone language model remains frozen during both stages of training. This inherently limits the learning capacity compared to fully fine-tuning the entire model, thereby reducing the risk of overfitting—a common issue with more complex models like MoEs. This effect is also observed by Lian et al. [6] when proposing SSF (Scaling & Shifting your Features): "*Such improvements in robustness and OOD [out-of-domain] datasets might come from the fact that SSF freezes most of the pre-trained parameters, which maximally preserves the knowledge learned from the large-scale dataset and thus maintains a better generalization ability.*" Overall, we recognize the conceptual parallels with MoEs. Our approach avoids this phenomenon by keeping the backbone model frozen and, thus, sidestepping issues like overfitting and collapse, ensuring robust and generalizable performance across a variety of tasks.
>
> [1] AdapterFusion: Non-destructive task composition for transfer learning. EACL, 2021.
>
> [2] https://adapterhub.ml/
>
> [3] Few-Shot Parameter-Efficient Fine-Tuning is Better and Cheaper than In-Context Learning. NeurIPS, 2022.
>
> [4] https://adapterhub.ml/blog/2022/09/updates-in-adapter-transformers-v3-1/#ia3
>
> [5]  CrossFit: A Few-shot Learning Challenge for Cross-task Generalization in NLP. EMNLP, 2021
>
> [6] Scaling & shifting your features: A new baseline for efficienmodel tuning. NeurIPS, 2022.

---

> > ### Author Response · Authors · 2023-11-20
> > **Results on (IA)^3**
> >
> > **[W2] Comparison with (IA)^3 [3]**: We thank the reviewer for raising this point. Indeed, the approach of scaling key and value as well as the intermediate feed-forward output representations as done in (IA)^3 is conceptually related to ScaLearn. As suggested by the reviewer, we performed new experiments using (IA)^3. Since (IA)^3 is commonly applied in a single-task learning (STL) setting like other PEFT methods, we do the same in our experiments.
> >
> >
> > RTable 1: Performance on GLUE using RoBERTa-base.
> >
> > | Model                 | MNLI  | QQP   | QNLI  | SST-2 | STS-B | MRPC  | RTE   | CoLA  | Avg.  |
> > |-----------------------|-------|-------|-------|-------|-------|-------|-------|-------|-------|
> > | Adapter               | 86.50 | 90.18 | 92.25 | 93.65 | 90.23 | 86.64 | 72.89 | 58.28 | 83.83 |
> > | ProPETL               | 86.19 | 88.88 | 92.05 | 93.81 | 90.03 | 85.93 | 74.19 | **59.29** | 83.80 |
> > | Compacter++           | 85.62 | 88.84 | 91.79 | 93.58 | 89.67 | 87.21 | 72.02 | 58.49 | 83.40 |
> > | *(IA)^3*               | 83.78 | 88.37 | 90.57 | 93.35 | 89.93 | 87.11 | 72.56 | 56.57 | 82.78
> > | ------                | - | - | - | - | - | - | - | - | - |
> > | ScaLearn              | **86.97** | **90.32** | **92.51** | **93.88** | **90.96** | **87.75** | **82.06** | 58.47 | **85.36** |
> >
> >
> > RTable 2: Performance on SuperGLUE.
> >
> > | Model                      | ReCoRD | MultiRC | BoolQ  | WiC    | WSC    | COPA   | CB     | RTE    | Avg.   |
> > |----------------------------|--------|---------|--------|--------|--------|--------|--------|--------|--------|
> > | Adapter                    | 79.02  | 72.84   | 76.71  | 65.58  | **63.46**  | 70.20  | 84.82  | 72.89  | 73.19  |
> > | ProPETL                    | **80.29** | 73.07   | 76.58  | 66.60  | **63.46**  | 70.60  | 84.46  | 74.19  | 73.69  |
> > | Compacter++                | 77.69  | 70.44   | 75.88  | 66.46  | **63.46**  | 68.30  | 87.68  | 72.02  | 72.74  |
> > | *(IA)^3*                     | 75.27  | 70.32   | 76.31  | **67.07**  | 63.35  | 69.30  | 87.32  | 72.56  | 72.69  |
> > | ------                     | -      | -       | -      | -      | -      | -      | -      | -      | -      |
> > | ScaLearn            | 79.52  | **73.22** | **77.27** | 66.35  | **63.46** | **74.80**  | **90.89**  | **78.88**  | **75.55**  |
> >
> >
> > RTable 3: Performance on HumSet using XLM-R-base.
> >
> > | Model                   | Sectors | Pillars 1D | Subpillars 1D | Pillars 2D | Subpillars 2D | Avg.  |
> > |-------------------------|---------|------------|---------------|------------|---------------|-------|
> > | Adapter                 | 71.38   | 51.02      | 43.26         | 61.43      | 42.46         | 53.91 |
> > | ProPETL                 | 71.69   | 49.69      | 41.63         | 60.58      | 39.85         | 52.69 |
> > | Compacter++             | 69.97   | 37.37      | 37.76         | 58.13      | 33.10         | 47.26 |
> > | *(IA)^3*                  | 70.22   | 45.55      | 40.05         | 58.54      | 39.27         | 50.73 |
> > | ------                  | -       | -          | -             | -          | -             | -     |
> > | ScaLearn         | **72.36** | **51.63**  | **44.06**     | **61.52**  | **42.81**     | **54.48** |
> >
> >
> >
> >
> > RTable 1, 2, and 3 show that (IA)^3 slightly underperforms other STL methods in our setup. We attribute this difference to our different experimental setup, including tasks, dataset sizes, and backbone language models, which vary meaningfully from the original (IA)^3 paper [3]. Aside from the original study, hardly any comparisons of (IA)^3 with other PEFT techniques exist; the closest is [4], which is aligned with our findings.
> >
> > We incorporated the results of (IA)^3 as an additional baseline in the revised version of the paper, supplementing our comparative analysis. In addition, we added another baseline, AdapterSoup. We are committed to providing a robust and comprehensive evaluation of ScaLearn against existing methods, as requested, and we hope that these efforts will be reflected in the revised score.

---

> > > ### Comment · Reviewer_tpyz · 2023-11-22
> > > **Response**
> > >
> > > I appreciate the authors' responses to some of my questions, which provide additional experimental support. However, the parameter efficiency that you emphasized needs to be better demonstrated on benchmarks that include more tasks. In addition, I still believe that evaluating this method on benchmarks that include more tasks would better demonstrate its robustness and adaptability. Therefore, I maintain the original score unchanged.

---

> > > > ### Author Response · Authors · 2023-11-22
> > > >
> > > > We thank the reviewer for their valuable feedback. We acknowledge the points made and would like to reiterate that our earlier response has incorporated a large range of experiments across the three benchmarks of GLUE, SuperGLUE, and HumSet, spanning highly diverse task formulations (many of which are highly unrelated to each other, e.g., NLI and QA), dataset sizes (250–several 100k), and even languages.
> > > >
> > > > We show that ScaLearn outperforms not only the initial baselines across all benchmarks but also another two-stage MTL method, AdapterSoup, as requested by reviewer YAiw, and (IA)^3, as requested by the reviewer themselves. This is consistently the case among all benchmarks and in the few-shot setting, reminiscent of the initially proposed CrossFit benchmark that also focuses on few-shot adaptation. Moreover, other recently proposed works with a similar problem setup to ours do not consider more tasks either [2, 3, 4, 5, 6].
> > > > Hence, we believe that our current experiment setup reflects a fair and broad comparison of different methods.
> > > >
> > > > We appreciate any additional insights and are open to any further guidance or suggestions.
> > > >
> > > > Best regards,
> > > >
> > > > The Authors
> > > >
> > > >
> > > >
> > > >
> > > > [1] Few-Shot Parameter-Efficient Fine-Tuning is Better and Cheaper than In-Context Learning. NeurIPS, 2022.
> > > >
> > > > [2] Multitask Prompt Tuning Enables Parameter-Efficient Transfer Learning. ICLR, 2023.
> > > >
> > > > [3] Conditionally Adaptive Multi-Task Learning: Improving Transfer Learning in NLP Using Fewer Parameters & Less Data. ICLR, 2021.
> > > >
> > > > [3] ATTEMPT: Parameter-Efficient Multi-task Tuning via Attentional Mixtures of Soft Prompts. EMNLP, 2022.
> > > >
> > > > [4] Parameter-efficient Multi-task Fine-tuning for Transformers via Shared Hypernetworks. ACL, 2021.
> > > >
> > > > [5] One Network, Many Masks: Towards More Parameter-Efficient Transfer Learning. ACL, 2023.
> > > >
> > > > [6] AdapterFusion: Non-Destructive Task Composition for Transfer Learning. EACL, 2021.

---

### Official Review · Reviewer_Exzr · 2023-11-01

**Soundness:** 4 excellent
**Presentation:** 3 good
**Contribution:** 2 fair
**Rating:** 8
**Confidence:** 4

**Summary:**

The paper introduces a new method for two-stage parameter-efficient multi-task learning called ScaLearn. It builds on prior work that fusing Adapters, and shows that a simpler parameterization of the source-task mixing module works well. It does this by replacing the attention mechanism in (Pfeiffer et al)[https://arxiv.org/abs/2005.00247] with a simpler task-vector or task-scalar parameterization.

**Strengths:**

- The paper is clearly written, and is generally easy to follow.
 - The method is simple, well-motivated, and works well.
 - In my opinion, the method proposed in this paper should have been a baseline in the original [Pfeiffer et al](https://arxiv.org/abs/2005.00247) paper that introduced AdapterFusion. Unfortunately it wasn't, so this paper is valuable in that it shows that attention could be unnecessary to fuse adapters, and simple task-vectors or task-scalars may be sufficient.

**Weaknesses:**

- I don't think this is a glaring weakness, but I do think the paper could benefit from more diverse source/target tasks, especially sequence-generation tasks. It could be possible that the simpler ScaLearn parameterization doesn't work as well for different configurations of source and target tasks. I don't particularly see why *both* GLUE and SuperGLUE had to be included, it might have been better to replace one of them with a different benchmark if resources are/were the constraint.
 - Some paragraphs are very long and hard to parse (e.g. "Models and Baselines" on page 6), and could be written in a more organized manner in my opinion.

**Questions:**

- Did the authors try other parameterizations of the fusion module? For example, a low-rank MLP per task instead of a vector per task would be a step towards finding a sweet spot (if it exists) between attention-based fusing and ScaLearn. It's also not clear to me whether a task-vector-per-layer would be better than an MLP-shared-across-layers. This paper essentially shows that what should have been a baseline in [Pfeiffer et al](https://arxiv.org/abs/2005.00247) works very well, which is valuable, but I think expanding the study to include a larger range of parameterizations (starting from ScaLearnUniform++ → low-rank MLPs → efficient attention variants) would make this paper much stronger and be a conclusive piece of work on fusing Adapters. Of course this is not necessary to be a solid paper, but definitely worth considering and discussing in the paper in my opinion.

---

> ### Author Response · Authors · 2023-11-20
>
> We thank Reviewer Exzr for their positive feedback on our method and for acknowledging the clarity of the paper. We appreciate the reviewer’s positive feedback and constructive suggestions. Below, we address each of the comments and concerns in detail.
>
> - **[W1] Selection of source and target tasks**: We agree with the importance of diversity in target and source tasks. We included SuperGLUE for this very purpose, as it presents considerably more challenging tasks than GLUE, which are also more varied in terms of tasks and corpora sizes. We relied on GLUE and SuperGLUE as we wanted to use benchmarks that are most commonly used within the research community. To include a more distinct benchmark, we added additional experiments on the multilingual, multi-class classification benchmark HumSet, which spans the domain of human crisis response, making it very different from both GLUE and SuperGLUE. Using a different backbone model, XLM-R, the similarly strong results on HumSet showcase our method’s applicability across different models, languages, and task formulations. In addition, we evaluate our method on the combination of all GLUE and SuperGLUE tasks and show that it equally outperforms the baselines at a fraction of the parameter cost when considering a higher number of source tasks as well. These challenging and different settings demonstrate the robustness and adaptability across different tasks, domains, and languages of our method.
> - **[W2] Paragraph length and clarity**: We thank the reviewer for highlighting the importance of clarity in our paper. In response to the reviewer's valuable feedback, we have thoroughly revised the "Models and baselines" section on page 6 to enhance its readability. Additionally, we have reviewed the entire manuscript to ensure clarity in all sections. We appreciate any further specific feedback on sections that could benefit from increased clarity.
> - **[Q1] Exploration of other parametrizations**: We acknowledge the idea of a broader study encompassing a larger range of parametrizations; however, it would meaningfully deviate from the core contribution of the paper. Our experiments on scaling the output representations of adapters have provided a strong motivation for learning the scaling coefficients to combine different adapter output representations. This has led to our proposed method, ScaLearn, and its variations, showing improved performance and efficiency compared to previous methods.

---

### Meta-Review · Area_Chair_UnHP · 2023-12-08

**Metareview:**

This paper attacks the problem of "two-stage multitask learning", where a set of task-specific models (which are created by the addition of adapters or other parameter-efficient modules) are leveraged to achieve good performance on a target task via a transfer layer that introduces a small number of parameters. The primary innovation of this paper compared to past papers is a suite of new transfer layers called "ScaLearn" that linearly scale and combine the output of adapters. The primary benefit of this is a small boost in performance and, in some cases, a significant reduction in the total number of task-specific parameters that are ultimately learned. Experiments are done on older, and small by today's standards, pre-trained LMs on GLUE, SuperGLUE, and a relatively uncommon benchmark HumSet, where ScaLearn is shown to outperform AdapterFusion by a meaningful amount. The primary issues with this paper lie in the experimental setup - experiments primarily focus on out-of-date language models and out-of-date benchmarks. Initially, the paper was also missing important baselines, though the authors resolved that during the rebuttal. In addition, the ultimate contribution of this paper is quite small - it could be considered a simplification of AdapterFusion. While simplification is important, the rest of the paper (specifically, the experiments) are weak enough that it should not be accepted in its current form.

**Justification For Why Not Higher Score:**

As discussed above, the paper's contribution is somewhat small and the experimental setting is out-of-date. If the paper had presented a new and more compelling experimental setting, the contribution might have been more substantial.

**Justification For Why Not Lower Score:**

N/A

---

### Decision · Program_Chairs · 2024-01-16

Reject